# *Emx2* underlies the development and evolution of marsupial gliding membranes

Jorge A. Moreno[1,16,18], Olga Dudchenko[2,3,18], Charles Y. Feigin[1,4,17], Sarah A. Mereby[1], Zhuoxin Chen[5], Raul Ramos[5], Axel A. Almet[6,7], Harsha Sen[1], Benjamin J. Brack[1], Matthew R. Johnson[1], Sha Li[1], Wei Wang[8], Jenna M. Gaska[1], Alexander Ploss[1], David Weisz[2], Arina D. Omer[2], Weijie Yao[2], Zane Colaric[2], Parwinder Kaur[9], Judy St. Leger[10], Qing Nie[5,6,7,11], Alexandria Mena[12], Joseph P. Flanagan[13], Greta Keller[14], Thomas Sanger[14], Bruce Ostrow[15], Maksim V. Plikus[5], Evgeny Z. Kvon[5], Erez Lieberman Aiden[2,3✉] & Ricardo Mallarino[1✉]

Phenotypic variation among species is a product of evolutionary changes to developmental programs[1,2]. However, how these changes generate novel morphological traits remains largely unclear. Here we studied the genomic and developmental basis of the mammalian gliding membrane, or patagium—an adaptive trait that has repeatedly evolved in different lineages, including in closely related marsupial species. Through comparative genomic analysis of 15 marsupial genomes, both from gliding and non-gliding species, we find that the *Emx2* locus experienced lineage-specific patterns of accelerated *cis*-regulatory evolution in gliding species. By combining epigenomics, transcriptomics and in-pouch marsupial transgenics, we show that *Emx2* is a critical upstream regulator of patagium development. Moreover, we identify different *cis*-regulatory elements that may be responsible for driving increased *Emx2* expression levels in gliding species. Lastly, using mouse functional experiments, we find evidence that *Emx2* expression patterns in gliders may have been modified from a pre-existing program found in all mammals. Together, our results suggest that patagia repeatedly originated through a process of convergent genomic evolution, whereby regulation of *Emx2* was altered by distinct *cis*-regulatory elements in independently evolved species. Thus, different regulatory elements targeting the same key developmental gene may constitute an effective strategy by which natural selection has harnessed regulatory evolution in marsupial genomes to generate phenotypic novelty.

The developmental program of an organism comprises a set of tightly regulated processes that operate during ontogenesis to specify embryonic axes, control spatiotemporal gene expression patterns and, ultimately, establish tissue phenotypes[3]. As such, alterations to different genetic components of developmental programs may lead to the evolution of new traits[1–4]. In many instances, similar traits have repeatedly evolved in different species experiencing common ecological pressures[5–7]. This phenomenon—termed convergent evolution—has been of long-standing interest in evolutionary and developmental biology because it can reveal the extent to which developmental programs are inherently flexible or constrained to follow predictable paths[8–10]. Dissecting the molecular mechanisms responsible for evolutionary convergence can therefore provide key insights into the relationship between phenotypic novelty and the underlying developmental causes.

Patagia are specialized skin membranes that enable animals to glide, an effective mode of movement for escaping predators and accessing different food sources (Fig. 1a). Probably reflecting the adaptive advantage they provide, patagia have arisen convergently among diverse mammals, including different species of rodents, bats, colugo and Australian marsupials[11,12]. Notably, patagia have evolved independently in three closely related marsupial species, all members of the superfamily Petauroidea[12,13]. This phylogenetic arrangement constitutes an ideal scenario for comparative studies at the genomic level. Moreover, marsupial gliders are a uniquely tractable group for experimental manipulation because their patagia arise after birth, while the neonates are in their mother's pouch.

We previously developed a model in the petauroid gliding marsupial species *Petaurus breviceps* (the sugar glider) to study patagium

[1]Department of Molecular Biology, Princeton University, Princeton, NJ, USA. [2]The Center for Genome Architecture, Department of Molecular and Human Genetics, Baylor College of Medicine, Houston, TX, USA. [3]The Center for Theoretical Biological Physics, Rice University, Houston, TX, USA. [4]School of BioSciences, The University of Melbourne, Parkville, Victoria, Australia. [5]Department of Developmental and Cell Biology, University of California, Irvine, Irvine, CA, USA. [6]Department of Mathematics, University of California, Irvine, Irvine, CA, USA. [7]NSF-Simons Center for Multiscale Cell Fate Research, University of California, Irvine, Irvine, CA, USA. [8]Lewis Sigler Center for Integrative Genomics, Princeton University, Princeton, NJ, USA. [9]The University of Western Australia, Crawley, Western Australia, Australia. [10]Cornell University College of Veterinary Medicine, Ithaca, NY, USA. [11]Center for Complex Biological Systems, University of California, Irvine, Irvine, CA, USA. [12]SeaWorld San Diego, San Diego, CA, USA. [13]Houston Zoo, Houston, TX, USA. [14]Department of Biology, Loyola University, Chicago, IL, USA. [15]Department of Biology, Grand Valley State University, Allendale, MI, USA. [16]Present address: Stowers Institute for Medical Research, Kansas City, MO, USA. [17]Present address: Department of Environment and Genetics, La Trobe University, Bundoora, Victoria, Australia. [18]These authors contributed equally: Jorge A. Moreno, Olga Dudchenko. ✉e-mail: erez@erez.com; rmallarino@princeton.edu

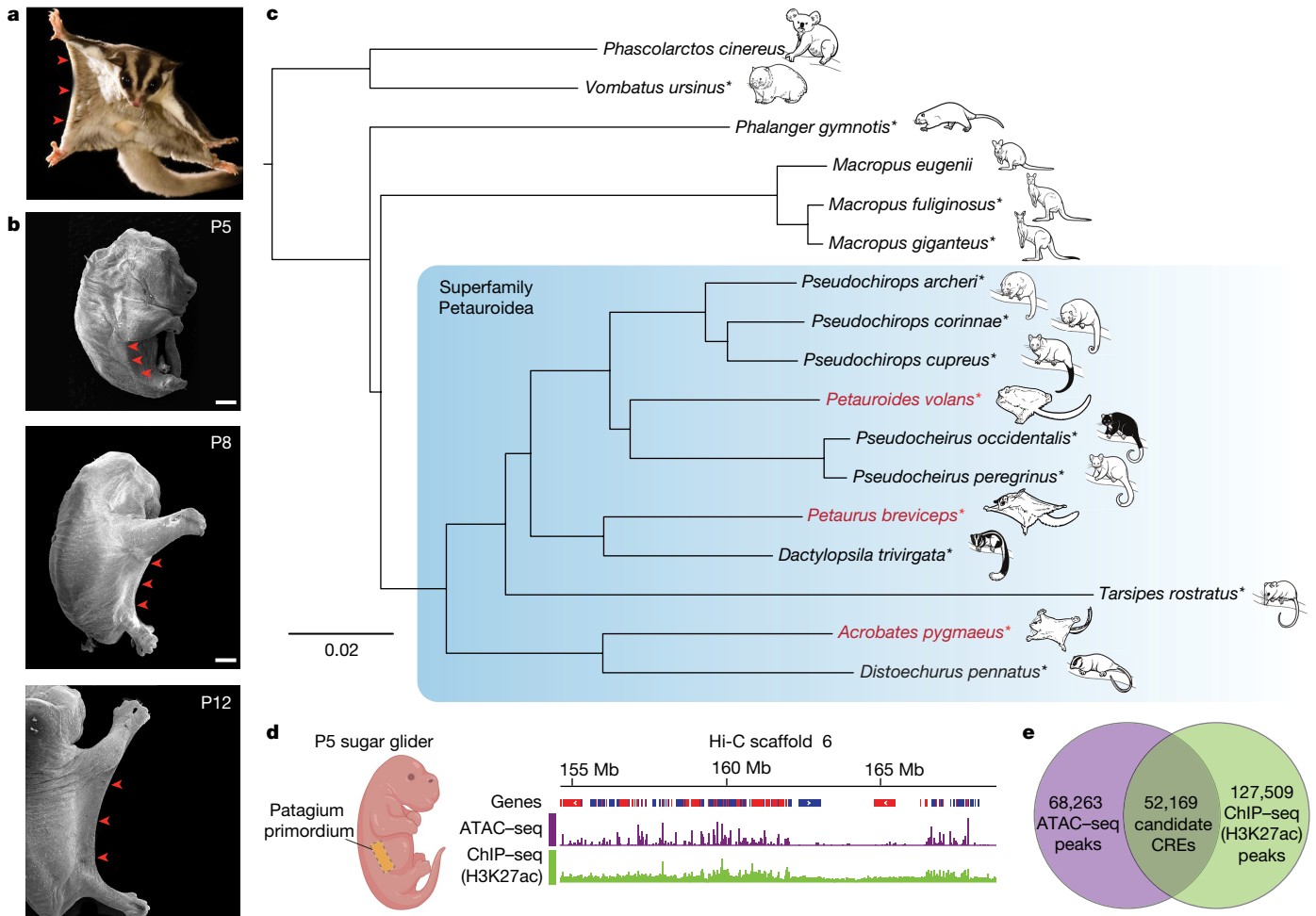

**Fig. 1 | Convergent evolution of patagia among closely related marsupial species. a**, An adult sugar glider extending its patagium (red arrowheads) during gliding flight. Photo credit: Joe MacDonald. **b**, Scanning electron micrograph showing dorsolateral views of P5, P8 and P12 sugar glider joeys. The patagium primordium (red arrowheads) becomes externally visible at P5 and continues to grow and extend in subsequent days. Scale bars, 1 mm. **c**, Species tree topology estimated from whole-genome data. All displayed branches have 100% bootstrap support. Phylogeny is consistent with the independent evolution of patagia in three petauroid species (labelled in red font): *P. breviceps*, *P. volans* and *A. pygmaeus*. Species for which we generated genome sequences and assemblies are indicated by asterisks. **d**, ATAC–seq and ChIP–seq traces from the P5 sugar glider patagium primordium. **e**, The experimental strategy used to identify the set of candidate *cis*-regulatory elements used for downstream analyses. The joey schematic in **d** was created using BioRender.

formation[14]. In sugar gliders, the patagium primordium differentiates from the interlimb lateral skin shortly after birth and grows outward during subsequent days, extending distally from limbs and body wall (Fig. 1b). Reflecting its distinctive developmental program, the patagium transcriptome differs substantially from that of neighbouring skin regions, including many differentially expressed genes (DEGs) with known roles in patterning and growth processes[14]. Although the precise molecular events controlling patagium development remain largely unclear, it is probable that such transcriptional differences are driven by evolutionary changes in *cis*-regulatory elements that are active in the lateral skin. Here, using a combination of comparative genomics and functional approaches, we set out to dissect the *cis*-regulatory architecture of patagium formation and uncover the developmental mechanisms underlying the repeated evolution of this morphological adaptation.

## Identifying GARs

To identify the *cis*-regulatory regions underlying patagium evolution, we used comparative genomics across gliding and non-gliding marsupials. As described below, we devised an experimental and computational strategy to uncover glider accelerated regions (GARs), which we define as candidate *cis*-regulatory elements that are highly conserved across species but show a disproportionate rate of nucleotide substitutions in glider lineages. As lineage-specific rapid changes in *cis*-regulatory elements have been associated with evolutionary shift of function[15–17], we reasoned that identifying GARs could provide insights into the genomic loci involved in patagium formation.

We first obtained tissues and sequenced the genomes of 14 marsupials, including 2 gliding and 7 non-gliding petauroids, as well as 5 outgroup species (Fig. 1c and Supplementary Table 1). With the exception of two non-gliding possums (*Dactylopsila trivirgata* and *Tarsipes rostratus*), for which tissue quality restricted us to short-read sequencing data, we used Hi-C and short-read sequencing[18] to obtain genome assemblies for all other species. Using this same strategy, we also generated an improved sugar glider genome assembly. Analysis using Benchmarking Universal Single-Copy Orthologs (BUSCO)[19] showed that our resulting genome assemblies varied in recovery of complete single-copy orthologues, ranging from 72% to 94% (Extended Data Fig. 1). These 14 new genome assemblies, together with our improved

sugar glider assembly (15 total), considerably increase the number of marsupial genomes available and provide a valuable resource for evolutionary genomic studies.

Having obtained a nearly complete genus-level sequencing of petauroids, we next sought to identify a high-confidence set of lateral skin candidate *cis*-regulatory elements that we could then test for increased rates of nucleotide substitution across the gliding species in this group. To this end, we performed assay for transposase-accessible chromatin (ATAC–seq) and chromatin immunoprecipitation followed by sequencing (ChIP–seq) of acetylated lysine on histone 3 (H3k27Ac) analyses of the patagium primordium of postnatal day 5 (P5) sugar gliders (Fig. 1d). While ATAC–seq identifies genome-wide open chromatin regions[20], H3k27Ac marks provide information on which of those regions correspond to active enhancers and promoters[21]. Thus, coupling DNA accessibility and H3k27Ac marks is a useful strategy to identify active *cis*-regulatory elements[22]. After independent processing and analysis, we intersected our ATAC and ChIP (H3k27ac) datasets and generated a final list of 52,169 candidate *cis*-regulatory elements, comprising ATAC peaks overlapping with ChIP (H3k27ac) marks (Fig. 1e).

Starting from our set of candidate *cis*-regulatory elements, we developed a pipeline to extract orthologous regions across 17 marsupial genomes, including the 15 produced in this study and 2 others that were publicly available and used as outgroups[23,24] (Fig. 1c and Supplementary Tables 1 and 2). After aligning and filtering each set of orthologous regions, we used phyloP[25] to identify signatures of evolutionary acceleration. As phyloP performs comparisons on a per-species basis, this method is useful for identifying sequences undergoing accelerated rates of substitution in each glider species. We identified 4,414 elements accelerated in at least one glider (that is, GARs). Among GARs, 1,635 were found in the branch leading to *P. breviceps*, 1,801 in *Acrobates pygmaeus* and 978 in *Petauroides volans*. There were 112 GARs that were shared among different pairs of gliding species, although none that were shared by all three (Extended Data Fig. 2). Taken together, these results show that the three gliding marsupials have a limited set of overlapping GARs and imply that each species has experienced lineage-specific patterns of regulatory evolution.

## Convergence of GARs around the *Emx2* locus

Across mammals, enhancers exhibit a high turnover rate and evolve more rapidly compared with other functional regions of the genome[26]. We hypothesized that, if selection is not acting on orthologous *cis*-regulatory elements, it may be operating on different elements controlling the same key developmental genes. To test this, we sought to establish whether there were genes that showed a statistically higher than expected number of neighbouring GARs. We first performed Micro-C[27], a variant of Hi-C that uses micrococcal nuclease digestion, using tissue from the patagium primordium of P5 sugar gliders. This approach enabled us to map contact domains or contiguous genomic intervals in which the loci exhibit enhanced contact frequency with one another (sometimes called topologically associating domains, or TADs). The data also allowed us to define potential interactions between *cis*-regulatory elements and genes. For example, *cis*-regulatory elements and genes grouped under the same contact domain were considered as possibly interacting whereas those found on separate contact domains were not. On the basis of these criteria, we assigned candidate *cis*-regulatory elements to corresponding genes and applied a hypergeometric test to determine whether there were genes statistically enriched for GARs (Methods). For each gene, we computed the probability of observing a given number of GARs based on a hypergeometric distribution, where the number of trials was defined as the total number of candidate *cis*-regulatory elements assigned to a gene and the probability of success as the proportion of GARs relative to the total number of candidate *cis*-regulatory elements. This analysis yielded

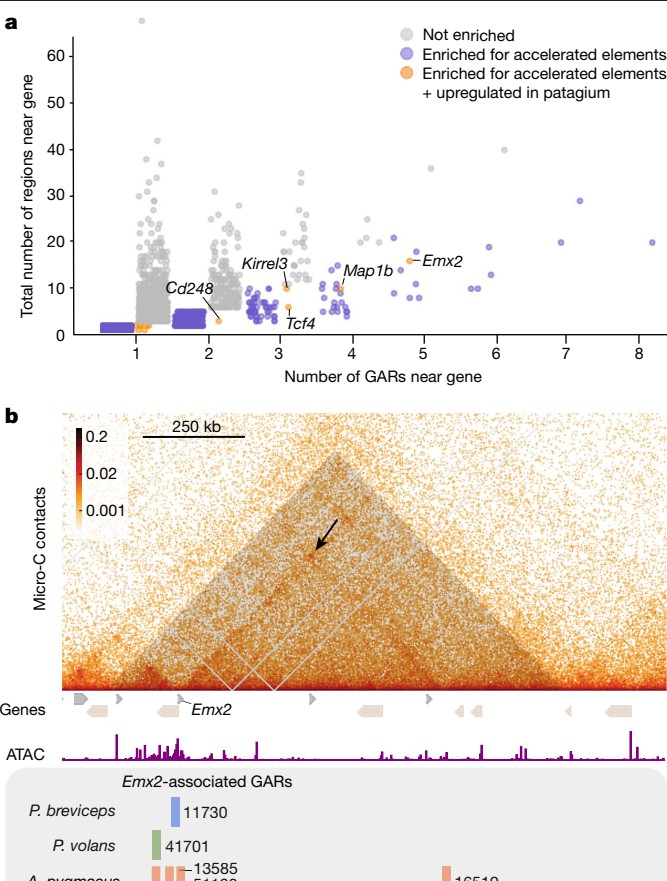

**Fig. 2 | *Emx2* is enriched for GARs. a**, Enrichment analysis identified genes with an overrepresentation of GARs. Genes containing at least one GAR were plotted. **b**, Contact domain containing the *Emx2* locus. *Emx2*-associated GARs for *P. breviceps* (blue), *P. volans* (green) and *A. pygmaeus* (light red), and Micro-C contact loops are shown. The contact loop between GAR 16519 and the *Emx2* promoter is shown in light red (the black arrow shows the contact point); other called loops are shown in grey. There were no other distant GARs displaying contact interactions with *Emx2*.

a total of 1,116 genes enriched for GARs (Fig. 2a and Supplementary Data 1). To further narrow our list of candidates, we reanalysed RNA-seq data from pouch young spanning P2–5 stages[14] and examined whether any of the 1,116 candidate genes were upregulated in the patagium primordium (false-discovery rate (FDR) < 0.1) compared with other skin regions, as this would suggest a functional relationship (Extended Data Fig. 3 and Supplementary Data 2). Of the 23 candidates that fulfilled this criterion, there was only one—*Emx2*—that had at least one associated GAR in each of the three glider species (3 GARs in *A. pygmaeus*, 1 GAR in *P. volans* and 1 GAR in *P. breviceps*) (Fig. 2a,b and Supplementary Table 3). By contrast, the remaining genes had GARs either in only one glider species (17 genes) or in two (5 genes) (Supplementary Table 3). Furthermore, reanalysis of our Micro-C chromatin conformation data, which can uncover long-range interactions not detected by our enrichment test, identified a clear chromatin contact loop between the *Emx2* promoter and an *A. pygmaeus* GAR located 1 Mb downstream, increasing the total number of *Emx2*-associated GARs to 6 (4 GARs in *A. pygmaeus*, 1 GAR in *P. volans* and 1 GAR in *P. breviceps*) (Fig. 2b).

Owing to their accelerated patterns of evolution, GARs may be important for driving changes in gene expression. To test whether

the different *Emx2*-associated GARs had regulatory activity, we performed luciferase assays in an immortalized dermal fibroblast cell line generated from sugar glider trunk skin. We synthesized all six *Emx2*-associated GAR sequences identified in the different glider species, as well as the orthologous regions from the respective non-gliding sister species. We next cloned these sequences into a luciferase reporter vector and conducted transfections into sugar glider dermal fibroblasts. Quantification of fluorescence after 48 h revealed that several of the GARs drove significantly higher levels of luciferase activity compared with the empty control vector. Specifically, the *P. volans* GAR (GAR 41701) showed almost double the enhancer reporter activity compared with the empty control vector, while activity from the orthologous sequence from *Pseudocheirus peregrinus*, the non-glider sister species, was indistinguishable from the control (Extended Data Fig. 4; $n = 6$). The same pattern was observed for the long-distance GAR of *A. pygmaeus* (GAR 16519), with this sequence driving significantly higher enhancer reporter activity compared with the control, while the sequence from its non-glider sister species, *Distoechurus pennatus*, did not differ from the control (Extended Data Fig. 4; $n = 6$). In contrast to these results, none of the other *A. pygmaeus* GARs (GAR 51182, GAR 32020 or GAR 13585) or the corresponding orthologous sequences from *D. pennatus* showed significant differences in luciferase activity compared to the empty controls (Extended Data Fig. 4; $n = 6$). In the case of *P. breviceps*, the *Emx2*-associated GAR (GAR 11730), located in the *Emx2* promoter (200 bp upstream of the *Emx2* transcription start site (TSS)), showed significantly higher promoter-reporter activity compared with the control vector (Extended Data Fig. 4). Notably, the orthologous sequence of *D. trivirgata*, the non-glider sister species of *P. breviceps*, also exhibited similarly strong promoter-reporter activity compared with the control vector (Extended Data Fig. 4; $n = 6$). We performed computational scans[28] of the *P. breviceps* GAR and the *D. trivirgata* orthologous sequence and found no detectable differences in the presence/absence of predicted transcription binding sites between both species. Together, the findings from our in vitro luciferase assays demonstrate that three of the *Emx2*-associated GARs, one from each of the three glider species, drive elevated levels of transcriptional activity.

We investigated whether the functionally active *Emx2*-associated GARs (GAR 41701, GAR 16519, GAR 11730) exhibited shared transcription-factor-binding motifs, which might suggest glider-specific regulatory interactions. Through computational analysis[28], we identified an enrichment of four motifs within the GARs compared to orthologous sequences from non-glider species. Among the transcription factors predicted to bind to these motifs, 13 were found to be upregulated in the sugar glider patagium (*Ascl2*, *Egr1*, *Klf10*, *Klf12*, *Maz*, *Nr2f6*, *Patz1*, *Prdm1*, *Rara*, *Sp2*, *Sp3*, *Vezf1* and *Znf740*). While none of these transcription factors are known upstream regulators of *Emx2*, it is plausible that novel regulatory interactions have evolved between some of these genes and the regulatory regions of *Emx2* in gliding species, potentially influencing the expression of this gene.

Taken together, our selection and enrichment analysis, coupled to our long-range interaction data, identify *Emx2* as a locus containing an unusually high number of associated GARs (that is, 6), with at least one GAR present in each of the three gliding species. Moreover, our transcriptomic analyses showing *Emx2* expression during patagium formation, paired with the results of our in vitro luciferase assays, indicate that selection on *Emx2*-associated GARs across marsupial gliders is likely to have had functional consequences during patagium evolution.

## *Emx2* downregulation through in-pouch transgenesis

*Emx2* is an attractive candidate for controlling patagium formation because it encodes a homeodomain transcription factor with known roles in patterning and fate specification of mesenchymal tissues[29–33].

Furthermore, *Emx2* is not only significantly upregulated during the early stages of patagium development, as described above (Extended Data Fig. 3 and Supplementary Data 2), but it also continues to be expressed at elevated levels in the patagium for at least 14 days postnatally, as indicated by additional transcriptional comparisons between different skin regions (FDR < 0.1) (Extended Data Fig. 5a and Supplementary Data 3). Notably, immunohistochemistry (IHC) analysis of transverse sections from pouch young revealed that EMX2 was strongly and specifically localized to the dermis of the patagium primordium, both before and during patagium outgrowth (Extended Data Fig. 5b).

To establish a direct link between *Emx2* and patagium phenotypes, we next sought out to test the function of this gene in vivo. Like other marsupials, sugar glider joeys complete much of their physical development ex utero, inside the maternal pouch, which facilitates the development of experimental methods. We first attempted to stably transduce the skin of sugar glider joeys by delivering lentiviral particles through intradermal injection. We temporarily exposed a P3 joey by everting the pouch of an anaesthetized female and intradermally injected its lateral skin with a lentivirus carrying a GFP reporter[34], before returning the joey to the pouch interior (Fig. 3a). At P10, 7 days after the injection, the lentivirus had stably transduced a large proportion of dermal cells in the developing patagium, as evidenced by widespread GFP expression (Fig. 3b).

Having successfully established proof-of-concept in-pouch stable transgenesis, we next used short hairpin RNA (shRNA) to downregulate *Emx2* in the developing sugar glider patagium. Using our sugar glider immortalized fibroblasts, we tested five different shRNA constructs for their ability to cause a reduction in *Emx2* expression. The top shRNA from our in vitro analysis, shEmx2-3, as well as the scramble control sequence (shScram), were selected for downstream in vivo injections. We intradermally injected either the shEmx2-3 ($n = 4$) or the shScram control ($n = 4$) virus into the lateral skin of P3 joeys, targeting only one side of each individual. This strategy enabled us to perform comparisons between injected and uninjected patagia within the same joey, while controlling for body size. Seven days after the injection, measurements performed on cryosections revealed that the area of patagia injected with shEmx2-3 virus was significantly smaller relative to the uninjected side (Fig. 3c,d). By contrast, patagia injected with the shScram control virus did not show a comparable size difference relative to the uninjected tissue (Fig. 3c,d). Using quantitative PCR (qPCR) and IHC analysis, we confirmed that patagia transduced with shEmx2-3 had a reduction in *Emx2* compared with the uninjected side (Fig. 3e–g; $n = 5$). Thus, the results from our in vivo functional experiments indicate that mesenchymal expression of *Emx2* is required for patagium patterning/outgrowth.

## *Emx2* is an upstream patagium regulator

To study the molecular effects of *Emx2* downregulation in the developing patagium, we next performed RNA-seq analysis of patagia transduced with shEmx2-3 ($n = 4$) and shScram control ($n = 4$) virus. Differential gene expression analysis identified 420 genes downregulated in patagia transduced with shEmx2-3 (Fig. 3h and Supplementary Data 4; FDR < 0.1). Notably, 59 out of these 420 genes overlapped with genes that are upregulated in the patagium primordium of wild-type sugar gliders, relative to neighbouring skin[14] (Fig. 3i and Supplementary Table 4). This number is significantly higher than what is expected by chance ($P = 0.000126$, hypergeometric test), indicating that *Emx2* is an upstream regulator of the patagium's transcriptional program. Included in the list of 59 genes were various transcription factors with known roles in mesenchymal patterning, such as *Pax1*, *Barx2*, *Hoxb9* and *Lmx1b*. Moreover, this gene set also contained *Wnt5a*, encoding a WNT ligand that we previously implicated in controlling characteristic phenotypes of the patagium primordium, including increased cell density, epidermal hyperplasia and increased cell proliferation[14].

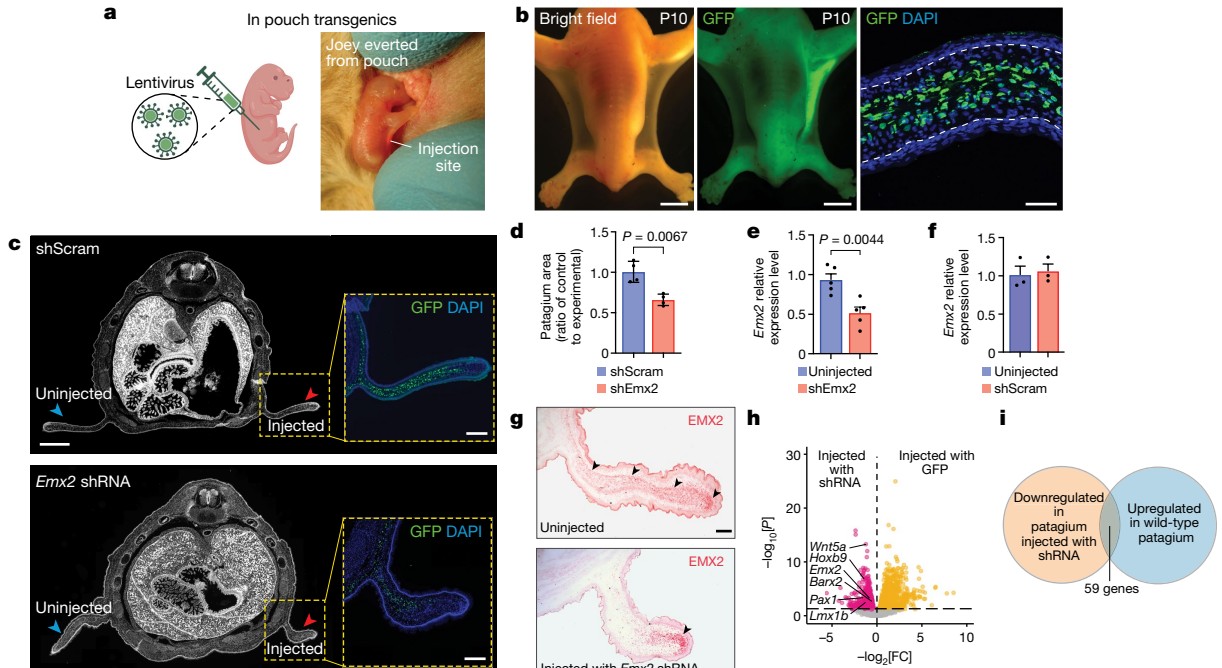

**Fig. 3 | In-pouch transgenesis to probe *Emx2* function. a**, The strategy used to deliver lentiviral particles. **b**, A lentivirus carrying a GFP reporter stably transduced the developing patagium, as seen in dorsal images of P10 joeys and transverse cryosections. **c**, Transverse cryosections of joeys injected with either an shRNA lentivirus targeting *Emx2* (shEmx2-3) or a control lentivirus (shScram). **d**, The ratio between the area of uninjected and injected patagia. Data are mean ± s.e.m. **e,f**, qPCR analysis of relative *Emx2* expression in patagia transduced with shRNA against *Emx2* (**e**) or shScram (**f**) and non-transduced patagia. Data are mean ± s.e.m. **g**, IHC analysis of EMX2 (arrowheads) distribution. **h**, DEGs (data corrected for multiple comparisons; FDR < 0.1)

between patagia injected with the experimental and control virus. Genes downregulated in patagia injected with the shEmx2-3 virus are shown in pink; genes more highly expressed in patagia injected with the shScram control are shown in yellow; genes without differential expression are shown in grey. FC, fold change. **i**, The number of overlapping genes downregulated in patagia injected with shEmx2-3 and upregulated in the native patagium primordium. Statistical significance in **d** (*n* = 4; *P* = 0.0067), **e** (*n* = 5; *P* = 0.0044) and **f** (*n* = 3; *P* = 0.7585) was assessed using two-tailed *t*-tests. Scale bars, 500 μm (**b** (left and middle), **c** (main image)) and 50 μm (**b** (right)), 100 μm (**c** (inset) and **g**). The schematic in **a** was created using BioRender.

These results are also consistent with previous findings showing that *Emx2* positively regulates *Wnt5a* and other WNT signalling factors during forebrain development[35]. Indeed, Gene Ontology and KEGG pathway enrichment analysis of genes downregulated in patagia injected with the shEmx2-3 virus identified WNT signalling as one of the most significant terms (Extended Data Fig. 6), suggesting that *Emx2* may be exerting its effect in the patagium through modulation of this pathway.

To uncover *Emx2*'s mechanism of action and identify its direct downstream targets, we performed ChIP–seq analysis in the patagium primordium of wild-type P5 sugar gliders using an anti-EMX2 antibody. We identified 30,676 genome-wide EMX2-bound sites, and de novo motif discovery[36] recovered the canonical motif for EMX2, confirming the specificity of our antibody. We next sought to establish whether any of the 59 genes from the analysis described above contained EMX2-bound sites that overlapped with ATAC/ChIP (H3k27ac) peaks, as this would indicate direct regulation by EMX2. Using computational methods[37], we scanned the contact domains containing each of the 59 different loci, and found that 38 out of the 59 genes fulfilled these criteria, including *Wnt5a* (Supplementary Table 4). This number was significantly higher than what is expected by chance (*P* < 0.0001, hypergeometric test). Among these genes, eight (*Adamts15*, *Cacna2d3*, *Cadm1*, *Cdc42ep4*, *Dok6*, *Emx2*, *Hmcn1* and *Plekhg1*) had assigned GARs that overlapped with EMX2-bound sites. Although we did not find evidence of previously established protein–protein interactions between EMX2 and any of these genes, as determined by STRING analysis[38], and did not identify any enriched pathways through Gene Ontology analysis, it is possible that some of these genes have undergone glider-specific functional

rewirings to regulate different processes in patagium formation, such as muscle development (*Adamts15*, metalloprotease), cell migration (*Cdc42ep4*, pseudopodia formation) and cell division (*Hmcn1*, cleavage furrow maturation).

Taken together, the results from our transcriptomic analyses and ChIP–seq experiment indicate that *Emx2* is an upstream regulator of patagium development, directly binding to the regulatory sequence of key signalling molecules/transcription factors expressed in the patagium primordium, and suggest this gene may be exerting its effects by modulating WNT signalling.

## *Emx2* directly regulates *Wnt5a*

As our in vivo functional experiments and EMX2 ChIP–seq assays suggested a direct connection between *Emx2* and *Wnt5a*, two genes that are critical for patagium formation, we next investigated the nature of their interaction in more detail. First, using RNA in situ hybridization analysis of *Wnt5a* coupled with IHC analysis of EMX2, we confirmed that both genes were co-expressed in the same cells of the developing sugar glider patagium (Fig. 4a). We next inspected the contact domain containing *Wnt5a* and found that, out of the 20 candidate *cis*-regulatory elements assigned to *Wnt5a*, 8 of them overlapped with EMX2-bound sites (Fig. 4b). Among these 8 elements, 7 were located away from the *Wnt5a*-coding sequence, at distances ranging from 389 kb upstream to 283 kb downstream of the *Wnt5a* TSS (Fig. 4b). The remaining element was situated near the *Wnt5a*-coding sequence, immediately upstream of a TSS (Fig. 4b). Notably, the majority of EMX2 binding sites found in these elements were present in most marsupial species

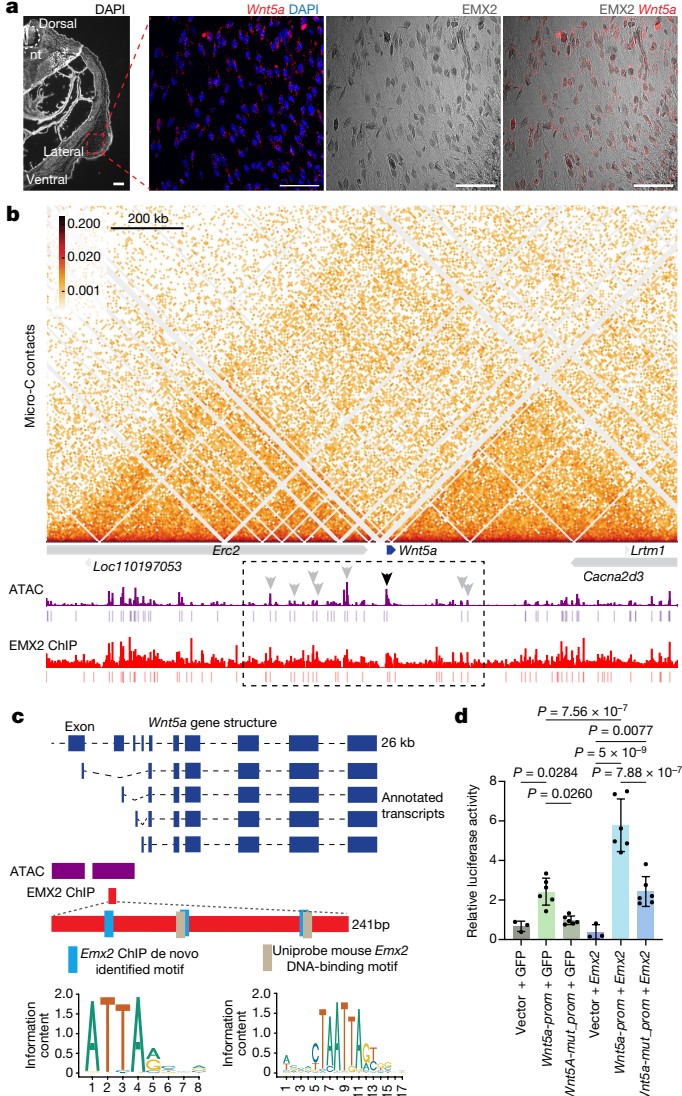

**Fig. 4 | *Emx2* directly regulates *Wnt5a*. a**, In situ hybridization analysis of *Wnt5a* coupled with IHC analysis of EMX2 in a cross-section of the developing patagium of a P5 sugar glider joey. *Wnt5a* can be visualized as individual puncta while EMX2 staining shows nuclear localization. Scale bars, 500 μm (low magnification) and 50 μm (high magnification). nt, neural tube. **b**, The contact domain containing the *Wnt5a* locus. The Micro-C contacts, ATAC peaks (purple) and EMX2-bound sites (red) are shown. The dotted box delineates all of the candidate regulatory elements assigned to *Wnt5a*. The arrowheads within that region show overlapping ATAC and EMX2-bound sites; the grey arrowheads show peaks that are distant from the *Wnt5a* promoter and the black arrowhead shows the peak located near to the *Wnt5a* coding sequence that was chosen for characterization. **c**, Schematic of sugar glider *Wnt5a* showing annotated transcripts recovered from RNA-seq data. The 241 bp region overlapping between the ATAC peak (purple box) and an EMX2-bound site (red box) was chosen for downstream analysis. Shown at the bottom of the panel are the different EMX2 binding motifs contained in this 241 bp sequence. **d**, The relative luciferase activity of different constructs tested. Data are mean ± s.e.m. Statistical significance was assessed using one-way analysis of variance (ANOVA). $n = 6$ (experimental constructs) and $n = 3$ (controls). $P = 7.56 \times 10^{-7}$ (*Wnt5a-prom* + GFP versus *Wnt5a-prom* + *Emx2*), $P = 7.88 \times 10^{-7}$ (*Wnt5a-prom* + *Emx2* versus *Wnt5a-mut_prom* + *Emx2*), $P = 5 \times 10^{-9}$ (*Wnt5a-prom* + *Emx2* versus vector + *Emx2*), $P = 0.0260$ (*Wnt5a-prom* + GFP versus *Wnt5a-mut_prom* + GFP), $P = 0.0284$ (*Wnt5a-prom* + GFP versus vector + GFP), $P = 0.9980$ (*Wnt5a-mut_prom* + GFP versus vector + GFP), $P = 0.0077$ (*Wnt5a-mut_prom* + *Emx2* versus vector + *Emx2*).

in our alignment, suggesting that the interaction between *Emx2* and *Wnt5a* may be conserved (Supplementary Data 5). Inspection of our Micro-C data did not reveal contacts between any of the seven distant candidate *cis*-regulatory elements and the *Wnt5a* locus. Although this does not rule out that these distant sites mediate direct EMX2–*Wnt5a* interactions, we reasoned that the element near the *Wnt5a* coding sequence represented the strongest candidate for further analysis. Within this peak, there was a 241 bp region, marked by overlapping ATAC and H3k27ac ChIP peaks, containing three EMX2 binding sites (Fig. 4c). To determine whether the 241 bp fragment contributed to mediating a direct interaction between EMX2 and *Wnt5a*, we cloned it into a luciferase promoter reporter vector (*Wnt5a-prom*) and performed a series of luciferase assays in sugar glider dermal fibroblasts. First, we co-transfected the *Wnt5a-prom* and a GFP control expression construct, then compared luciferase production with that of cells that were co-transfected with an empty vector and the GFP control. We found that *Wnt5a-prom* drove a significantly higher production of luciferase, confirming that the 241 bp sequence had promoter activity (Fig. 4d). Next, to establish whether promoter activity was dependent on EMX2, we mutated the three EMX2-binding sites (producing *Wnt5a-mut_prom*) and examined whether this would alter luciferase production. Indeed, we found that *Wnt5a-mut_prom* yielded luciferase levels that were indistinguishable from the empty vector when co-transfected with the GFP control (Fig. 4d). Finally, to test the 241 bp sequence's response to EMX2, we co-transfected *Wnt5a-prom* or *Wnt5a-mut_prom* with an *Emx2* expression construct, observing significantly higher luciferase production in the former compared with all other combinations of promoter and expression construct (Fig. 4d). These results demonstrate that promoter activity of the 241 bp sequence is dependent on EMX2 binding. Taken together, our results show that the putative *Wnt5a* alternative promoter contains regulatory motifs that respond to EMX2 binding, further strengthening the evidence that *Emx2* regulates patagium development by directly binding to *Wnt5a*.

## A modified ancestral program

The regulatory mechanisms responsible for directing the spatial expression of *Emx2* to the interlimb lateral skin may be an evolutionary innovation of sugar gliders or, instead, represent a conserved feature of mammalian development. To distinguish between these alternatives, we examined the expression of *Emx2* in laboratory mouse embryos using whole-mount in situ hybridizations. At embryonic day 11.5 (E11.5), *Emx2* was expressed at the base of the limb buds as well as in the Wolffian ridge, a band of flank mesenchyme that adjoins the forelimb and hindlimb buds[39] (Fig. 5a and Extended Data Fig. 7a). At E13.5, *Emx2* expression was restricted to portions of the limbs and no longer visible in the Wolffian ridge (Fig. 5b). While *Emx2* expression at the base of the limbs is critical for regulating the formation of the pectoral and pelvic girdles[32,33], it is unclear whether the transient expression seen in the Wolffian ridge mesenchyme is consequential for mouse embryogenesis. Nonetheless, the close anatomical correspondence between the Wolffian ridge and the marsupial interlimb region from which the patagium primordium originates, suggests that the spatial expression of *Emx2* is partially conserved across mammals.

Although *Emx2* is expressed in the interlimb skin region of both sugar gliders and laboratory mice, its temporal expression pattern differs markedly among both species. In sugar gliders, *Emx2* is upregulated for at least the first 2 weeks of postnatal development, whereas, in laboratory mice, it is transiently expressed from E9.5 to E11.5 (refs. 32,33) and declines at subsequent embryonic/postnatal stages[40,41]. To study the effect of prolonging *Emx2* expression in the mouse dermis, thereby resembling the temporal expression pattern seen in sugar gliders, we used the *Rosa^{Emx2-GFP}* mouse strain[42], in which *Emx2-GFP* can be overexpressed in the presence of Cre recombinase. Overexpression

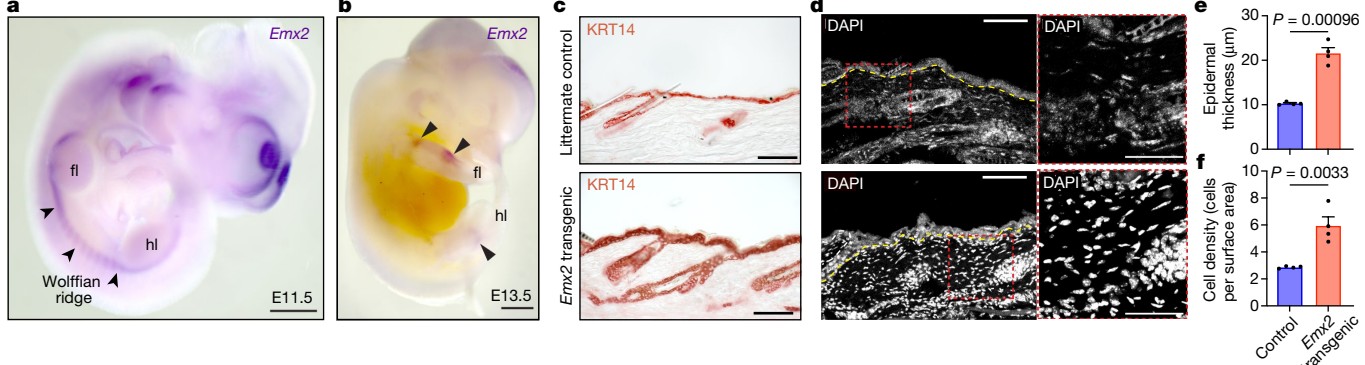

**Fig. 5 | Spatial expression and function of *Emx2* in mice. a,b**, Whole-mount in situ hybridization of *Emx2* in E11.5 (**a**) and E13.5 (**b**) laboratory mouse embryos. *Emx2* is transiently expressed in mesenchymal cells of the Wolffian ridge (arrowheads) of E11.5 embryos. At E13.5, staining is restricted to portions of the limbs (arrowheads) and is no longer visible in the Wolffian ridge. fl, forelimbs; hl, hindlimbs. **c–f**, Relative to control littermates, *Pdfgra^{creERT2/+}Rosa^{Emx2-GFP/+}* double transgenic mice overexpressing *Emx2* in dermal fibroblasts show a significant increase in epidermal thickness, as determined by KRT14 staining (**c**; quantification in **e**), and an increase in mesenchymal cell density, as determined by DAPI staining (**d**; quantification in **f**). For **e** and **f**, statistical significance ($n = 4$; $P = 0.00096$; **e**) and ($n = 4$; $P = 0.0033$; **f**) was assessed using a general mixed-effects model one-way ANOVA. The yellow dotted line in **d** denotes the dermis–epidermis boundary. For **e** and **f**, data are mean ± s.e.m. Scale bars, 500 μm (**a** and **b**) and 100 μm (**c** and **d** (left)) and 50 μm (**d** (right)).

of *Emx2* using a Cre driver under the control of the *Emx2* promoter led to marked overexpansion of the brain, resulting in embryonic lethality (at around E17.5). To circumvent this problem, we crossed *Rosa^{Emx2-GFP}* mice with *Pdfgra^{creERT2/+}* mice, a strain that can inducibly express Cre recombinase in dermal fibroblasts[43]. We induced *Emx2* upregulation in the skin of juvenile mice by injecting tamoxifen into the peritoneum of P30 mice and collected skin tissue for analysis 7 days later (P37) (Fig. 5c–f and Extended Data Fig. 7b–d). IHC analysis of skin sections confirmed that double transgenic animals (*Pdfgra^{creERT2/+}Rosa^{Emx2-GFP/+}*) expressed robust levels of EMX2 in the dermis, compared with the control littermates (Extended Data Fig. 7b). *Emx2* overexpression led to epidermal hyperplasia and expanded expression of the basal keratinocyte marker KRT14, as indicated by histological analysis and IHC (Fig. 5c,e and Extended Data Fig. 7c). This phenotype was accompanied by a significant increase in cell proliferation and mesenchymal cell density, as seen by incorporation of EdU and DAPI stain, respectively (Fig. 5d,f and Extended Data Fig. 7d). Notably, these are the same cellular phenotypes that we previously observed during our histological characterization of sugar glider patagium development[14]. Moreover, these are also the same phenotypes that we observed when we over-expressed *Wnt5a* in the mouse skin[14] (Extended Data Fig. 8), further showing that *Emx2* and *Wnt5a* act in coordination (Supplementary Note 1 and Supplementary Fig. 1).

Having established that the spatial expression and function of *Emx2* are partially conserved in laboratory mice, we next examined whether any of the three GARs that drove strong reporter activity in our in vitro assays (that is, GAR 41701, GAR 16519, GAR 11730) had the ability to recapitulate the sugar glider *Emx2* expression patterns in laboratory mice. To this end, we cloned each of these sequences into a LacZ reporter vector and used CRISPR–Cas9 to insert the resulting constructs into the *Igs2* (also known as H11) mouse locus. This locus serves as an intergenic safe harbour for genomic insertions, enabling efficient and reproducible tests of enhancer activity[44]. Screening at E9.5 and E11.5, stages at which *Emx2* is robustly expressed in different regions of laboratory mouse embryos (Fig. 5a), indicated that none of the GARs drove LacZ reporter expression, despite successful integration of the respective constructs into the desired locus (Extended Data Fig. 9a,b). To uncover a potential explanation for this result, we performed evolutionary conservation analyses using the marsupial genomes reported in this study, as well as publicly available genomes for laboratory mice and humans. We also examined the genome of the opossum (*Monodelphis domestica*), a non-petauroid marsupial from the New World. This latter comparison enabled us to assess conservation across marsupials, but outside of petauroids. Our results indicated that, while GARs showed a high conservation score among marsupials (Extended Data Fig. 9c), only the GAR located on the *Emx2* promoter was conserved in eutherians (Extended Data Fig. 9d). Specifically, even though syntenic relationships within the *Emx2* locus are largely similar among marsupials, laboratory mice and humans, we were unable to find high-confidence orthologous matches between most of our *Emx2*-associated GARs (that is, 5 out of 6) and laboratory mouse/human sequences (Extended Data Fig. 9d).

Taken together, our experiments show that, while the spatial expression and function of *Emx2* are partially conserved between laboratory mice and sugar gliders, its temporal expression pattern differs significantly. Notably, we find evidence that the *Emx2–Wnt5a* axis acts to pattern elements of the forebrain and the craniofacial skeleton during mouse embryogenesis (Supplementary Note 1). Thus, these results may imply that the sugar glider *Emx2* expression pattern evolved through modifications of a pre-existing, ancestral gene expression program that is present in all mammals. Finally, we found that none of the marsupial GARs drove LacZ activity in mouse embryos, a result that is probably explained by the inability of mouse *trans*-acting factors to bind to glider *cis*-regulatory elements and/or by inherent differences in the *trans*-acting environment between laboratory mice and marsupials (Supplementary Note 2 and Supplementary Fig. 2).

## Discussion

Understanding how changes in developmental programs lead to phenotypic novelty is a fundamental question in biology. Here, by analysing 15 new marsupial genome assemblies, we present evidence that three independent instances of gliding flight in marsupials have coincided with selection on different *cis*-regulatory elements near *Emx2*. Moreover, using functional genomic approaches and in-pouch transgenic experiments in sugar gliders, we show that *Emx2* is a key regulator of patagium development (Supplementary Note 3).

*Emx2* is a vertebrate homeobox gene homologous to empty spiracles (*ems*) in *Drosophila*. In flies, *ems* controls brain segmentation[45], whereas, in mammals, *Emx2* patterns the forebrain[31,46], the olfactory epithelium[30], the urogenital system[29], sensory organs of the inner ear[42] and components of the appendicular skeleton[32,33], among others. Notably,

loss of function of *ems*/*Emx2* leads to defects in tissues/structures that constitute outgrowths of the body wall, such as insect antennae[47] as well as mammalian urogenital primordia and limbs[29,32,33]. Our in-pouch lentiviral in vivo experiments in sugar gliders demonstrate that downregulation of *Emx2* leads to a quantifiable decrease in the size/area of the patagium, suggesting that the function of *ems*/*Emx2* in regulating different outgrowth structures may be conserved across metazoan evolution (Supplementary Note 4).

*Emx2* is expressed in the lateral skin of both laboratory mice and sugar gliders. In mice, *Emx2* expression is transient. Prolonged over-expression of *Emx2* in the mouse dermis did not result in obvious skin outgrowth, like what is observed during sugar glider patagium formation. This result could be partly explained by the fact that *Emx2* was overexpressed throughout the skin, rather than restricted to a specific anatomical region. Moreover, while our experiments provide evidence that *Emx2* is critical for patagium formation, the patagium is almost certainly a polygenic trait that is patterned by multiple genes and species-specific gene regulatory networks working in concert[14]. Nonetheless, our mouse experiments revealed marked increases in mesenchymal cell density, epidermal thickness and cell proliferation, all of which are characteristic cellular features of the sugar glider patagium primordium[14]. This suggests that at least some of the networks downstream of *Emx2* are conserved across mammals. Given the similarities in spatial expression between mouse and sugar gliders, we postulate that the evolution of marsupial patagia was facilitated by lineage-specific changes in the levels and duration of *Emx2* expression, a gene expressed in a conserved domain across mammals, rather than through the evolution of novel spatial domains. From this perspective, our future efforts will be directed towards understanding the upstream molecular regulators of *Emx2* in gliding species.

Convergent evolution of traits in species experiencing similar selective pressures provides a natural experiment for examining the extent to which developmental programs change in predictable ways[8,9]. In mammals, enhancers evolve more rapidly compared with other regions of the genome, but the rate at which different enhancers evolve varies considerably[26]. For traits that are present in most taxonomic groups and have critical roles in organismal survival, such as the skull or the limbs, enhancer conservation tends to be high because of strong constraints imposed by purifying selection. Thus, changes in such traits, including those leading to convergence, arise through alterations in a small fraction of enhancers tolerating evolutionary sequence modification[48]. As a result, selection is predicted to act on orthologous elements across different species. By contrast, a novel phenotypic trait like the patagium probably arises from evolutionary tinkering with much less critical tissue parameters (that is, lateral skin morphology). While common ecological pressures may promote morphological convergence among species, there may be a larger number of possible genetic changes tolerated by purifying selection, leading to less strict regulatory convergence. Thus, non-orthologous accelerated enhancers targeting the same key developmental gene may represent a way in which rapid enhancer evolution in mammals is harnessed by natural selection to generate phenotypic novelty.

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

## Methods

### The *P. breviceps* sugar glider breeding colony

All experiments performed were approved by the IACUC committee at Princeton University. Captive-born, adult sugar gliders were obtained from the US pet trade and thereafter maintained in a breeding colony at Princeton University. The colony is kept under a 12 h–12 h light–dark cycle (temperature, 20–27 °C; humidity, 30–70%). Animals are fed daily a diet consisting of dried food, fruits and protein. Animals are housed in breeding pairs or trios. Female sugar gliders are inspected daily for pouch young by gently palpating the maternal pouch. Pouch young discovered by palpation are visually examined by briefly anaesthetizing the mother with isoflurane and gently everting the pouch to expose the neonate. For tissue collection, joeys are gently detached from the nipple, euthanized and processed in the laboratory. Both male and female joeys were used in all experiments. More details on our breeding colony can be found elsewhere[14].

### Scanning electron microscopy analysis

Sugar glider joeys were fixed and stored in 2% glutaraldehyde at 4 °C. After making several small incisions in the abdomen to increase infiltration, we treated the embryos 1% osmium tetroxide for 90 min at room temperature followed by critical point drying. We sputter-coated all specimens with a 1-Å-thick coating of palladium and then imaged them on a SU3500 scanning electron microscope under a high vacuum.

### Sample acquisition, genome sequencing and genome assemblies

The following samples were obtained from the ABTC collection of the South Australian Museum, following all established protocols and guidelines from museum authorities: *A. pygmaeus* (ABTC 77168, liver); *D. trivirgata* (ABTC 49304, kidney); *D. pennatus* (ABTC 44094, liver); *P. volans* (ABTC 83627, kidney); *P. peregrinus* (ABTC 138055, muscle); *Pseudocheirus occidentalis* (ABTC 7808, kidney); *Pseudochirops cupreus* (ABTC 45036, kidney); *Pseudochirops corinnae* (ABTC 49246, kidney); *T. rostratus* (ABTC 7742, kidney). A kidney sample from a female sugar glider, *P. breviceps*, was obtained from the Princeton University breeding colony. The samples were processed for genome sequencing and assembly using a combination of Illumina short-read sequencing and Hi-C for scaffolding. The full pipeline for sequencing and assembly used here has been described in detail previously[18,49]. Using this approach, we were able to generate genome assemblies for 13 out of the 15 species. For *D. trivirgata* and *T. rostratus*, we were not able to generate a Hi-C assembly owing to poor sample quality. Thus, after generating short-read sequencing data for these two species, we performed contig assembly using MEGAHIT (v.1.1.4.2)[50] and used Redundans (v.0.14a)[51] for haplotype purging, preliminary scaffolding, and gap closing. Lastly, we scaffolded against the Hi-C sugar glider genome with ragtag v.2.0.1 (parameters: -g 50 -m 10000000 -f 200)[52]. This approach is particularly suitable in our case, given the short phylogenetic divergence among petauroids. The representation of mammalian universal single-copy orthologues in the different genomes was assessed with BUSCO (v.5.4.4)[27] using the curated mammalian v.10 database[19]. The assembly metrics for all genome assemblies in this study are reported in Supplementary Table 1.

### ATAC–seq analysis

ATAC–seq was performed on live single-cell suspensions from the patagium primordium, which were prepared as follows: tissue was dissected from P5 sugar glider joeys, placed into 1× PBS, transferred into 0.2% dispase/RPMI and incubated at 37 °C for 1 h. After this incubation, the epidermis and dermis were separated using forceps. The dermis was washed once in 1× PBS, transferred to 0.2% collagenase/RPMI and incubated at 37 °C for 2 h. The resulting dermal cell suspension was passed through a 40 µm mesh filter, and the remaining enzyme was washed by adding 30 ml of 0.2% bovine serum albumin (BSA)/1× PBS and centrifuging the cells for 10 min at 500 rcf (4 °C). The supernatant was discarded, and the pellet was resuspended in 0.2% BSA/1× PBS and filtered again into a 1.5 ml Eppendorf tube. After determining cell viability using Trypan Blue (Sigma-Aldrich), a total of 100,000 live cells per sample (*n* = 2) was set aside and used for library preparation.

Library preparation was performed according to the Omni-ATAC method[53]. In brief, cells were lysed for 3 min on ice. The transposition reaction on the permeabilized nuclei was performed using TDE1 transposase (Illumina) at 37 °C for 60 min, and then purified using the Zymo DNA Clean and Concentrator-5 kit (Zymo). Illumina sequencing adapters and barcodes were added to the transposed DNA fragments by PCR amplification. The purified ATAC–seq library products were examined on Agilent Bioanalyzer DNA High Sensitivity chips for size distribution, quantified using the Qubit fluorometer (Invitrogen) and pooled at equal molar amounts. The ATAC–seq library pool was sequenced on the Illumina NovaSeq 6000S Prime flowcell as paired-end 61 nucleotide reads. We generated 62,387,513 and 59,728,230 reads for libraries 1 and 2, respectively. Raw sequencing reads were filtered by the NovaSeq control software and only the pass-filter reads were used for further analysis. Raw ATAC–seq reads were trimmed using NGmerge v.0.2_dev and mapped to the Hi-C *P. breviceps* assembly using Bowtie2 (v.2.4.2)[54]. Alignments were further processed by removing duplicates using picard MarkDuplicates SNAPSHOT v.2.21.4 (http://broadinstitute.github.io/picard), and samtools (v.1.12)[55] was used to filter reads and convert files into BAM format. Peak calling on each biological replicate (*n* = 2) was conducted using MACS2 (v.2.2.7.1)[56] (parameters: --nomodel -q 0.05 --keep-dup all --shift -100 --extsize 200 -g 2456432000 --nolambda). To assess the concordance of peak calls between our replicates, we used the Irreproducible Discovery Rate IDR (v.2.0.4.2)[57] and only those peaks passing a false-discovery rate threshold of 0.05 were used. We then used BEDTools intersect (v.2.27.1)[58] to remove regions of the ATAC peak calls that directly overlapped annotated exons.

### ChIP–seq analysis

ChIP assays were performed using flash-frozen samples from the patagium primordium of P5 sugar glider joeys. For each reaction, patagium primordia from 6–7 joeys were pooled into a single tube. Approximately 20 mg of tissue was cross-linked with 1% formaldehyde and chromatin was extracted and sheared. The following antibodies were used in the ChIP assays: anti-H3K27ac antibody (Abcam, ab4729, GR232896-1, 4 µg), anti-EMX2 antibody (Novus, NBP2-39052, 27711) and a negative control anti-IgG antibody (Millipore, 12-370, 297424, 4 µg). For H3K27Ac ChIP–seq, chromatin was divided into two experimental samples (each experimental sample consisting of pooled tissue from 6–7 joeys) and one control sample, and ChIP assays were performed in triplicate for each of the experimental samples and in duplicate for the negative control sample. In all cases, we used 7 µg of chromatin and 4 µg of antibody. ChIP DNA was then processed into three standard Illumina ChIP–seq libraries (two experimental and one control) and sequenced to generate the following number of paired end reads: 56,614,872 (control); 56,957,310 (experimental 1); and 70,392,940 (experimental 2). EMX2 ChIP–seq analysis was performed in a similar manner, except that only one experimental (pooled tissue from 6–7 joeys) and one control library were generated and sequenced to generate the following number of single-end reads: 43,909,990 (control) and 41,672,698 (experimental). Raw ChIP–seq reads were trimmed using NGmerge v.0.2_dev and mapped to the Hi-C *P. breviceps* assembly using Bowtie2 (v.2.4.2)[54]. Alignments were further processed by removing duplicates using picard MarkDuplicates SNAPSHOT v.2.21.4, and samtools (v.1.12)[55] was used to filter reads and convert files into BAM format. Broad peaks were processed and broad peaks on each biological replicate (*n* = 2) were called with MACS2 (v.2.2.7.1)[56] (parameters: --broad -f BAMPE -g 2456432000 -q 0.05 --nolambda), using the input as a background control. Broad peak calls could not be filtered using IDR and, instead,

peaks between the two replicates were concatenated and overlapping peaks were merged using BEDTools merge (v.2.27.1)[58]. To identify enriched motifs in our EMX2 ChIP–seq data, we scanned all called ChIP peaks using the Simple Enrichment Analysis (SEA) program[59] from the MEME suite (v.5.5.4)[60].

## Computational analyses

**Identifying GARs.** To identify GARs, we used a statistical phylogenetic test for acceleration along a specific branch of a phylogeny, as implemented in PhyloP[25]. PhyloP requires two inputs: (1) a species tree, typically estimated from genome-wide data; and (2) a multiple-sequence alignment for each genomic region to be tested for acceleration. To obtain a species tree, we aligned orthologous coding sequences from the seventeen species included in our analysis (Supplementary Tables 1 and 2) and extracted fourfold degenerate sites. We then used RAxML (v.8.2.12)[61] (parameters: -f a -x 50217 -p 50217 -# 1000 -o Pcine,Vursi -m GTRGAMMA) and Phylofit (RPHAST suite v.1.6.9)[62] to produce a guide species tree and mod file, respectively (Fig. 1c). In parallel, we generated a tree using the first and second codon positions and found that it was identical to that produced by fourfold degenerate sites (Supplementary Fig. 3). To obtain a set of candidate sequences (that is, candidate cis-regulatory elements) that we could then test for acceleration, we focused on overlapping peaks between our ATAC–seq and our ChIP–seq sugar glider datasets. Peaks were considered overlapping if they had at least 1 bp in common between both datasets. Using this approach, we identified 52,169 candidate cis-regulatory elements. The size distribution of the candidate cis-regulatory elements is shown in Supplementary Fig. 4.

To identify orthologues of the 52,169 sugar glider candidate cis-regulatory elements across the 15 other diprotodont genomes examined in our study, we used a comparative annotation approach. First, we annotated conserved coding genes in each species by lifting-over gene model from the high-quality RefSeq annotation of the koala genome to each other genome using LiftOff v.1.6.3 (parameters: -d 4). We then conducted a second lift-over of sugar glider candidate cis-regulatory elements to each other species using the same procedure but with the addition of a flanking sequence to improve candidate cis-regulatory element mappability and reduce the chances of multi-mapping (parameter: -flank 1). We next used synteny anchoring[63] to validate candidate orthologues of sugar glider candidate cis-regulatory elements. For each of the 15 non-sugar-glider marsupial genomes, we created a list of candidate cis-regulatory element orthologues and their flanking genes (excluding genes that were not annotated in the reference sugar glider genome). Then, for each candidate cis-regulatory element orthologue in each species, we compared the identities of their flanking genes to those in the sugar glider genome. We considered elements to be orthologues of their reference sugar glider candidate cis-regulatory element candidate if (1) the first flanking gene, upstream or downstream, matched that in the sugar glider genome and (2) if those flanking genes in the target species were no greater than four times the distance from the candidate cis-regulatory element measured in the sugar glider genome. Candidate cis-regulatory element orthologues that passed this synteny check were then extracted from their respective genomes using gffread (v.0.12.7)[64].

Candidate cis-regulatory element orthologues across all species were then combined into a multi-fasta file and aligned using MAFFT v.7.453 (parameters: --adjustdirectionaccurately --localpair --maxiterate 1000). As our downstream evolutionary analyses were based on nucleotide substitution rates, alignment filtering was designed to address two key considerations: first, that estimated substitution rates in very gappy alignment regions containing a large number of gaps may not be reliable; and, second, that filtering based on similarity risks biasing the results of analyses that are predicated on measuring sequence divergence. We therefore used a filtering approach similar to that implemented by a previous study[65] and the

Avian Phylogenomics Project[66]. First, we trimmed the flank region from each non-sugar-glider target species by removing alignment columns outside the bounds of the sugar glider reference candidate cis-regulatory element sequence. Next, gappy alignment ends were trimmed using a 20 bp sliding window until 75% of species used in our analyses had a window with less than or equal to 5 gaps (Ns). To mitigate internal gap columns, we first used TrimAI v.1.4.rev15 (parameter: --gappyout) and then removed individual orthologue sequences that, after our gap filtering, still contained greater than 25% gaps or retained any individual gaps greater than 10% of the alignment length. Finally, we retained only those alignments that included sequences from at least five species, including at least one glider and non-gliding sister species and least one species from each outgroup. GAR alignments are shown in Supplementary Data 6. We next used phyloP[25] from the RPHAST suite (v.1.6.9)[62] (method="LRT", mode="ACC") to identify sequences exhibiting accelerated substitution rates, testing each of the three gliding species (*P. breviceps*, *A. pygmaeus* and *P. volans*) independently. PhyloP performs comparisons on a per-species basis. That is, it compares a given genomic region (ATAC/ChIP peaks in our case) to the average genome substitution rate in the corresponding species. Thus, each species will have its own set of accelerated regions. Once we had a set of accelerated elements for each species, the resulting lists were compared to each other to establish which of those showed overlap among the glider species.

The superfamily Petauroidea is part of the Diprodontia, the largest extant order of marsupials that also includes the superfamily Phalangeroidea and the suborders Vombatiformes and Macropodiformes. Although Vombatiformes are an accepted outgroup, the phylogenetic relationships among the other three groups remain unclear[13,67,68]. In agreement with a previous study[13], our analysis using whole-genome data, both from fourfold degenerate sites and first and second codon positions, supports the placing of Petauroidea + Macropodiformes as sister groups (Fig. 1c and Supplementary Fig. 3 (topology 1)). However, other studies have placed Petauroidea + Phalangeroidea (topology 2)[68] or Petauroidea + Phalangeroidea (topology 3)[67] as sister groups. To verify that topology had no effect on our results, we conducted additional phyloP analyses forcing topologies 2 and 3. Our analysis recovered around 97–99% of the elements in each glider species, including all the ones around *Emx2*. Specifically, topologies 2 and 3 recovered, respectively, 99.4% and 97.4% of elements for *A. pygmaeus*; 99.6% and 97.1% of elements for *P. breviceps*; and 99.8% and 97.9% of elements for *P. volans*. Thus, while resolving the phylogenetic relationships among Diprotodonts falls outside the scope of our study, our analyses show that our results are not affected by differences in topology.

**Gene enrichment analysis.** To test for enrichment of accelerated sequences near genes differentially expressed in the patagium, we first associated candidate cis-regulatory elements with their putative target genes using contact domains determined from our Micro-C data. First, contact domains called with windows sizes of 10, 25 and 50 kb were integrated using an approach described previously[69]. In brief, contact domain calls were combined and filtered such that fully overlapping and fully nested contact domains across all call sets were merged, whereas single and partially overlapping contact domains were retained. For all genes, we grouped all transcripts, sorted them by length and selected the largest transcript. In other words, we used the longest transcript as a representative for each gene. The TSS for each gene was determined based on the location of the first annotated exon. If this exon did not start with an ATG, it was considered the 5′ untranslated region, and the TSS was annotated as the site 1 bp directly upstream of the first exon. For exons that did begin with an ATG codon, the TSS was estimated to be approximately 1 kb upstream from the translation start site. For each remaining TAD, its overlap with annotated TSSs and a candidate cis-regulatory element was calculated using BEDTools (v.2.27.1)[58],

using the intersect function with parameter -wo. We then assigned each candidate *cis*-regulatory element to the nearest gene TSS using BEDTools closest v.2.27.1, using the default settings, and created a table that included information on whether the sequences assigned to each gene were GARs. For this analysis, we merged the list of GARs from each of the three gliding species, and all of the gene–enhancer annotations were conducted based on sugar glider genome coordinates. Out of 24,495 genes annotated in the sugar glider genome, we tested 11,044. The remaining genes were either not assigned to contact domains or had no candidate *cis*-regulatory elements assigned to them. For each gene we computed the one-tailed hypergeometric *P* value of observing a given number of GARs from the total number of candidate *cis*-regulatory elements near a gene, where the number of trials is the total number of GARs and candidate *cis*-regulatory elements in our dataset. This analysis yielded a total of 1116 genes enriched for GARs.

Our contact domain-based gene enrichment analysis represents a biologically meaningful approach because it relies on three-dimensional genomic interactions occurring when the patagium primordium is developing. However, previous studies have also used a distance-based strategy, in which genes are assigned to candidate *cis*-regulatory elements based on physical proximity along the chromosome[65]. To compare the outcomes of using a contact domain-based and a distance-based method, we reanalysed our data using a distance-based approach. As described above, we computed the probability of observing a given number of accelerated candidate *cis*-regulatory elements (GARs) based on a binomial distribution, where the number of trials is the total number of candidate *cis*-regulatory elements assigned to a gene and the probability of success is the proportion of all accelerated candidate *cis*-regulatory elements (GARs) over the total number of candidate *cis*-regulatory elements. Among the 24,495 genes annotated in the sugar glider genome, we tested 15,178 (the remaining genes did not have any candidate *cis*-regulatory elements assigned to them) and found that 1,638 were enriched for GARs. Of the 1,638 genes enriched, 27 were upregulated in the developing patagium compared with both the dorsal and shoulder skin. Of these 27, 21 were recovered with our contact domain-based analysis, demonstrating that the two approaches yield comparable results.

Finally, to establish the extent to which our results were being biased by contact domain size, we examined the size distribution of our contact domains alongside the corresponding number of enhancers. Our analysis revealed a low correlation between contact domain size and the number of enhancers ($R^2 = 0.05$) (Supplementary Fig. 5a). The majority of contact domains had 25 or fewer enhancers and were between 100,000 and 1,000,000 bp in size (Supplementary Fig. 5b,c). Notably, the contact domains containing our genes of interest are no larger than other contact domains and do not contain the greatest number of enhancers. Moreover, the largest contact domain (8,000,000 bp) did not contain the highest number of enhancers, and the contact domain containing the highest number of enhancers (~4,000,000 bp and 449 enhancers) does not contain any of our 59 genes of interest. Together, this analysis supports the notion that longer contact domains do not contain more active enhancers.

**Conservation between petauroid marsupials.** To examine whether *Emx2*-associated GARs are evolutionarily conserved elements, we analysed all marsupial genomes shown in Fig. 1c and generated conservation scores for genomic regions surrounding these elements. In brief, ~20 kb genomic regions surrounding *Emx2*-associated GAR orthologues were extracted from each species. These regions were then aligned with MAFFT v.7.453 (parameters: --genafpair, --ep 0 and --maxiterate 1000). Per-base conservation scores were then calculated using phyloP (parameters: --method LRT and --wig-scores). Conservation scores were then visualized against the reference sugar glider sequence in IGV as a heat map.

**Conservation between marsupials, laboratory mice and humans.** We assessed the conservation of *Emx2*-associated GARs in eutherian mammals using the laboratory mouse (mm10) and human (hg38) genomes. We used the UCSC genome browser blat tool[70,71] to identify orthologous regions between human, mouse and marsupial GARs. The *P. breviceps* ATAC peaks sequences were used as queries. For 5 out of the 6 GARs (that is, GAR 41701, GAR 16519, GAR 51182, GAR 32020 or GAR 13585), there were no hits that had either a strong score or that were located on the same chromosome as *Emx2*, even though the syntenic relationships among eutherians and marsupials in the region surrounding *Emx2* were conserved. For GAR 11730, located in the *Emx2* promoter, we identified a clear eutherian orthologue. We then used blat to search for these same *P. breviceps* sequences in the genome of the grey short-tailed possum (*M. domestica*), a non-petauroid marsupial from the New World. We were able to identify high-confidence hits for all six GARs. We then used the multiz alignment conservation scores in human and mouse and found that, other that the promoter element, the rest of the GARs had either no or very sparse conservation. We used both NCBI BLASTn and discontinuous mega-blast[72] to identify orthologous sequences for our six GARs. We performed a search in both mouse and human databases using the default parameters. Subsequently, we selected the top hit from each species located on the same chromosome as *Emx2*. We then conducted a reciprocal best BLAST against our *P. breviceps* database. Our findings mirrored the results obtained through our blat analysis, as we could only confidently recover an orthologous element for GAR 11730.

**Gene Ontology and KEGG pathway analysis.** Genes were examined for enrichment of KEGG Pathway and GO Biological Process terms using the Enrichr web server[73]. Protein–protein predicted associations were assessed using the STRING web server[38].

**Transcription-factor-binding analysis.** Transcription-factor-binding motif analysis was performed using the MEME suite (v.5.5.4)[60]. The *P. breviceps* GAR 11730 sequence was tested for enrichment of motifs using the *D. trivirgata* sequence used as a control. We also conducted the inverse to search for potential *P. breviceps* loss of motifs. Transcription factor motif scans across the orthologous GARs were performed using XSTREME[28]. The sequence of each species was run using the default parameters. For indel analysis, aligned sequences were scanned by eye to identify *P. breviceps*-specific insertions and deletions. These regions, plus 5–10 bp flanking sequences, were then run through XSTREME, altering parameters to account for the length of the sequences used (-nmotifs 10 -minw 4 -maxw 30). Tomtom[74] was used to identify genes associated with the identified motifs.

To determine whether candidate genes were potentially regulated by *Emx2*, we scanned the contact domains containing our candidate genes and used Bedtools intersect v.2.27.1 to determine whether there was overlap between the ATAC peaks associated to each gene and the *Emx2* ChIP–seq peaks.

For transcription factor motif conservation analysis, we first used FIMO[37] (MEME suite v.5.5.4) to identify sites matching the de novo identified sugar glider EMX2-binding motif, originally identified from the ChIP–seq experiment (ATTARCNV), with a *P* value of 0.01 or less. We then created a multiple-sequence alignment of the ATAC peaks from all of our species and looked for these identified sites. If they were present in at least half of the species, we considered the site conserved. If a site was not conserved in at least half of the species, we then assessed whether it displayed glider-specific conservation.

**scRNA-seq analysis of laboratory mouse data.** An existing scRNA-seq dataset[75] from dorsal skin of mouse embryos at E12.5, E13.5, E14.5 and E15.5 was reprocessed using the Seurat package (v.4.3.0)[76]. Libraries from each timepoint were filtered to contain only cells with >200 and <4,000 expressed genes and <5% mitochondrial gene counts. After

the filtering steps, Seurat Objects generated for reads from each time-point were merged, gene counts were normalized and variable features were identified for each timepoint. The objects were then integrated by selection of integration features and anchors ($n = 30$ dimensions) and the integrated object was scaled. Significant principal components ($n = 30$) were identified and used to generate a uniform manifold approximation and projection (UMAP) for dimensional reduction. User specified dimensions were used to define neighbours and clusters. Cell type annotation was performed as previously described[75].

Dermal fibroblasts identified at all timepoints in the integrated object were subset based on established markers (*Lum*, *Pdgfra*, *Crabp1* and *Lox*). The integrated object was split and reclustered, after normalization of the gene counts and reselection of variable features, integration features and integration anchors. Significant principal components, UMAP generation and neighbour/cluster identification were performed as described above. After subclustering, the expression of *Emx2* and 13 patagium upregulated transcription factors was examined in each cluster using violin plots. For transcription factors showing expression in the subcluster marked by *Emx2* (cluster 2), co-expression with *Emx2* was explored by plotting normalized counts of each gene with a blended matrix of normalized expression of the two genes.

## Micro-C analysis

Micro-C libraries were prepared using the Dovetail Micro-C Kit, according to the protocol suggested by the manufacturer (Dovetail Genomics). In brief, single-cell suspensions of the patagium primordium of P5 sugar glider joeys were generated as described for ATAC−seq and frozen at −80 °C. A total of ~1,000,000 pooled primary cells, corresponding to approximately 4–5 joeys, was used to generate each library. After thawing the cells, chromatin was fixed using disuccinimidyl glutarate and formaldehyde. Cross-linked chromatin was then digested in situ with micrococcal nuclease (MNase). After digestion, cells were lysed with SDS and chromatin fragments were bound to chromatin capture beads. Chromatin ends were then repaired and ligated to a biotinylated bridge adapter followed by proximity ligation of adapter-containing ends. After proximity ligation, cross-links were reversed, associated proteins were degraded, DNA was purified and a sequencing library was generated using Illumina-compatible adapters. In total, three libraries were generated. Before PCR amplification, biotin-containing fragments were isolated using Streptavidin beads and the libraries were sequenced on the Illumina HiSeq X platform to generate 660,352,634 ($2 \times 150$ bp) reads.

Raw Micro-C reads were analysed according to the Dovetail documentation (Dovetail Genomics). Reads from the three libraries were pooled and mapped to the Hi-C *P. breviceps* assembly using BWA (v.0.7.15-r1188)[77]. Subsequently, pairtools v.0.3.0 was used to remove PCR duplicates, sort and record valid ligation events. Moreover, pairtools select v.0.3.0 was used to determine unique pairs (with the '(pair_type=="UU") or (pair_type=="UR") or (pair_type=="RU") or (pair_type=="uu") or (pair_type=="Uu") or (pair_type=="uU")' option) and samtools (v.1.12)[55] was used to convert files into BAM format. The Hi-C matrix was produced using juicer tools (v.1.22.01)[78] and contact domains (at 10 kb, 25 kb and 50 kb) were called using juicer tools arrowhead (v.1.22.01) (parameters: -k KR --ignore-sparsity). We used Mustache (v.1.0.1)[79] to conduct loop calling (at 5 kb and 10 kb) on individual scaffolds (parameters: -d 250000 -st 0.7 -pt 0.05) and Higlass-manage v0.8.0[80] to visualize the data. We used Cooler (v.0.8.5)[81] to prepare files and generate cooler multi-resolution contact maps. Clodius (v.0.3.5) and Higlass-manage (v.0.8.0)[80] were used to format and ingest other files.

## Generation of immortalized sugar glider fibroblasts

The method for generating immortalized dermal fibroblasts has been described in detail previously[75,82]. In brief, a skin biopsy was obtained from the trunk region (encompassing dorsal and lateral skin) of a P10

sugar glider joey and fat/connective tissue was scraped away. The sample was digested overnight at 4 °C in a solution containing HBSS without $Ca^{2+}$ and $Mg^{2+}$ (Thermo Fisher Scientific), dispase (500 caseinolytic units per ml (Corning)), and an antibiotic/antimycotic solution (100 µg ml$^{-1}$ streptomycin, 100 IU ml$^{-1}$ penicillin and 250 ng ml$^{-1}$ amphotericin B (HyClone)). After removing and discarding the epidermis, the dermis was cut into small pieces and fibroblasts were expanded as described previously[75]. To generate an immortalized dermal fibroblast cell line, cells were transduced with undiluted γ-retroviral pseudoparticles according to procedures described previously[75]. Cells were verified as negative for mycoplasma by testing with the MycoAlert Mycoplasma Detection Assay kit (Lonza) according to the manufacturer's instructions. We profiled our cell line using RNA-seq and found that it displays robust levels of *Emx2* and other key genes that we previously identified as being expressed in the patagium tissue (such as *Wnt5a*, *Tbx3*, *Tbx5*, *Hand3* and *Osr1*)[14]. Moreover, we did not detect expression of genes like *Shh* or *Pax5*, a result that is also consistent with our previous transcriptional analysis of patagium tissue[14] (Supplementary Fig. 6 and Supplementary Data 7). These results suggest that our sugar glider cell line provides an adequate in vitro model to test hypotheses about the upstream and downstream regulation of *Emx2*.

## Luciferase assays

**GAR analysis.** For each candidate GAR, we synthesized the sequence from the glider species in which the sequence was found to be accelerated as well as from the corresponding non-gliding sister species (Twist Biosciences). The size of the sequence was defined by the overlap between the ATAC and the H3K27Ac ChIP peak identified in the sugar glider tissue. GARs and corresponding orthologues were cloned either into the pGL4.23 Luciferase Enhancer Reporter Vector (Promega) (GARs 41701, 16519, 32020, 51182, 13585) or the pGL4.10 Luciferase Promoter Reporter Vector (Promega) (GAR 11730) using In-fusion cloning (Takara). All of the resulting constructs were verified by Sanger sequencing. Immortalized sugar glider cells were seeded at a density of 5,000 cells per well and, the next morning, were transfected with the experimental constructs by using Lipofectamine (Invitrogen) (300 nl Lipofectamine to 200 ng plasmid DNA per well). To control for transfection, a control *Renilla* reporter vector (Promega 4.73) was co-transfected into each well (20 ng). After 48 h, cells were collected and processed using the DualGlo Luciferase Assay System (Promega) according to the protocol guidelines, and luciferase production was measured using a SpectraMax L luminometer (Molecular Devices). Luciferase activity was normalized relative to *Renilla* activity.

***Wnt5a*–*Emx2* interaction analysis.** We synthesized a 241 bp fragment corresponding to the region immediately downstream of the sugar glider *Wnt5a* first exon and a second one, in which the three *Emx2* binding sites in this sequence were replaced by G bases (that is, ATTA to GGGG) (Twist Biosciences). We used In-fusion cloning (Takara) to clone each of the fragments into the pGL4.10 Luciferase Promoter Reporter Vector (Promega), in front of the luciferase coding sequence. Co-transfection experiments were carried out with either a GFP expression plasmid (Addgene 11153) or an *Emx2* expression plasmid (Origene, NM_010132). All of the resulting constructs were verified by Sanger sequencing. For co-transfection experiments, we used 300 nl Lipofectamine to 200 ng plasmid DNA per well (100 ng of Luciferase vector and 100 ng of co-transfection GFP/*Emx2*). Cell seeding and all downstream experiments and analyses were performed as described above.

## Immunostaining

For sugar glider and mouse immunofluorescence and IHC analysis, tissue samples were collected and fixed in 4% paraformaldehyde (PFA) overnight, washed in 1× PBS and incubated in 30% sucrose at 4 °C overnight. The samples were then embedded in optimal cutting

temperature (OCT), flash-frozen and cryosectioned (16 μm thickness) using the Leica CM3050S Cryostat. The slides were kept at −80 °C until use. Antibody stains were performed on tissue sections using standard procedures. In brief, slides were washed with 1× PBS with 0.1% Tween-20 (PBT) and blocked with 1× PBT/3% BSA for 1 h. Rabbit anti-EMX2 (Novus NBP2-39052; 1:50), chicken anti-GFP (Novus Biologicals, NB100-1614, 1:200) or rabbit anti-KRT14 (BioLegend, 905301, 1:1,000) primary antibodies were diluted in 1× PBT/3% BSA and slides were incubated at 4 °C overnight. The next morning, the slides were washed several times with 1× PBT and incubated with secondary antibodies (Alexa-Fluor 488 (Thermo Fisher Scientific, ab150169, dilution 1:500); goat anti-rabbit Biotinylated (Vector Labs, R.T.U. (BP-9100-50; ready to use dilution). The reactions were visualized with HRP–streptavidin and the AEC substrate kit (Vector Labs: 568SK4200) or Alexa-dye-conjugated secondary antibodies (Thermo Fisher Scientific). The slides were washed several times with 1× PBT and mounted for imaging. AEC images were taken on a Nikon NiE upright microscope and fluorescence images on the Nikon A1R confocal microscope. NIS Elements v5 (Nikon) was used to acquire microscopy images.

## Statistics and reproducibility
Micrographs shown in Fig. 3b,c,g, 4a and 5a–c,e and Extended Data Figs. 5b, 7a,b and 8 constitute representative data from at least three biological samples.

## In vitro shRNA experiments
The RNAi Consortium (TRC) mouse lentiviral library carries several *Emx2* shRNA constructs[83]. We aligned the laboratory mouse and sugar glider *Emx2* coding sequence and designed five different sugar glider species shRNA constructs targeting the same regions that were previously used for targeting the laboratory mouse locus[83]. Moreover, we used the scrambled control recommended by the RNAi Consortium[83] (a list of all sequences is provided in Supplementary Table 5). We cloned the different sequences into the LV-GFP plasmid[34], which is designed for RNU6-1 promoter-driven shRNA expression. Large-scale production of VSV-G pseudotyped lentivirus was performed using calcium phosphate transfections in HEK293FT cells and the helper plasmids PMD2.G and psPAX2 (Addgene plasmids 12259 and 12260, respectively), as described previously[34]. For viral infections, we plated cells in six-well dishes at 300,000 cells per well, and incubated them overnight with unconcentrated lentiviruses in the presence of polybrene (100 μg ml⁻¹). The medium was replaced the next morning and cells were allowed to grow for 5 days. After this period, cells were sorted using FACS, grown for five additional days and sorted using fluorescence-activated cell sorting (FACS) a second time. Once cells reached confluency, RNA was extracted using the Zymo Directzol Kit (Zymo Research). The ability of the different shRNA constructs to induce *Emx2* downregulation was determined using qPCR, using the primers listed in Supplementary Table 5. When introduced into cultured sugar glider immortalized dermal fibroblasts, the different shRNAs reduced *Emx2* mRNA levels to 27% (shEmx2-1), 48% (shEmx2-2), 5% (shEmx2-3), 47% (shEmx2-4) and 38% (shEmx2-5), compared to a scrambled control shRNA (shScram). We selected shEmx2-2 and the shScram control for downstream analyses.

## In-pouch lentiviral transgenesis
To develop sugar glider transgenesis, we used the LV-GFP plasmid[34]. For subsequent shRNA experiments, we used shEmx2-3 (the most effective of the constructs that we tested) and the shScram control. Large-scale production of lentiviruses was performed as described for in vitro work, except that lentiviruses were subjected to ultraconcentration according to established protocols[34,84]. For injections, we anaesthetized female sugar gliders, gently exposed a P3 joey from inside the pouch and intradermally injected ~2.5 μl of concentrated virus into the interlimb region (one side only) using a 33-gauge needle. After injection,

the joey was gently placed back inside the pouch, and the mother was monitored until fully awake from the effect of anaesthesia. Several days later (P6 for RNA-seq/qPCR analysis and P9 for histology/measurements), females were anaesthetized, joeys were collected by gently detaching their jaw from the nipple, placed in a temporary container and brought back to the laboratory, where they were euthanized and processed for downstream analysis.

**Phenotypic measurements.** After confirming GFP expression by visually inspecting the tissue under a dissecting scope, joeys were fixed, embedded, cryosectioned and stained with DAPI. We used Fiji v.2.1.0 to measure the area of the left and right patagium in at least three non-consecutive sections per sample ($n = 4$ (shEmx2-3) and $n = 4$ (shScram) samples) and calculated the control/experimental ratio. Each measurement was performed three independent times and the results were averaged. All counts were performed in a blinded manner (that is, the researcher performing the measurements was unaware of which images corresponded to which genotypes/experimental conditions). Statistical differences were established using a mixed effects model one-way ANOVA test.

**Tissue collection for qPCR and RNA-seq analysis.** We injected the lateral skin of P3 joeys with either the shEmx2-3 or shScram lentivirus. Joeys were euthanized at P6 and infected patagium tissue, as visualized using GFP, was dissected and preserved in RNAlater. RNA was extracted using the RNeasy fibrous tissue mini kit (Qiagen) according to the manufacturer's protocol. For qPCR, we used $n = 5$ shEmx2-3 samples and $n = 5$ shScram samples. For RNA-seq, we used $n = 5$ (shEmx2-3) and $n = 5$ (shScram) samples.

## qPCR analysis
We used the qScript cDNA SuperMix (Quanta BioSciences) to generate complementary DNA (cDNA) and then performed qPCR using PerfeCTa SYBR Green FastMix (Quanta BioSciences). We assayed gene expression in triplicate for each sample and normalized the data using the housekeeping gene *Actb*. Primers used for qPCR were designed using the sugar glider genome and are reported in Supplementary Table 5. We analysed data from all qPCR experiments using the comparative $C_t$ method and established statistical significance of expression differences using two-tailed *t*-tests.

## Bulk RNA-seq
**Tissues.** RNA was extracted as described above and RNA-seq libraries were prepped using the TruSeq RNA Library Prep kit v2 (Illumina) and sequenced on the NovaSeq 6000 system (2 × 65 bp, paired-end). Pairwise differential expression analyses between the transcriptomes of shEmx2-3 or shScram skin were performed using DeSeq2 v.1.34.1 from BioConductor (https://bioconductor.org/)[85]. Only genes differentially expressed with an adjusted $P < 0.05$ were considered.

**Cells.** Immortalized *P. breviceps* cells were grown in a six-well dish until confluent. RNA was collected using the TRizol reagent according to the manufacturer's protocol. FASTQ reads were trimmed using Trimmomatic v.0.39 and aligned to the *P. breviceps* genome using STAR v.2.7.9a[86]. Counts were generated using featureCounts (v.2.0.1)[87] (featureCounts -p -t transcript -g transcript_id -O --minOverlap 10) and RPKM was calculated using the calculation reads for a gene/(all reads for the sample/1,000,000) × (1,000/length of gene).

## Whole-mount in situ hybridization analysis
The *Emx2* mouse riboprobe was a generous gift from T. Capellini and has been previously described[33]. Whole-mount in situ hybridizations was performed using previously described protocols[88]. In brief, mouse E11.5 and E13.5 (CD-1) embryos were post-fixed with 4% PFA in 1× PBS, washed with 1× PBS, treated with 20 μg ml⁻¹ proteinase in 1× PBT for

45 min and incubated overnight with the *Emx2* riboprobe at 65 °C. The next morning, the probe was washed with MABT (Maleic acid, NaCl, Tween-20) and incubated overnight with secondary anti-DIG antibodies (1:2,000) diluted in MABT, 2% Boehringer Blocking Reagent and 20% heat-treated sheep serum. After washing several times with MABT, the signal was developed by incubating with NBT/BCIP. Once a signal had developed sufficiently, the reaction was stopped by washing several times with PBT and fixing in 4% PFA overnight. The embryos were visualized using a SMZ18 stereo microscope (Nikon). NIS Elements v5 (Nikon) was used to acquire microscopy images.

### HCR in situ hybridization
The sugar glider *Wnt5a* coding region was used to generate 20 probe binding sequences for in situ hybridization chain reaction (HCR). HCR was performed using the standard protocol for fixed frozen tissue sections available from Molecular Instruments. For visualizing *Wnt5a* and EMX2 co-expression, sections hybridized with the *Wnt5a* probe were subsequently incubated with the EMX2 antibody, according to the procedure described for immunostainings.

### Stereoseq analysis
We analysed *Emx2* and *Wnt5a* co-expression in a previously generated spatial Stereo-seq dataset[89]. After examining all available datasets, we chose samples from E14.5 embryos, as this timepoint exhibited the most significant expression of both *Emx2* and *Wnt5a*. We analysed the E1S1 sample, as it contained both the highest mean expressions of *Emx2* and *Wnt5a*, as well as the highest percentage of spots with non-zero expression of *Emx2* and *Wnt5a*. All of the datasets were processed and analysed using Scanpy[2].

We analysed the brain and craniofacial regions separately. First, we extracted the $x$ coordinates and $y$ coordinates of the spatial spots from the entire tissue, which we denote as $X$ and $Y$, respectively. We reflected $Y \mapsto -Y$ so that the $y$ coordinates were positive. We defined the brain region as the collection of spots, $B$, that were originally annotated as brain and where $Y > 425$. The latter threshold was chosen to remove any spots that may have been mislabelled as the brain region due to cell type deconvolution in the original study.

From trial and error and visual inspection, we defined the craniofacial region as the collection of points, $C$, such that:

$$C = \{X, Y : X \leq 165 \text{ and } 425 < Y < 540.96 + 1.22X - 132.39 \text{ or } 165 < X < 275 \text{ or } 425 < Y < 462 - 1.23(X - 201.62)\}$$

To focus specifically on craniofacial gene expression, we further removed any points that were originally annotated as belonging to either the brain or meninges regions[51]. To determine the likely cellular identities of spots expressing *Emx2*, we performed spatially informed dimensionality reduction using SpaceFlow[90]. The latent embedding produced by SpaceFlow was used instead of principal component analysis to generate spatially coherent clusters, that is, clusters where spots are grouped based on both gene expression similarity and spatial proximity. We then generated a $k$-nearest neighbours ($k$-NN) graph ($k = 15$) to characterize spot–spot similarity. Using the $k$-NN graph, we used the Leiden algorithm to generate spatially resolved clusters that balanced spatial proximity with gene expression similarity, setting the clustering resolution parameter to Resolution=1.0. After constructing spatial clusters, we used differential expression analysis to determine the potential cellular identities of *Emx2*- and *Wnt5a*-expressing regions.

We used Mann–Whitney $U$-tests to identify spatial DEGs for each spatial cluster obtained by SpaceFlow. Specifically, we defined two particular groups of clusters. First, we aggregated the clusters in which *Emx2* and *Wnt5a* were co-expressed significantly into a '*Emx2* on/*Wnt5a* on' cluster. Second, we aggregated the clusters in which *Wnt5a* was expressed significantly but *Emx2* was absent into a '*Emx2*

off/*Wnt5a* on' cluster. We then performed two separate differential expression tests, one between the *Emx2* on/*Wnt5a* on cluster and the other SpaceFlow clusters and another between the *Emx2* off/*Wnt5a* on cluster and the other clusters to identify the DEGs specific to *Emx2*^high^*Wnt5a*^high^ regions, enabling us to determine the likely cellular identities of these regions.

### RNAscope
Frozen tissue sections (12 μm) of mouse embryos at E14.5 and E16.5 were used for RNA in situ hybridization using the RNAscope kit v2 (323100, Advanced Cell Diagnostics), according to the manufacturer's instructions. The *Mus musculus Wnt5a* and *Emx2* probes (316791 and 319001-C3, respectively; Advanced Cell Diagnostics) were used and labelled with OPAL 520 reagent (FP1487001KT, Akoya Biosciences) and TSA Vivid 650 dye (7536, Advanced Cell Diagnostics), respectively. For all of the experiments, slides were mounted using VECTASHIELD Anti-fade Mounting Medium with DAPI (H-1200-10; Vector Laboratories). Fluorescent images were captured with a Fluoview3000 laser-scanning microscope (Olympus) with UPLSAPO ×20/0.75 NA and ×40/1.25 NA Silicone objectives (Olympus).

### Mouse transgenics
All mouse experiments performed were approved by the IACUC committee at Princeton University. All mouse strains were provided with food and water ad libitum and kept on a 12 h–12 h light–dark cycle, at a temperature of 20 °C and 60% humidity. Experiments were carried out in both males and females.

***Emx2* overexpression.** The $Rosa^{Emx2\text{-}GFP}$ strain was a gift from D. Wu (NIH) and has been previously described[42]. The $Emx2^{cre}$ strain was obtained from Riken (RBRC02272) and has been previously described[46]. The $Pdgfra^{creERT2}$ strain was obtained from JAX (032770) and has been previously described[43]. The FVB/N769Tg(tetO-Wnt5a)17Rva/J and B6.Cg Gt(ROSA)26Sortm1(rtTA*M2)Jae/J strains were obtained from JAX (022938 and 006965, respectively) and have been previously described[91,92]. Both male and female mice were used in all experiments. For *Emx2* induction, P30 mice were intraperitoneally injected with tamoxifen (100ul; 20 mg ml$^{-1}$) during 5 consecutive days. Then 7 days after the last injection, mice were intraperitoneally injected with EdU (200ul; 10 nM) and euthanized 4 h later. For *Wnt5a* induction, P30 mice were placed on 1 mg ml$^{-1}$ doxycycline-containing water ad libitum for 7 days (doxycycline water was replaced every other day). Skin tissue was collected, fixed in 4% PFA at 4 °C overnight and embedded in OCT for cryosectioning.

**Phenotypic measurements.** Epidermal thickness was quantified using Fiji v.2.1.0 by measuring tissue stained with haematoxylin and eosin from the base of the epidermis to the top of the epidermis. Measurements were taken exclusively from interfollicular regions (10 measurements per dorsal skin section, 3 dorsal skins sections per sample, 4 samples per treatment). Cell density and cell proliferation were quantified on tissue sections by counting the number of DAPI$^+$ cells per surface area (measurements were done on three dorsal skin sections per sample, four samples per treatment). Statistical significance was assessed using a general mixed-effects model one-way ANOVA test (fixed effect = treatment; random effect = individual/sample).

**LacZ enhancer reporter assays.** We performed all mouse enhancer reporter assays using enSERT methodology, a recently developed strategy that uses CRISPR–Cas9 technology for site-directed insertion of enhancer reporter transgenes into the *Igs2* intergenic safe-harbour site[44,93]. As constructs are integrated into a specific genomic site, this method allows for reproducible and efficient testing of enhancer-reporter activity. After the constructs were synthesized, as described for our luciferase assays, we cloned them into a previously described

LacZ targeting vector (Addgene, 139098)[93] containing a minimal *Shh* promoter. The resulting vectors were injected into mouse zygotes (FVB/NCrl strain; Charles River) in pools of 4–5 constructs, together with *Igs2* gRNA and Cas9 protein (IDT), followed by embryo transfer into pseudopregnant females, as described previously[44]. Embryos were collected at E9.5 and E11.5. After collection, the embryos were screened for the corresponding GAR insertion using junction PCR and Sanger sequencing, and stained with X-gal according to previously established protocols[44]. At least 2 transgenic embryos with integration at the *Igs2* locus per construct were analysed.

## Reporting summary

Further information on research design is available in the Nature Portfolio Reporting Summary linked to this article.

## Data availability

All genome assemblies reported and corresponding genome sequencing reads are submitted under NCBI PRJNA512907. The ATAC–seq, ChIP–seq, Micro-C and RNA-seq reads are submitted under NCBI BioProject accession code PRJNA1078418. Source data are provided with this paper.

## Code availability

Code used for all analyses is available at FigShare (https://figshare.com/s/81cf39b7de363f1526a1)[94].

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

**Acknowledgements** We thank the members of the Mallarino laboratory for discussions and assistance with sugar glider husbandry; the members of the Australian Biological Tissue Collection at the South Australian Museum for providing tissues used to sequence marsupial genomes; staff at Princeton LAR (K. Gerhart, G. Barnett and J. McGuire) for help with sugar glider husbandry and care; J. M. Miller and J. Arley Volmar from the LSI Genomics Core (Princeton) for help with library preparation; the staff at the Nikon Center of Excellence Confocal Microscopy Core (S. Wang and G. Laevsky); D. Wu (NIH) and the members of the Riken BioResource Research Center for providing *Emx2* mouse lines; L. Howard and the veterinary team at the Houston Zoo; C. Hu for marsupial illustrations; F. Rogers for help with microscopy and statistics; J. Kim for help with genome assembly; S. Kocher and G. Deshpande for insights and discussion; C. Perdomo, E. Mallarino and A. Mallarino for logistical support. Genome assembly was performed in association with the DNA Zoo Consortium (www.dnazoo.org). DNA Zoo acknowledges support from Illumina, IBM and the Pawsey Supercomputing Center. This project was supported by an NIH grant to R.M. (R35GM133758). R.M. is partially supported by the Searle Scholars Program, the Sloan Foundation and the Vallee Scholars Program. E.L.A. was supported by grants from the Welch Foundation (Q-1866), an NIH Encyclopedia of DNA Elements Mapping Center Award (UM1HG009375), a US–Israel Binational Science Foundation Award (2019276), the Behavioral Plasticity Research Institute (NSF DBI-2021795), NSF Physics Frontiers Center Award (NSF PHY-2210291) and an NIH CEGS (RM1HG011016-01A1). J.A.M. was supported by an NSF fellowship (DGE-2039656). C.Y.F. was supported by an NIH fellowship (F32 GM139240-01). H.S. and B.J.B. were partially supported by an NIH training grant (T32GM007388). M.V.P. and Q.N. are supported by NIH grant R01-AR079150. M.V.P. is also supported by LEO Foundation grants LF-AW-RAM-19-400008 and LF-OC-20-000611, and W.M. Keck Foundation grant WMKF-5634988.

**Author contributions** R.M. conceived the project, secured funding and is the lead corresponding author. J.A.M. and R.M. led the design of all experiments, assembled the figures, analysed the data and wrote the manuscript with input from all authors. O.D., E.L.A., C.Y.F., D.W., A.D.O., W.Y., Z. Colaric and P.K. sequenced and assembled all genomes. J.A.M., C.Y.F. and R.M. designed the comparative genomics pipeline, and J.A.M. and C.Y.F. performed the analyses. W.W. performed the sugar glider ATAC–seq and J.A.M. analysed the data. J.A.M. and M.R.J. performed the Micro-C experiments and J.A.M., C.Y.F. and M.R.J. analysed the data. B.O. collected tissue samples for ChIP–seq and J.A.M. analysed the data. R.M. and S.A.M. designed shRNA constructs and prepared lentiviruses. J.A.M., S.A.M. and S.L. performed in vitro experiments. J.A.M. and C.Y.F. performed lentiviral injections and J.A.M., S.A.M. and R.M. analysed and characterized the resulting phenotypes, including histology and RNA-seq. S.A.M., R.M., J.A.M., H.S. and B.J.B. performed IHC, hybridization chain reactions and

whole-mount in situ hybridizations. S.A.M. and R.M. performed mouse overexpression transgenic experiments. Z. Chen and E.Z.K. performed the LacZ reporter assays. R.R. and M.V.P. performed the mouse RNAscope experiments. A.A.A., B.J.B. and M.V.P. analysed the spatial transcriptomics and scRNA-seq datasets with help from Q.N. G.K. and T.S. performed scanning electron microscopy. J.M.G. and A.P. generated the immortalized sugar glider fibroblasts. J.S.L. and A.M. provided tissues for *Macropus giganteus, M. fuliginosus* and *P. gymnotis,* and J.P.F. provided tissue for *Vombatus ursinus.*

**Competing interests** The authors declare no competing interests.

**Additional information**
**Correspondence and requests for materials** should be addressed to Erez Lieberman Aiden or Ricardo Mallarino.

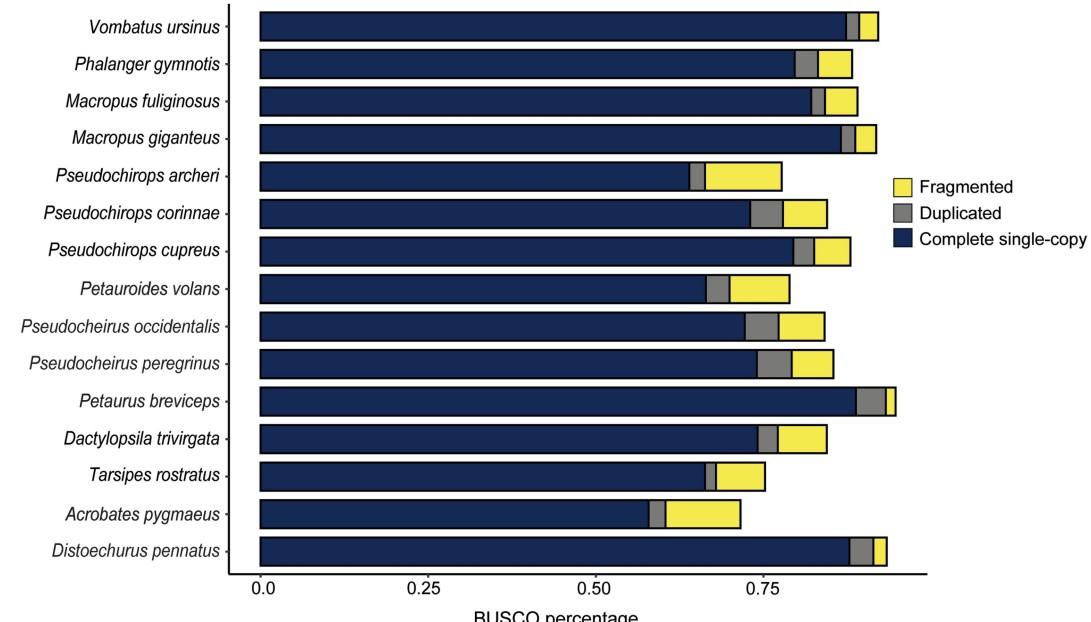

**Extended Data Fig. 1 | Comparison of benchmarking universal single-copy ortholog (BUSCO) recovery for all genomes sequenced and assembled in this study.**

**a**

## Glider Accelerated Regions (GARs)

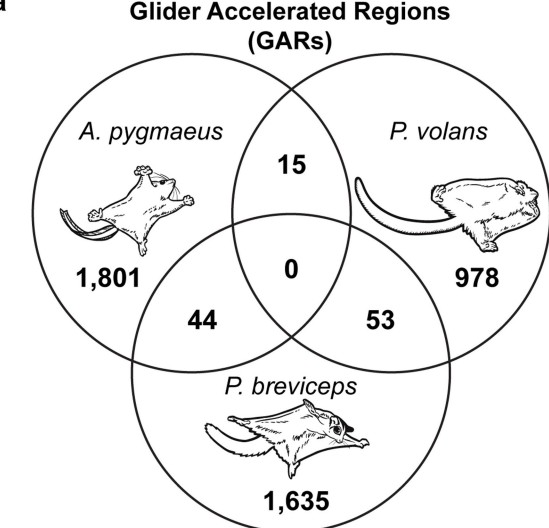

*A. pygmaeus*

*P. volans*

1,801

44

53

978

*P. breviceps*

1,635

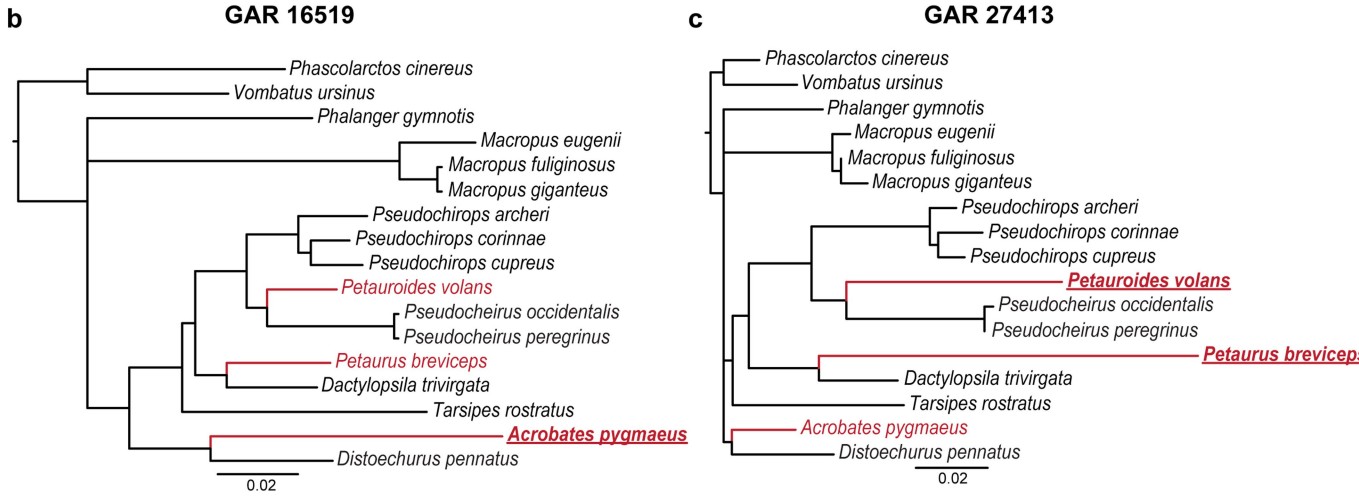

**b** **GAR 16519**

*Phascolarctos cinereus*
*Vombatus ursinus*
*Phalanger gymnotis*
*Macropus eugenii*
*Macropus fuliginosus*
*Macropus giganteus*
*Pseudochirops archeri*
*Pseudochirops corinnae*
*Pseudochirops cupreus*
*Petauroides volans*
*Pseudocheirus occidentalis*
*Pseudocheirus peregrinus*
*Petaurus breviceps*
*Dactylopsila trivirgata*
*Tarsipes rostratus*
**Acrobates pygmaeus**
*Distoechurus pennatus*

0.02

**c** **GAR 27413**

*Phascolarctos cinereus*
*Vombatus ursinus*
*Phalanger gymnotis*
*Macropus eugenii*
*Macropus fuliginosus*
*Macropus giganteus*
*Pseudochirops archeri*
*Pseudochirops corinnae*
*Pseudochirops cupreus*
**Petauroides volans**
*Pseudocheirus occidentalis*
*Pseudocheirus peregrinus*
**Petaurus breviceps**
*Dactylopsila trivirgata*
*Tarsipes rostratus*
*Acrobates pygmaeus*
*Distoechurus pennatus*

0.02

**Extended Data Fig. 2 | Glider Accelerated Regions (GARs) in marsupial gliding species. a**, Venn diagram showing number of unique and shared GARs among the three glider species (*Acrobates pygmaeus*, *Petauroides volans*, and *Petaurus breviceps*). **b** and **c**, Example trees showing branch lengths of GARs that were unique to one gliding species (**b**) or shared among two glider species (**c**). Gliding species are labelled in red and species in which the element is accelerated are underlined.

**a**

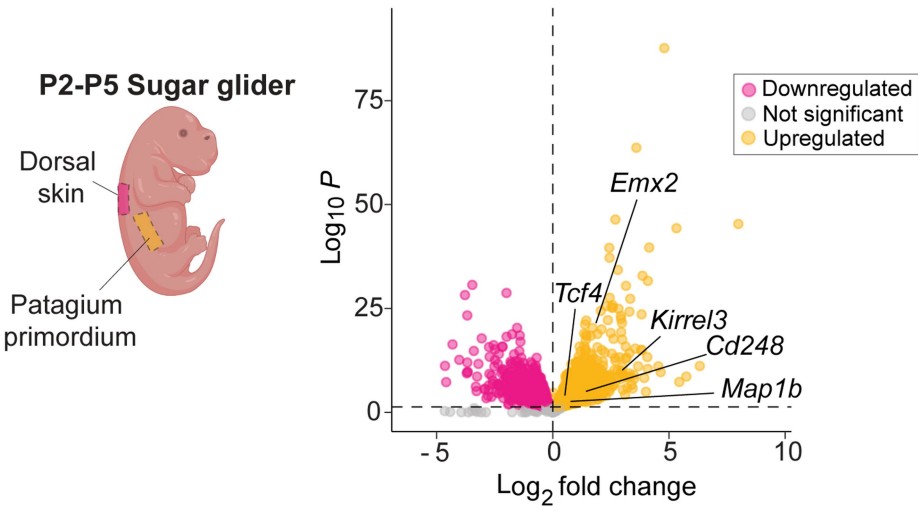

### Patagium vs. Dorsal skin

**P2-P5 Sugar glider**

Dorsal skin

Patagium primordium

- Downregulated
- Not significant
- Upregulated

*Emx2*

*Tcf4*

*Kirrel3*

*Cd248*

*Map1b*

**b**

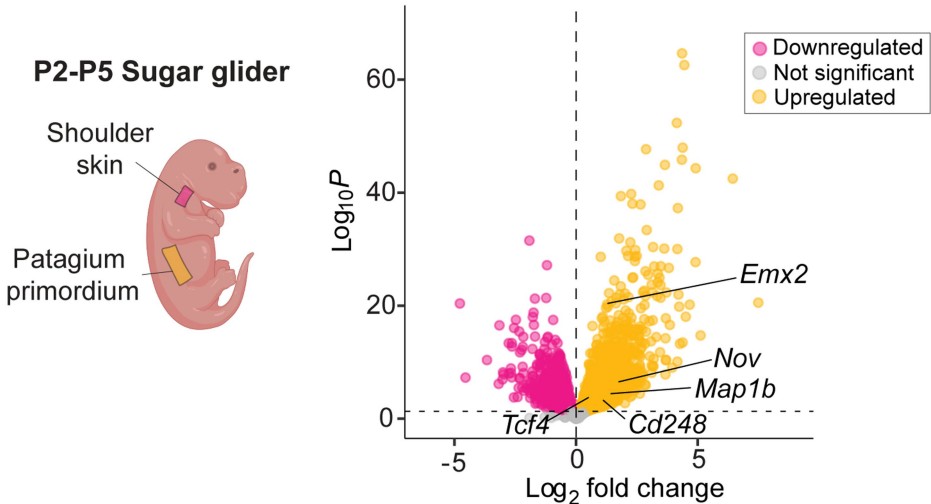

### Patagium vs. Shoulder skin

**P2-P5 Sugar glider**

Shoulder skin

Patagium primordium

- Downregulated
- Not significant
- Upregulated

*Emx2*

*Nov*

*Map1b*

*Tcf4*

*Cd248*

**Extended Data Fig. 3 | Differentially expressed genes between the patagium primordium and different skin regions. a-b**, Patagium vs Dorsal (**a**) and Patagium vs Shoulder (**b**). Yellow represents genes more highly expressed in the patagium; pink are genes more highly expressed in the dorsal skin; grey are genes without differential expression. Data corrected for multiple comparisons; FDR < 0.1. Joey schematics created with BioRender.com.

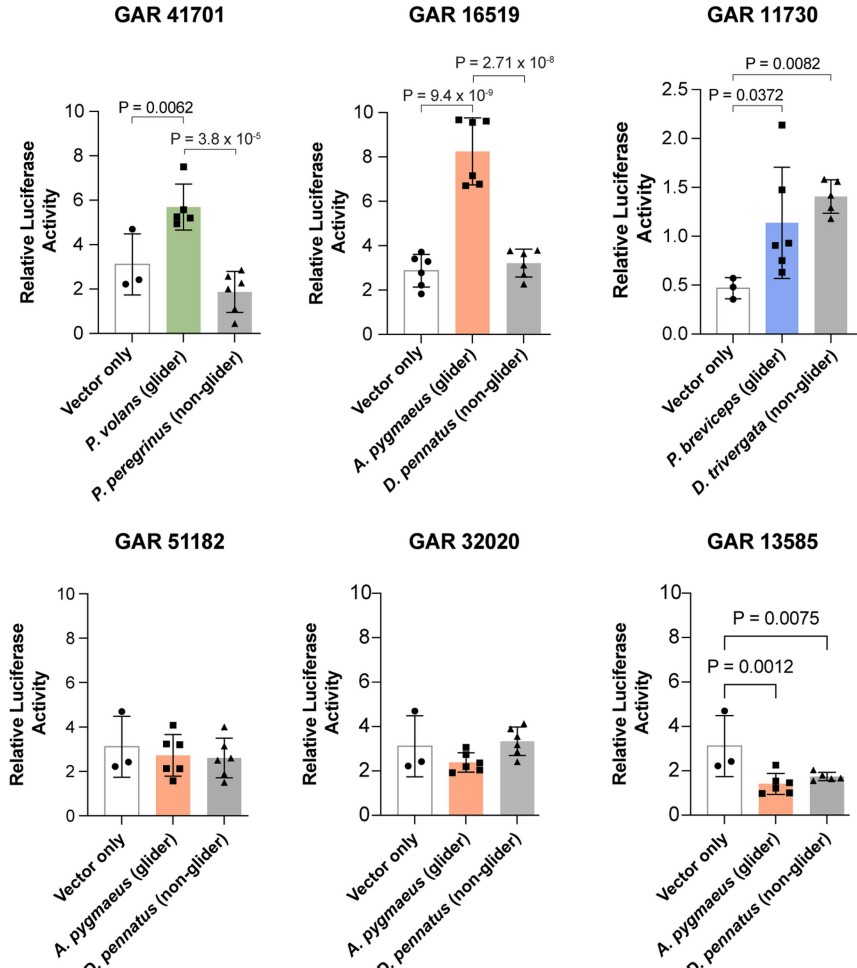

**Extended Data Fig. 4 | Relative luciferase activity (Mean + SE) for GARs and orthologous sequences of the corresponding non-glider sister species.** Statistical significance was assessed using one-way ANOVA tests. GAR 41701; *P. volans* (N = 5), *P. peregrinus* (N = 6), vector (N = 3): $P$ = 0.0062 (vector vs. *P. volans*), $P$ = 0.1373 (vector vs. *P. peregrinus*), $P$ = 3.8 × 10$^{-5}$ (*P. volans vs P. peregrinus*); GAR 16519; *A. pygmaeus* (N = 6), *D. pennatus* (N = 6), vector (N = 3): $P$ = 9.4 × 10$^{-9}$ (vector vs. *A. pygmaeus*), $P$ = 0.5568 (vector vs. *D. pennatus*), $P$ = 2.71 × 10$^{-8}$ (*A. pygmaeus vs D. pennatus*); GAR 11730; *P. breviceps* (N = 6), *D. trivergata* (N = 5), vector (N = 3): $P$ = 0.0372 (vector vs. *P. breviceps*), $P$ = 0.0082 (vector vs. *D. trivergata*), $P$ = 0.2895 (*P. breviceps vs D. trivergata*). GAR 51182; *A. pygmaeus* (N = 6), *D. pennatus* (N = 6), control (N = 3): $P$ = 0.5388 (vector vs. *A. pygmaeus*), $P$ = 0.4228 (vector vs. *D. pennatus*), $P$ = 0.8150 (*A. pymaeus vs D. pennatus*); GAR 32020; *A. pygmaeus* (N = 6), *D. pennatus* (N = 6), control (N = 3): $P$ = 0.2724 (vector vs. *A. pygmaeus*), $P$ = 0.7296 (vector vs. *D. pennatus*), $P$ = 0.0865 (*A. pygmaeus vs D. pennatus*); GAR 13585; *A. pygmaeus* (N = 6), *D. pennatus* (N = 5), control (N = 3): $P$ = 0.0012 (vector vs. *A. pymaeus*), $P$ = 0.0075 (vector vs. *D. pennatus*), $P$ = 0.3902 (*A. pymaeus vs D. pennatus*).

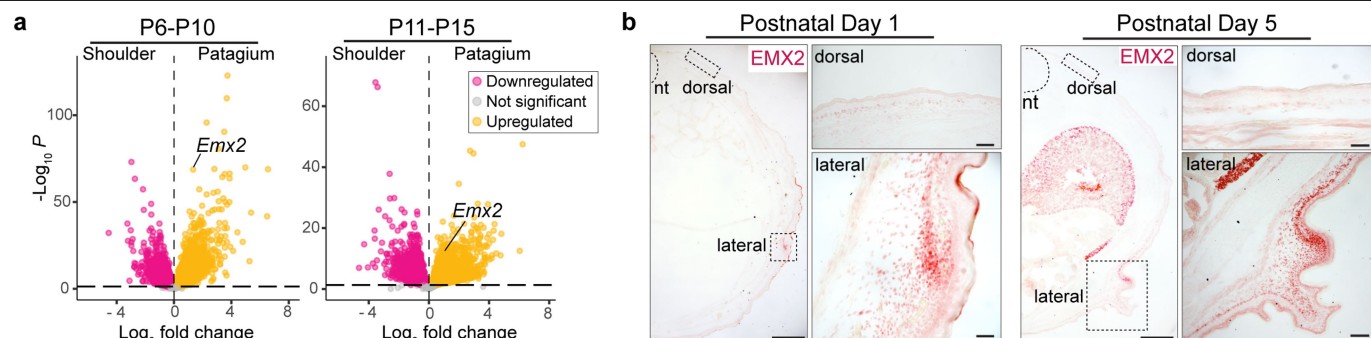

**Extended Data Fig. 5 | *Emx2* is upregulated for a prolonged period and specifically expressed in the lateral skin mesenchyme. a**, Differential gene expression between patagium and shoulder skin (Data corrected for multiple comparisons; FDR < 0.1) from postnatal day 6 (P6) to P10 (left) and P11 to P15 (right). Yellow represents genes more highly expressed in the patagium; pink are genes more highly expressed in the shoulder; grey are genes without differential expression. **b**, Immunohistochemistry for EMX2 in sugar glider tissue prior to patagium outgrowth (P1) and as outgrowth begins (P5). Scale bars in **b**: 500 μm (zoomed out) and 100 μm (zoomed in).

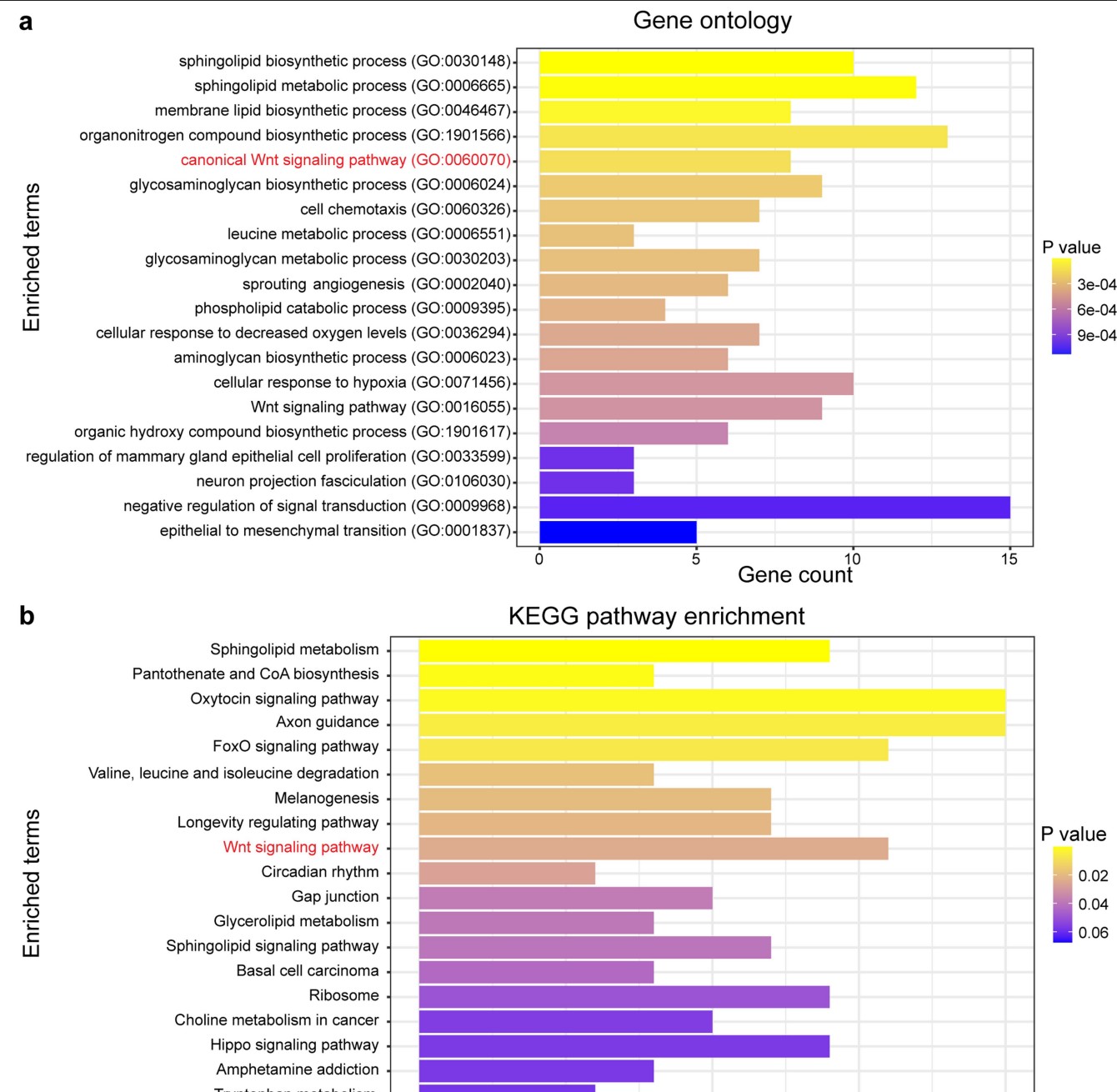

**Extended Data Fig. 6 | Gene Ontology (top) and KEGG pathway (bottom) enrichment analysis for genes downregulated in patagia injected with shRNAEmx2-3.** Wnt signalling (shown in red font) was significantly enriched in both analyses. Data corrected for multiple comparisons; FDR < 0.5).

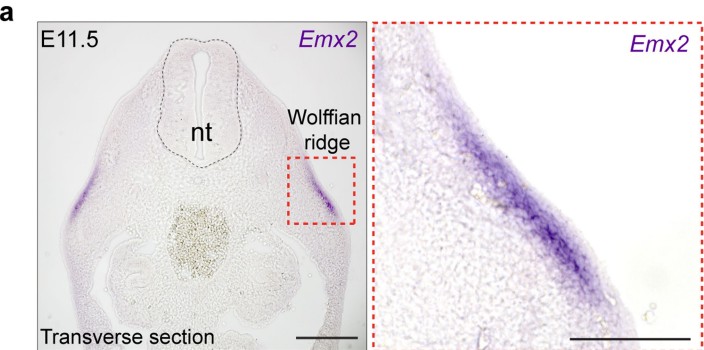

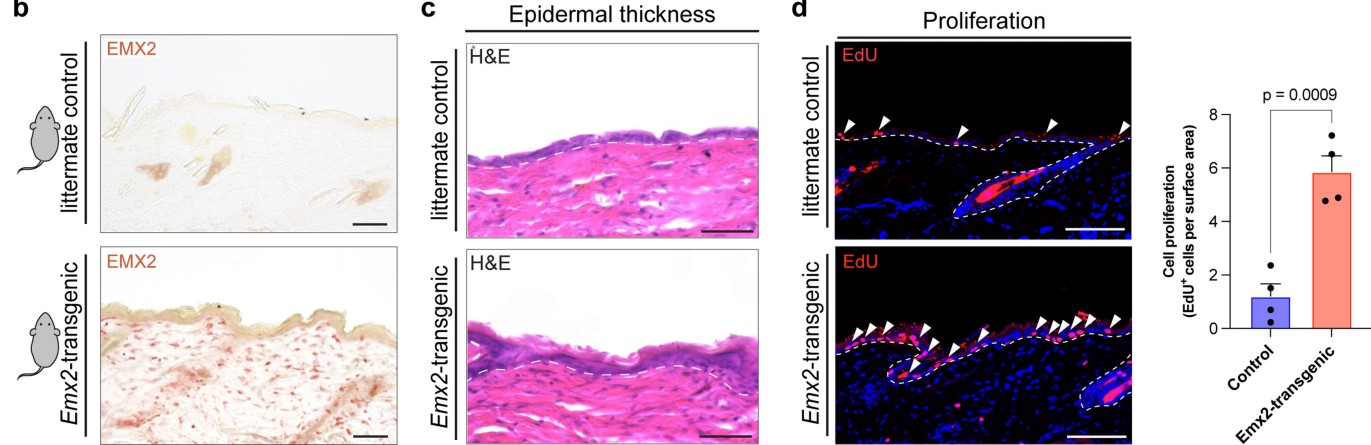

**Extended Data Fig. 7 | Spatial expression and function of *Emx2* in mice.**
**a**, Transversal sections of E11.5 laboratory mouse showing expression of *Emx2* in the Wolffian ridge. **b**, Immunohistochemistry showing the distribution of EMX2 in the skin of *Pdfgra^CreERT2/+*/*Rosa^Emx2-GFP/+* double transgenic animals and control littermates. **c, d**, Relative to control littermates, double transgenic mice over-expressing *Emx2* in dermal fibroblasts show a significant increase in epidermal thickness, as seen by H&E stains (**c**) and cellular proliferation, as established by EdU incorporation (**d**). Statistical significance in panel **d** (N = 4; *P* = 0.0009) was assessed using a general mixed effects model one-way ANOVA test. Dotted line in **c,d** denotes the dermis-epidermis boundary. Mean + SE are plotted in **d**. Scale bars: 500 μm (zoomed out) and 100 μm (zoomed in) in (**a**); 100 μm in **b-d**.

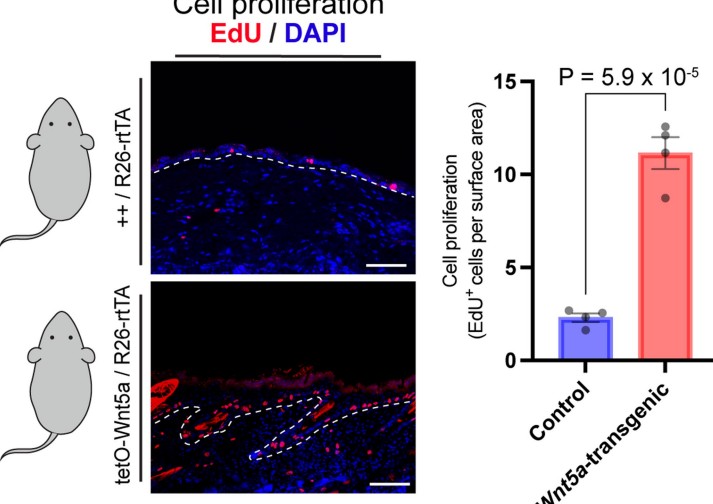

**Extended Data Fig. 8 | *Wnt5a* overexpression promotes cell proliferation in mouse skin.** Induction of *Wnt5a* in postnatal day 37 laboratory mice results in a significant increase in cellular proliferation, as measured by EdU incorporation (Mean + SE). Shown are comparisons between double transgenic ($R26rtTA^{HET};tetO\text{-}Wnt5a^{HET}$) and control littermates $R26rtTA^{HET};wt/wt$. Statistical significance (N = 4; $P = 5.9 \times 10^{-5}$) was assessed using a general mixed effects model one-way ANOVA test. Dotted line delineates the dermis-epidermis boundary. Scale bars: 100 μm.

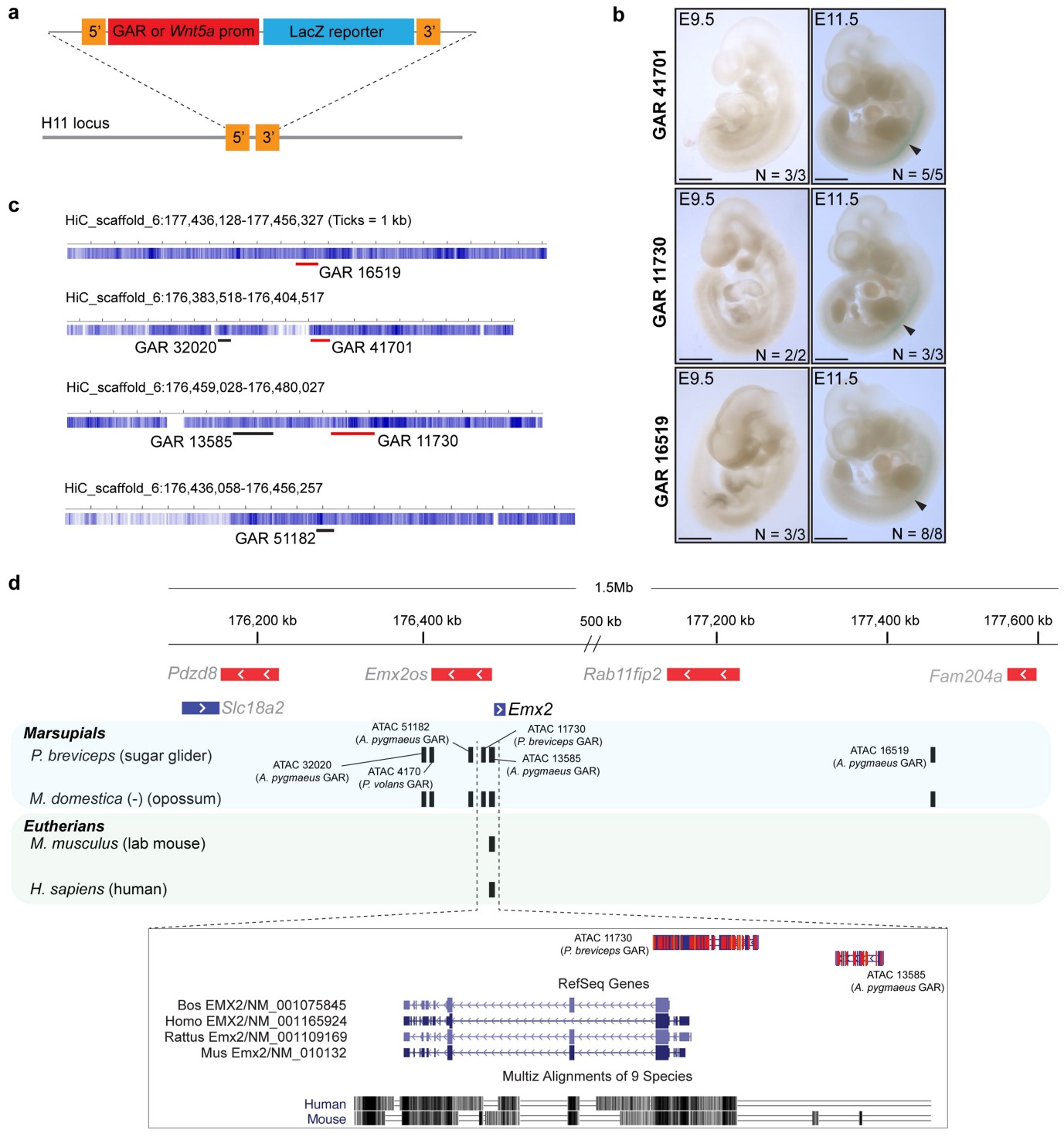

**Extended Data Fig. 9 | Conservation analysis and lab mouse LacZ transgenic assays of Glider Accelerated Regions (GARs). a, b,** LacZ transgenic assays in E9.5 and E11.5 laboratory mouse embryos. Shown is the schematic of the construct and insertion site used to test GAR activity (**a**) and results from the assays (**b**). E11.5 embryos in (**b**) show expected neural tube staining from basal *Shh* promoter activity (black arrowheads). Scale bars: 500 μm (E9.5) and 1000 μm (E11.5). **c,** Conservation scores for genomic regions surrounding GARs and visualized against the reference sugar glider sequence as a heatmap (darker colours indicate higher conservation). Labelled in red are

GARs that showed functional activity through luciferase assays, whereas GARs in black are those that did not. **d,** Opossum (*Monodelphis domestica*) genomic region containing the *Emx2* locus and all *Emx2*-associated GARs. All *Emx2*-associated GARs have high-confidence orthologous matches with the opossum. Only the GAR located in the *Emx2* promoter had a high-confidence orthologous match with eutherian genomes (*Mus musculus* and *Homo sapiens*). The orientation of this region in opossum is inverted with respect to the other species.

| | |
|---|---|

# Reporting Summary

## Statistics

For all statistical analyses, confirm that the following items are present in the figure legend, table legend, main text, or Methods section.

| n/a | Confirmed | |
|---|---|---|
| ☐ | ☒ | The exact sample size (*n*) for each experimental group/condition, given as a discrete number and unit of measurement |
| ☐ | ☒ | A statement on whether measurements were taken from distinct samples or whether the same sample was measured repeatedly |
| ☐ | ☒ | The statistical test(s) used AND whether they are one- or two-sided *Only common tests should be described solely by name; describe more complex techniques in the Methods section.* |
| ☐ | ☒ | A description of all covariates tested |
| ☐ | ☒ | A description of any assumptions or corrections, such as tests of normality and adjustment for multiple comparisons |
| ☐ | ☒ | A full description of the statistical parameters including central tendency (e.g. means) or other basic estimates (e.g. regression coefficient) AND variation (e.g. standard deviation) or associated estimates of uncertainty (e.g. confidence intervals) |
| ☐ | ☒ | For null hypothesis testing, the test statistic (e.g. *F*, *t*, *r*) with confidence intervals, effect sizes, degrees of freedom and *P* value noted *Give P values as exact values whenever suitable.* |
| ☒ | ☐ | For Bayesian analysis, information on the choice of priors and Markov chain Monte Carlo settings |
| ☒ | ☐ | For hierarchical and complex designs, identification of the appropriate level for tests and full reporting of outcomes |
| ☒ | ☐ | Estimates of effect sizes (e.g. Cohen's *d*, Pearson's *r*), indicating how they were calculated |

*Our web collection on statistics for biologists contains articles on many of the points above.*

## Software and code

Policy information about availability of computer code

| | |
|---|---|
| Data collection | NIS Elements v5 (Nikon) was used to acquire microscopy images. |
| Data analysis | megahit v1.1.4.2, redundans v0.14a, ragtag v2.0.1,  NGmerge v0.2_dev, picard MarkDuplicates v2.21.4-SNAPSHOT, bowtie2 v2.4.2, IDR v2.0.4.2, bedtools v2.27.1,  RAxML v8.2.12, R v3.5.3, rphast v1.6.9, mafft v7.453, trimAl v1.4.rev15, bwa v0.7.17-r1188, pairtools v0.3.0, juicertools v1.22.01, samtools v1.12, mustache v1.0.1, cooler v0.8.5, higlass-manage v0.8.0, clodius v0.3.5, DESeq2 v1.34.0, LiftOff v1.6.3, BUSCO v5.4.4, FIJI v2.1.0, gffread v0.12.7, trimmomatic v0.39, MEME v5.5.4, BLAT, STAR v2.7.9a, feattureCounts v2.0.1, MACS2 v2.2.7.1, Seurat package v4.3.0.<br><br>Code availability<br>Code used for all analyses is available in a FigShare repository: https://figshare.com/s/81cf39b7de363f1526a1 |

For manuscripts utilizing custom algorithms or software that are central to the research but not yet described in published literature, software must be made available to editors and reviewers. We strongly encourage code deposition in a community repository (e.g. GitHub). See the Nature Portfolio guidelines for submitting code & software for further information.

## Data

Policy information about availability of data

All manuscripts must include a data availability statement. This statement should provide the following information, where applicable:

- Accession codes, unique identifiers, or web links for publicly available datasets
- A description of any restrictions on data availability
- For clinical datasets or third party data, please ensure that the statement adheres to our policy

Data availability: All genome assemblies reported, and corresponding genome sequencing reads are submitted under NCBI PRJNA512907. The ATAC-seq, ChIP-seq, Micro-C, and RNA-seq reads are submitted under NCBI BioProject: PRJNA1078418

## Human research participants

Policy information about studies involving human research participants and Sex and Gender in Research.

| Reporting on sex and gender | n/a |
| Population characteristics | n/a |
| Recruitment | n/a |
| Ethics oversight | n/a |

Note that full information on the approval of the study protocol must also be provided in the manuscript.

# Field-specific reporting

Please select the one below that is the best fit for your research. If you are not sure, read the appropriate sections before making your selection.

☒ Life sciences       ☐ Behavioural & social sciences       ☐ Ecological, evolutionary & environmental sciences

For a reference copy of the document with all sections, see nature.com/documents/nr-reporting-summary-flat.pdf

# Life sciences study design

All studies must disclose on these points even when the disclosure is negative.

| Sample size | No predetermined sample size calculation was performed in this study. Sample size was chosen based on the numbers accepted by the field to run the relevant statistical tests described. The sample sizes used are sufficient to provide the desired statistical power for our analyses |
| Data exclusions | No data were excluded for experiments involving micro-C, RNA-seq, ATAC-seq, immunohistochemistry, histology, in situ hybridizations, or lentiviral experiments. We carried out outlier analysis for luciferase assays, as luminescence data from plate readers can be noisy. For all our GARs, we used 6 technical replicates and, in a few cases (as noted in the figure legends), we excluded a maximum of one data point per treatment based on our statistical outlier analysis. |
| Replication | All experiments involving antibody stains and in situ hybridizations were carried out in three different individuals and they all yielded the same results. LacZ based transgenic experiments were replicated in at least two animals per construct tested and all yielded the same results. Each luciferase assay was carried out in duplicate and both experiments yielded the same result. All other experiments, including RNAseq, ATACseq, ChIPseq, and Micro-C were done with multiple biological replicates, as indicated in the figure legends and methods. |
| Randomization | Samples/organisms were allocated into groups based on genotype/experimental conditions |
| Blinding | All measurements were taken from images in which the researcher performing the measurements was unaware of which images corresponded to which genotypes/experimental conditions |

# Reporting for specific materials, systems and methods

We require information from authors about some types of materials, experimental systems and methods used in many studies. Here, indicate whether each material, system or method listed is relevant to your study. If you are not sure if a list item applies to your research, read the appropriate section before selecting a response.

## Materials & experimental systems

| n/a | Involved in the study |
|---|---|
| ☐ | ☒ Antibodies |
| ☐ | ☒ Eukaryotic cell lines |
| ☒ | ☐ Palaeontology and archaeology |
| ☐ | ☒ Animals and other organisms |
| ☒ | ☐ Clinical data |
| ☒ | ☐ Dual use research of concern |

## Methods

| n/a | Involved in the study |
|---|---|
| ☐ | ☒ ChIP-seq |
| ☒ | ☐ Flow cytometry |
| ☒ | ☐ MRI-based neuroimaging |

# Antibodies

| Antibodies used | Primary antibodies: anti-H3K27ac antibody (Abcam ab4729: lot # GR232896-1; 4ug), Anti-EMX2 antibody (Novus NBP2-39052: lot # 27711; (dilution: 1:50), anti-IgG antibody (Millipore 12-370; lot # 297424; 4 ug). Anti-GFP (Novus Biologicals # NB100-1614, Dilution: 1:200). anti-KRT14 (BioLegend #905301, Dilution: 1:1000)<br>Secondary antibodies: Alea-Fluor 488 (ThermoFisher; ab150169; Dilution 1:500); Goat Anti-Rabbit Biotinylated (Vector Labs, R.T.U. (BP-9100-50; Ready-to-use dilution) |
|---|---|
| Validation | Primary antibody validation:<br>1) Anti-EMX2 antibody: The target protein was examined with an antibody independent strategy (in this case, RNA sequencing in human tissue) and compared with results from an antibody-dependent strategy. A correlation between these two strategies indicated specificity between the antibody and Emx2. Manufacturer: https://www.novusbio.com/products/emx2-antibody_nbp2-39052<br>2) anti-H3K27ac antibody (Confirmed species reactivity: Mouse, Rat, Cow, Human; Predicted: Chicken, Xenopus laevis, Arabidopsis thaliana, Drosophila melanogaster, Monkey, Zebrafish, Plasmodium falciparum, Rice, Cyanidioschyzon merolae. Applications:ICC/IF, WB, IHC-P, ChIP; Validation: Manufacturer https://www.abcam.com/products/primary-antibodies/histone-h3-acetyl-k27-antibody-chip-grade-ab4729.html)<br>3) anti-IgG antibody: Validated by IP/WB as a non-specific IgG control. Manufacturer: https://www.emdmillipore.com/US/en/product/Normal-Rabbit-IgG,MM_NF-12-370)<br>4) anti-GFP: Validated on transgenic mice expressing recombinant GFP; https://www.novusbio.com/products/gfp-antibody_nb100-1614#datasheet<br>5) anti-KRT14 (Confirmed Species reactivity: Human; Applications: Immunohistochemistry; Validation: Manufacturer -https://d1spbj2x7qk4bg.cloudfront.net/Files/Images/media_assets/pro_detail/datasheets/905301_V09.pdf?v=20230922064116) |

# Eukaryotic cell lines

Policy information about cell lines and Sex and Gender in Research

| Cell line source(s) | Sugar glider dermal fibroblasts from a male joey (25 days of age) |
|---|---|
| Authentication | Sugar glider dermal fibroblasts were authenticated by RNA sequencing |
| Mycoplasma contamination | All cell lines tested negative for Mycoplasma |
| Commonly misidentified lines<br>(See ICLAC register) | No commonly misidentified cell lines were used in this study |

# Animals and other research organisms

Policy information about studies involving animals; ARRIVE guidelines recommended for reporting animal research, and Sex and Gender in Research

| Laboratory animals | Mus Musculus (laboratory mouse) from the following genotypes: (1) RosaEmx2-GFP, (2) Emx2-Cre, (3) Pdgfra-CreERT2, (4) FVB/N769Tg(tetO-Wnt5a)17Rva/J, (5) B6.Cg Gt(ROSA)26Sortm1(rtTA*M2)Jae/J. The age of all mice ranged from Embryonic Day 11.5 to Postnatal day 40. Petaurus breviceps (sugar glider) of ages ranging from Postnatal day 2-14 |
|---|---|
| Wild animals | Study did not involve wild animals |
| Reporting on sex | Sex was not considered in study design as the patagium growth is indistinguishable between sexes. |
| Field-collected samples | Study did not include field-collected samples |
| Ethics oversight | Princeton University's Institutional Animal Care and Use Committee approved all experiments and protocols. |

Note that full information on the approval of the study protocol must also be provided in the manuscript.

# ChIP-seq

## Data deposition

☒ Confirm that both raw and final processed data have been deposited in a public database such as GEO.

☒ Confirm that you have deposited or provided access to graph files (e.g. BED files) for the called peaks.

| | |
|---|---|
| Data access links<br>*May remain private before publication.* | ChIP-seq reads are submitted under NCBI BioProject: PRJNA1078418 |
| Files in database submission | Emx2_Pbrev_ChIP_2456432000_peaks.narrowPeak, Control_Pbrev_ChIP_2456432000_peaks.narrowPeak,<br>zr2392_2_1_NGm_TM_X1k_vs_HiC.sorted.rmDup.rmMulti.filtered_nolambda_g2456432000_peaks.broadPeak,<br>zr2392_3_1_NGm_TM_X1k_vs_HiC.sorted.rmDup.rmMulti.filtered_nolambda_g2456432000_peaks.broadPeak,<br>zr2392_2_3_NGm_TM_X1k_vs_HiC.sorted.rmDup.rmMulti.filtered_nolambda_g2456432000_concatenated_merged_peaks<br>Numbered.broadPeak |
| Genome browser session<br>(e.g. UCSC) | No longer applicable |

## Methodology

| | |
|---|---|
| Replicates | For H3K27Ac ChIP-seq, chromatin was divided into 2 experimental (each experimental sample consisting of pooled tissue from 6-7 joeys) and 1 control sample and ChIP assays were performed in triplicate for each of the experimental samples and in duplicate for the negative control sample. In all cases, we used 7 μg of chromatin and 4 μg of antibody. ChIP DNA was then processed into 3 standard Illumina ChIP-seq libraries (2 experimental and 1 control) and sequenced to generate ~25 million reads per sample.<br><br>EMX2 ChIP-Seq was carried out in a similar way as described above, except that only one experimental (pooled tissue from 6-7 joeys) and one control library were generated and sequenced. |
| Sequencing depth | H3K27ac: paired-end; control 56,614,872 97.3% mapped; experimental 1: 56,957,310 97.4% mapped; experimental 2: 70,392,940 97.4% mapped<br><br>Emx2: single-end; experimental: 41,672,698 ~100% mapped ; control: 43,909,990 ~100% mapped |
| Antibodies | anti-H3K27ac antibody (Abcam ab4729: lot # GR232896-1), Anti-EMX2 antibody (Novus NBP2-39052: lot # 27711), anti-IgG antibody (Millipore 12-370; lot # 297424). |
| Peak calling parameters | H3K27ac: --broad -f BAMPE -g 2456432000 -q 0.05 --nolambda<br>Emx2: -f BAMPE -g 2456432000 |
| Data quality | H3k27ac: The input was used as a background control. Peaks between the two replicates were concatenated and overlapping peaks were merged using BEDTools merge.<br>Emx2: Input was used as a background control. FIMO was used to scan for enriched motif and the canonical Emx2 binding motif was recovered. |
| Software | NGmerge v0.2_dev, picard MarkDuplicates v2.21.4-SNAPSHOT, bowtie2 v2.4.2, bedtools v2.27.1 |

