## [Peer Review File · Nature]

Manuscript Title: Emx2 underlies the development and evolution of marsupial gliding membranes

Reviewer Comments & Author Rebuttals

Reviewer Reports on the Initial Version:

Referees' comments:

Referee #1 (Remarks to the Author):

The study by Moreno et al. investigates the formation of the patagium, a type of gliding membrane that has evolved independently in three marsupial species from the Superfamily Petauroidea. The authors generate genome assemblies for 14 marsupial species, including 2 gliding and 8 non-gliding petauroid species and several outgroups, as well as an improved assembly for the sugar glider. By combining ATAC-seq and H3K27Ac ChIP-seq data from patagium primordium of the sugar glider, they annotate potential cis-regulatory elements (CREs) and subsequently perform comparative genomics analyses to identify orthologous regions across these species. By doing so, they identify Glider Accelerated Regions (GAR), which are those putative CREs that accumulate a substantial number of nucleotides substitutions in glider species, while being remarkably conserved in other species. By focusing on genes surrounded by numerous GARs, and also upregulated during patagium development as observed in RNA-seq data, they identify Emx2 as a candidate gene for the evolution of this trait. The authors then perform several in vivo experiments in the sugar glider and in mouse, to validate their findings. Specifically, they inactivate Emx2 expression by transgenic experiments during patagium formation, revealing that this gene is required for patterning and growth. By combining RNA-seq and ChIP-seq data, they provide evidence that Emx2 may directly regulate key patterning genes, notably Wnt5a, a gene recently demonstrated by the group to be involved in patagium formation (Feigin et al, Sci Adv, 2023). By generating and analyzing Micro-C data from patagium, the authors focus on one particular GAR that display high interaction frequency with Emx2. This GAR is then validated as a putative enhancer by performing luciferase assays in cultured cells. Finally, the authors examine the pattern of expression of Emx2 in the Wolffian ridge of mice, a region that shares anatomical similarities with the patagium of gliders. This analysis shows that Emx2 expression is present at later developmental stages in gliders compared to mice, in which this occurs during the early prenatal period. By functionally mimicking this late expression in transgenic mice, the authors recapitulate certain features of patagium development, including an increase in epidermal thickness, as well as an increase in cell proliferation and mesenchymal cell density.

Overall, this study illustrates how novel approaches can shed light into the evolutionary origin of phenotypical traits. The enablement of the sugar glider as a transgenic animal model is highly original and innovative, thus providing the adequate means for functional validations. The manuscript is well written and easy to understand. Overall, the authors make a compelling case for the involvement of Emx2 in patagium formation, as they similarly did for Wnt5a. However, the evidence supporting that the glider-specific Emx2 expression results from regulatory convergence of CRE activity is still preliminary and limits the impact of the study.

Major comments:

- The major claim of this manuscript, as reflected in the title, is that the patagium has evolved in several marsupial gliders through convergent regulatory evolution. Yet, the evidence to support this claim is simply a screening for accelerated regions. This is coupled with the functional validation of just one putative GAR in one species, which appears insufficient at all lights. What happens to the other GARs identified by the authors? Are they also functional? If the authors want to make such broad claims, they should provide evidence that changes in Emx2-related CRE activity have occurred in all the different glider species, not just in one.

- Regarding the assays to validate putative CRE activity, the authors perform these experiments in an immortalized dermal fibroblast cell line. I have trouble understanding why dorsal cells are a suitable model to investigate the activity of patagium CREs, simply because Emx2 is not expressed or it is at very low levels in dorsal tissue. In fact, the dorsal tissue is employed by the authors as a negative control for Emx2 expression in RNA-seq analysis.

Furthermore, a luciferase assay in a cell line does not provide any spatial information on the activity of these putative CREs. The authors have taken a considerable effort in implementing a transgenesis for the sugar glider, a system that opens ample opportunities for experimental validation. Have the authors consider validating these putative CREs *in vivo* by injecting the sequence coupled with reporter genes in the developing patagium (i.e. transient transgenesis)? Classical reporter assays in transgenic mice could also provide important insights on the evolution of these CREs: they could reveal if they can be activated in other Emx2-expressing tissues, potentially being regulated by the same transcription factors (TFs).

- In that respect, have the authors investigated if novel transcription factors binding sites have appeared in the GARs? Are there any common novel signatures across all Emx2 GARs? Do these potential new binding sites correspond to TFs that are also expressed in patagium?

- Besides individual gliding species, were all 3 gliding or combinations of 2 species used together to detect GARs? Could this be a reason for the low number of shared GARs?

- The authors suggest that Emx2 expression is prolonged in the sugar glider in comparison to mice, where it only expresses prenatally. However, they do not show any evidence of prenatal expression of Emx2. *In situ* hybridizations for Emx2 in similar prenatal stages as in mouse (Figs 5a and b) would be recommended to support their hypothesis.

Minor comments

- Besides those GARs within the Emx2 TAD, have the authors evaluated if there are GARs also within the Emx2 ChIP-seq peaks? Could those be associated with genes that are differentially expressed in patagium? This might reveal functional rewirings of the Emx2 networks that might be relevant for patagium formation.

- Are there any Emx2 binding sites in the Wnt5a locus that are conserved across non-gliding or in other species? Could there be evidence of conservation for the Emx2-Wnt5a axis in other species, or is it specific for glider patagium?

- Line105: it would be worth to inform the non-expert reader of the link between the CHIP-seq and ATAC-seq data with regards to identifying CREs.

- Line 123 What does the authors refer with “rapid” regulatory evolution? What is the time scale here?

- The Micro-C data from figure 4c looks very different from the data in Fig 2c. Maybe the authors could consider displaying the data with a different resolution or color scale, so that the 3D structures are more visible.

- Line 299: It is inappropriate to refer to this putative CREs as enhancers, because just one of these elements was functionally tested.

Referee #2 (Remarks to the Author):

In this manuscript, Moreno et al. sequenced and assembled several marsupial genomes and identified accelerated regions in glider versus non-glider species (GARs) as proxy for regulatory variation in the latter. To gain insight into the relevance of these alterations in patagium development in the sugar glider, the authors generated ATAC-seq and H3K27ac CHIP-seq datasets and integrated them with RNA-seq analyses (generated by the same group in a previous study; Feigin et al., 2023), leading to the identification of Emx2 as a good candidate to have accumulated regulatory alterations causally related to the formation of the patagium. Through a series of elegant experiments, the authors show that Emx2 is required in vivo for patagium development being Wnt5a (another gene relevant for patagium formation; Feigin et al. 2023) likely one of its direct transcriptional targets. Supported by gain of function experiments in the mouse, the authors hypothesize that the regulatory alterations in GARs identified in the Emx2 locus may participate in maintaining postnatal Emx2 expression in the flank dermis of the joey, leading to changes in cell proliferation and density that are required for patagium formation. This is a beautiful and original manuscript that provides compelling evidence that Emx2 plays a fundamental role in the development of the patagium in the sugar glider. While I am positive about its relevance, there are a few aspects that should be taken into consideration in revising the manuscript:

1) the authors place significant weight on regulatory convergence at the Emx2 locus in different glider species. While the authors identify GARs in the Emx2 genomic landscape of three species with patagium, all the functional genomics and in vivo data was generated on sugar gliders (*P. breviceps*), so the claim on regulatory convergence is essentially based on one luciferase assay using GAR16518 orthologous regions from two glider species (*P. breviceps* and *A. pigmaeus*). There is no evidence provided indicating that *P. volans* GARs are indeed cis-regulatory regions controlling Emx2 levels in

the patagium nor that Emx2 plays a role in patagium development in this species. I do not see that the data supporting regulatory convergence is strong enough to make such a statement. This claim should be removed from the title and tuned-down throughout the text. Alternatively, additional evidence could be gathered, but it would minimally require RNA-seq from patagium primordia vs dorsal skin in other glider species and extensive characterization of their putative Emx2 GARs in vitro and in vivo (eg. via reporter constructs).

2) In relation to the previous point, have the authors checked all Emx2 GARs/orthologous sequences from all three gliding species in luciferase assays to evaluate if they also potentiate reporter transcription? While I do not think these experiments would suffice to provide compelling evidence for regulatory convergence, they would at least provide grounds for it to be discussed.

3) The authors show that the sugar glider distant GAR16518 enhancer contacts the Emx2 promoter and is able to increase reporter expression in vitro, while the orthologous sequence from a non-glider species cannot. This region, which is not a GAR in *P. breviceps*, does have a minor but statistically significant effect on luciferase expression, lower than that of *A. pigmaeus*. Have the authors tested the activity of GAR16518 from any of these two species in reporter assays in transgenic mice? The luciferase experiments they report may indicate that this element potentiates the transcriptional activity of a basal promoter through yet-unidentified transcription factors present in sugar glider fibroblasts, but the working hypothesis is that the roles of Emx2 in patagium development are, at least partially, driven by its heterochronic expression in the patagium primordia when compared to Emx2 expression pattern in mouse embryos. Assaying the activity of this element in mice (eg. if they observe that glider enhancers can drive expression to the inter-limb region in mouse embryos) may help elucidating if this particular element contributes to shaping the spatio-temporal pattern of expression of Emx2 in the patagium primordia. This experiment, while not conclusive if negative (mouse trans-acting factors may simply not bind the glider enhancers), is far simpler than transgenesis in joeys using constructs to drive reporter expression to the patagium primordia, which is the other possibility.

4) The regulatory relation between Emx2 and Wnt5a should be further supported. It is surprising that the authors limit their analysis to 5Kb upstream and downstream of the TSS of their 59 candidate targets, as tissue-specific transcription factors frequently exert their activities through distant cis-regulatory regions (GAR16518 is itself a good example of this). Are there additional Emx2-binding regions in the Wnt5a genomic landscape? Are there any distant interactions that can be predicted from the Micro-C dataset, as done for GAR16518? This data should be also included, as well as the ChIP-seq binding profile of Emx2 in the Wnt5a locus, with particular focus on the peak identified in Fig. 3j. Concerning this particular region, it would be important to assay if it responds to Emx2 levels. For example, the ATAC-seq peak containing the Emx2 binding region should be assayed in luciferase assays to see if it responds to co-transfection with an Emx2-expressing plasmid. Ideally, the Emx2 binding motifs could be mutated/deleted to further sustain that the enhancer capacity of this particular cis-regulatory element relies, at least partially, on Emx2 binding. Finally, this element (or its mouse ortholog) could be assayed for its enhancer activity in mouse transgenic assays, as the Emx2-Wnt5a regulatory relation appears to be evolutionary conserved in other tissue contexts, as the authors point out in the case of the brain. Finally, the regulatory relation between Wnt5a and Emx2 should be further supported to show their co-expression in the same cells of the developing

patagium (for example via HCR ISH or, alternatively, though scRNA-seq).

Minor points

- The total number of called ATAC-seq and H3K27ac ChIP-seq regions should be indicated in figure 1.
- Quantitation of Emx2 levels by qPCR was done from 4 biological replicates, but the figure shows five data points. Please clarify.
- As reported in the methods section, ChIP-seq experiments were performed in triplicates, but in the reporting summary only two replicates are mentioned for H3K27ac and only one sample for Emx2 ChIP-seq. Please clarify. A single replicate is still acceptable when processing difficult to obtain tissue from non-conventional species, as is the case here.

Referee #3 (Remarks to the Author):

This is an exciting manuscript investigating the genomic and developmental basis of the gliding membrane in gliding marsupials. Using the sugar glider as an emerging model organism, the authors show that Emx2 has a high number of patagium cis regulatory elements that are accelerated in different glider species. Emx2 is an attractive candidate as it is upregulated during patagium development and likely controls the expression of Wnt genes that were previously implicated in patagium development. Experiments demonstrate that knock-down of Emx2 in the developing patagium of the sugar glider reduces patagium size and results in downregulation of 420 development and other genes. Linking sequence acceleration to Emx2 expression, reporter assays with one GAR that contact the Emx2 promoter show that the glider sequence has higher activity than the sequence from a non-glider species. Finally, the authors show that manipulating Emx2 expression in the dermis of laboratory mice induces cellular and tissue phenotypes that are observed in patagium development, suggesting that gliders modified an existing developmental program.

Overall, the manuscript is exciting and makes an important contribution to understanding the origin of evolutionary novelty. However, a few key experiments are missing to support the claims made and I would like to see some data reanalyzed with more appropriate methods.

Comments:

Assigning cis regulatory elements to genes:

- i) It is unclear how exactly this was done. Line 527 describes that each cis element was assigned to the closest gene TSS, likely within the same TAD. However, figure 2 shows that many GARs are much closer to other genes than to the TSS of Emx2.
- ii) A TSS was determined for every annotated transcript (lines 522 -525). What happens if a gene has 2 or more transcripts with different TSSs? Did the authors later chose one of the transcripts per gene or did they test all transcripts? Does the number of 11044 (line 532) refer to genes or transcripts?
- iii) The binomial distribution models sampling with replacement. I believe that a hypergeometric distribution would be more appropriate to compute the probability that n of N cis elements assigned

to a gene are accelerated, as the set of elements is finite. The binomial distribution will approximate the hypergeometric one, but only for large numbers.

The method states that the authors compute a probability, but what is reported in Table S3 is the P-value, likely obtained summing the right tail of the distribution. This should be clarified.

Figure 3d: This plot shows the size deviation from a 1:1 ratio. I find this counterintuitive, also since the results report a 'significant size difference' (line 185) without mentioning the direction of change. Since the bar for shEmx2 is higher than the control bar, this indicates that Emx2 knockdown results in a larger patagium. I would suggest to plot the size of the injected site relative to the size of the uninjected site. One should then clearly see the size reduction.

line 197: 59 of 420 genes downregulated with shEmx2 overlap genes upregulated in the patagium primordium. What is the P-value for this observation. Is this more or less than expected?

line 217: I don't understand why Emx2 CHIP-seq data was overlapped with a 5 KB window around the TSS for the 59 genes. This is an oversimplification since cis regulatory elements can be located distal, as nicely demonstrated in this manuscript. Instead of focusing on the 5 KB window, the authors should overlap Emx2 bound sites with the ATAC+H3K27ac cis elements that are assigned to these 59 genes.

GARs:

- i) Alignments of the GARs assigned to Emx2 should be shown in the supplement.
- ii) The procedure to infer orthologous sequences for PhyloP analysis is unclear and not well justified. For example, it is not clear where the orthologous genes used as a starting point come from (line 467). Why were 1 KB flanks trimmed? Why requiring that 75% of the species have a 20 bp window with less than or equal to 5 gaps/Ns? This feels arbitrary and the reason is not clear. Instead, the authors should perform a whole genome alignment, obtain the sequences that align to the 52169 cis elements and use them for PhyloP.
- iii) The authors did not recover a single GAR that is accelerated in all 3 gliders, indicating that convergent evolution is rare. However, in the PhyloP analysis all 3 gliders were tested simultaneously for acceleration. This likely limits the power, as absence of acceleration in 2 gliders makes it harder to detect significant acceleration in the 3rd glider if all species are tested at once. How many GARs are detected if only one individual glider lineage is tested for acceleration while the other other 2 are removed from the alignment?
- iiii) Figure 4a convincingly shows that GAR16519 interacts with the Emx2 TSS. However, the yellow GAR that is even further downstream does not seem to have such a signal. Does this GAR and also the other GARs interact with Emx2? Together with additional reporter assays, this would be important to link accelerated evolution to Emx2.
- v) If these GARs are novel enhancers, they may have arisen either from nonfunctional sequence or new enhancer activity was incorporated into an existing element. If the former, the GAR sequence in other marsupials should evolve neutrally. If the latter, it is likely that the GAR sequence evolved

under purifying selection in other marsupials. Do any of the Emx2 GARs overlap conserved marsupial sequences?

A key conclusion in the manuscript is that patagia evolved many times in gliders by extending Emx2 expression in the patagium primordium.

However, using a reporter assay, enhancer activity was only demonstrated for one GAR that is accelerated in only one glider and extended Emx2 expression was shown only for the sugar glider. I understand that the difficulties of working with non-model organism, therefore testing for extended Emx2 expression in the other gliders is likely not feasible. However, using the sugar glider fibroblast cell line to test whether the GARs in other gliders also have higher enhancer activity should be feasible and this would be crucial to link "Regulatory convergence in the Emx2 locus to the independent evolution of mammalian gliding membranes", as the title states.

Minor:

line 99: 'high recovery of complete single-copy orthologs' is probably an overstatement. Extended Figure 1 shows that most assemblies have less than 75% complete BUSCO genes. The version of BUSCO used for this assessment is also not mentioned in the manuscript.

What is the unit for scaffold N50 in Table S1? Likely not bp.

The phylogenetic tree in Figure 1c was inferred with four-fold degenerate sites, which evolve faster. Even though the species are likely closely related, the topology should be confirmed with codon site 1 and 2.

PdgfraCreERT2: If the authors refer to <https://www.jax.org/strain/032770>, it should be PdgfraCreERT2

How many read pairs were sequenced for each ATAC-seq library?

What is the size distribution of the 52169 cis elements? Please add a size histogram to the supplement.

Figure 2c: Please label GAR 16519 in the figure.

This panel is almost identical with figure 4a, but why does the Micro-C contact map has a different intensity than Fig 2c, despite the same scale bar?

I find it intriguing that Emx2 is expressed in the mouse Wolffian at E11.5 but not at E13.5. Is there any evidence that in mouse and other non-gliding mammals and marsupials, Emx2 expression at E13.5 equivalent time points is actively repressed to suppress patagium development? If that is the case, gliders could have overcome this repression not only by evolving new enhancers but also by abolishing the activity of the repressor element.

Author Rebuttals to Initial Comments:

Referee #1

The study by Moreno et al. investigates the formation of the patagium, a type of gliding membrane that has evolved independently in three marsupial species from the Superfamily Petauroidea. The authors generate genome assemblies for 14 marsupial species, including 2 gliding and 8 non-gliding petauroid species and several outgroups, as well as an improved assembly for the sugar glider. By combining ATAC-seq and H3K27Ac ChIP-seq data from patagium primordium of the sugar glider, they annotate potential cis-regulatory elements (CREs) and subsequently perform comparative genomics analyses to identify orthologous regions across these species. By doing so, they identify Glider Accelerated Regions (GAR), which are those putative CREs that accumulate a substantial number of nucleotides substitutions in glider species, while being remarkably conserved in other species. By focusing on genes surrounded by numerous GARs, and also upregulated during patagium development as observed in RNA-seq data, they identify *Emx2* as a candidate gene for the evolution of this trait. The authors then perform several *in vivo* experiments in the sugar glider and in mouse, to validate their findings. Specifically, they inactivate *Emx2* expression by transgenic experiments during patagium formation, revealing that this gene is required for patterning and growth. By combining RNA-seq and ChIP-seq data, they provide evidence that *Emx2* may directly regulate key patterning genes, notably *Wnt5a*, a gene recently demonstrated by the group to be involved in patagium formation (Feigin et al, *Sci Adv*, 2023). By generating and analyzing Micro-C data from patagium, the authors focus on one particular GAR that display high interaction frequency with *Emx2*. This GAR is then validated as a putative enhancer by performing luciferase assays in cultured cells. Finally, the authors examine the pattern of expression of *Emx2* in the Wolffian ridge of mice, a region that shares anatomical similarities with the patagium of gliders. This analysis shows that *Emx2* expression is present at later developmental stages in gliders compared to mice, in which this occurs during the early prenatal period. By functionally mimicking this late expression in transgenic mice, the authors recapitulate certain features of patagium development, including an increase in epidermal thickness, as well as an increase in cell proliferation and mesenchymal cell density.

Overall, this study illustrates how novel approaches can shed light into the evolutionary origin of phenotypical traits. The enablement of the sugar glider as a transgenic animal model is highly original and innovative, thus providing the adequate means for functional validations. The manuscript is well written and easy to understand. Overall, the authors make a compelling case for the involvement of *Emx2* in patagium formation, as they similarly did for *Wnt5a*. However, the evidence supporting that the glider-specific *Emx2* expression results from regulatory convergence of CRE activity is still preliminary and limits the impact of the study.

Thank you for your feedback and insightful comments/suggestions. We have performed additional experiments, analyses, and have amended the text in response to your comments. Thanks to these, we believe our manuscript is now much stronger, and we hope

you find that our changes adequately address all the concerns raised. All changes made to the main manuscript and supplemental information appear in blue font.

Below please find a point-by-point response to your comments with details on which changes were made and where they were made.

Major comments:

1. The major claim of this manuscript, as reflected in the title, is that the patagium has evolved in several marsupial gliders through convergent regulatory evolution. Yet, the evidence to support this claim is simply a screening for accelerated regions. This is coupled with the functional validation of just one putative GAR in one species, which appears insufficient at all lights. What happens to the other GARs identified by the authors? Are they also functional? If the authors want to make such broad claims, they should provide evidence that changes in *Emx2*-related CRE activity have occurred in all the different glider species, not just in one.

This is an excellent point, which was also brought up by the other reviewers. To address this, we have now performed functional assays for all our identified GARs. We synthesized constructs corresponding to the different GARs as well as the orthologous sequences for the corresponding non-glider sister species. Altogether, we tested a total of 12 sequences. Out of the 6 *Emx2*-associated GARs identified via our comparative genomics analyses and Micro-C experiments, 3 of them led to strong luciferase reporter activity. Importantly, each of the three gliding species had one *Emx2*-associated GAR that displayed this pattern, supporting our argument that the *Emx2* locus has experienced patterns of accelerated evolution across all three gliding marsupials. These new experiments are now presented in Fig 2d, Extended Data Fig. 4, and Lines 167-205.

2. Regarding the assays to validate putative CRE activity, the authors perform these experiments in an immortalized dermal fibroblast cell line. I have trouble understanding why dorsal cells are a suitable model to investigate the activity of patagium CREs, simply because *Emx2* is not expressed or it is at very low levels in dorsal tissue. In fact, the dorsal tissue is employed by the authors as a negative control for *Emx2* expression in RNA-seq analysis.

Our immortalized line was generated from a large piece of trunk skin, which includes dorsal and lateral skin. We now realize the original description in the Methods was misleading. Our rationale for using trunk skin cells is that we wanted to have a broad representation of genes expressed in skin.

To further address this comment, we performed RNA-seq on our sugar glider dermal fibroblast line and analyzed the expression levels of *Emx2*, *Wnt5a* and other key transcription factors that we previously identified as being expressed in patagium tissue (e.g., *Tbx3*, *Tbx5*, *Hand3*, and *Osr1*; Feigin et al 2023). Our new analysis shows that our cell

line displays robust levels of *Emx2* expression, comparable to the other genes listed above. Moreover, similar to what was found with patagium tissue, we did not detect expression of genes like *Shh* or *Pax5* (Feigin et al 2023). Thus, while *in vitro* models will never fully recapitulate *in vivo* models, these new data suggest that our sugar glider dermal fibroblast cell line provides a relevant *in vitro* model to test hypotheses about the upstream and downstream regulation of *Emx2*.

The revised version of our manuscript contains a more accurate description of the skin region used to generate these cells. Moreover, we now include the new RNA-seq profiling experiment and analysis as an Extended Data figure. The changes can be found in Lines 816-817; 826-832; and Extended Data Fig. 10).

3. Furthermore, a luciferase assay in a cell line does not provide any spatial information on the activity of these putative CREs. The authors have taken a considerable effort in implementing a transgenesis for the sugar glider, a system that opens ample opportunities for experimental validation. Have the authors consider validating these putative CREs in vivo by injecting the sequence coupled with reporter genes in the developing patagium (i.e. transient transgenesis)? Classical reporter assays in transgenic mice could also provide important insights on the evolution of these CREs: they could reveal if they can be activated in other *Emx2*-expressing tissues, potentially being regulated by the same transcription factors (TFs).

We completely agree that a cell line does not provide the relevant spatial information for testing candidate CREs. To answer the Reviewer's question, we have attempted tests of enhancer activity directly in the sugar gliders using a variety of constructs and delivery approaches (e.g., injection and electroporation). Unfortunately, however, we have encountered several technical difficulties, which currently makes this approach unfeasible.

Following the suggestion from Reviewers #1 and #2, we have now performed a set of LacZ-based transgenic reporter assays. For these experiments, we cloned our different GARs into a LacZ reporter, injected each construct into the H11 locus of mouse zygotes, and screened embryos at E9.5 and E11.5, stages at which *Emx2* is active in the brain and lateral skin of laboratory mice. We found that none of these GARs drove LacZ activity in mouse embryos at the stages analyzed. To potentially explain these results, we carried out evolutionary conservation analyses and found that, while the cis-regulatory regions corresponding to our GARs are conserved across marsupials (including non-petauroid marsupials), they are not conserved in eutherians (e.g., lab mice or humans). Specifically, we took individual GARs and carried out UCSC blat and NCBI Blastn searches against the mouse and human genomes. Five out of the six GARs did not yield any hits, despite the fact that synteny among marsupials and eutherians is largely conserved, and only the GAR located on the *Emx2* promoter was found to be conserved outside of marsupials. Although the promoter GAR was conserved, our result suggests that the trans environment of mouse embryos does not produce the necessary regulators to activate expression.

We agree with the statement made by Reviewer #2, in reference to mouse transgenic reporter assays: *'these experiments are not conclusive if negative'*. Although laboratory mice have been used to assess the functional significance of candidate enhancer sequences from various vertebrate species, the outcome of these experiments depends on the degree of spatial and temporal conservation in the trans-acting environment. Consequently, these experiments may exhibit a bias towards cis-regulatory elements that are evolutionary conserved across species, as exemplified by the widely characterized ZRS limb enhancer¹.

Besides the lack of sequence conservation, inter-species differences in the timing of deployment of trans-factors that control *Emx2* expression may be an additional factor contributing to our results. Marsupial neonates are often compared to 10-12 day mouse embryos; however, marsupials have experienced major shifts in the timing of organ and tissue development, displaying a combination of both advanced and delayed features compared to mice⁶². Consequently, comparing the developmental timing of skin and other tissues/organs between marsupials and eutherians poses a challenge. While our analysis in sugar gliders was conducted during early postnatal stages, we tested whether GARs could drive reporter activity in E9.5 and E11.5 mouse embryos. These stages were chosen because they correspond to periods when endogenous *Emx2* expression is widespread in lateral skin, limbs, and forebrain in laboratory mice. Notably, *Emx2* expression in the skin of laboratory mice significantly decreases after E11.5 and remains consistently low throughout postnatal stages^{2,3}. Nevertheless, it remains possible that screening GARs at additional embryonic and postnatal stages could yield different results, even though we consider such an outcome unlikely.

The new version of our manuscript includes the new mouse transgenic experiments and evolutionary conservation analyses (Extended Data Fig. 8 and Lines 396-417).

4. In that respect, have the authors investigated if novel transcription factors binding sites have appeared in the GARs? Are there any common novel signatures across all *Emx2* GARs? Do these potential new binding sites correspond to TFs that are also expressed in patagium?

Thanks for this suggestion. We agree that this analysis is very useful for identifying potential upstream regulators of *Emx2* and further support our claims that *Emx2* is a critical component of a patagium specific gene regulatory network. Thus, in response to this comment, we investigated whether the functionally active *Emx2*-associated GARs (GAR 41701, GAR 16519, GAR 11730) exhibited shared transcription factor binding motifs. We identified an enrichment of four motifs within the GARs, compared to orthologous sequences from non-glider species. Among the transcription factors predicted to bind to these motifs, 13 were found to be upregulated in the sugar glider patagium. While none of these transcription factors are known upstream regulators of *Emx2*, it is plausible that novel regulatory interactions have evolved between some of these genes and the regulatory regions of *Emx2* in gliding species, potentially influencing the expression of this gene. The revised version of our manuscript includes these new results (Lines 196-205).

5. Besides individual gliding species, were all 3 gliding or combinations of 2 species used together to detect GARs? Could this be a reason for the low number of shared GARs?

To clarify, phyloP analysis (i.e., the method we use to identify GARs) was not performed on all 3 gliders simultaneously. It was performed on each glider species independently and the resulting list of GARs for each species were compared among the three glider species to determine which of those showed overlap. We thank this Reviewer as well as Reviewer #3 for bringing up this point because we realize this was not stated clearly in the initial version of our manuscript.

We have now amended the Main text as well as the Methods to clarify and explicitly state that tests were performed on one glider at time and resulting GARs were overlapped (Lines 125-130; 667-675).

6. The authors suggest that *Emx2* expression is prolonged in the sugar glider in comparison to mice, where is only expressed prenatally. However, they do not show any evidence of prenatal expression of *Emx2*. In situ hybridizations for *Emx2* in similar prenatal stages as in mouse (Figs 5a and b) would be recommended to support their hypothesis.

One of the difficulties of working with a new model species is that its reproductive physiology is completely unknown. While lab mice form copulatory plugs that allow the accurate staging of embryos, we do not have the ability of staging prenatal sugar glider embryos, despite our best efforts. The alternative strategy would be to euthanize multiple females that are thought to be pregnant until we obtain the desired prenatal stages. However, this approach is difficult to pursue for three related reasons: (1) sugar gliders reach sexual maturity after 1 full year; (2) it is not straightforward to obtain healthy animals from breeders; and (3) due to space constraints related to caging requirements (breeding pairs require considerably large cages), our colony size is relatively small (~10 breeding pairs). Thus, we have a scheme in which we have established long-term breeding pairs that continuously produce joeys used for our experiments. Euthanizing multiple females would mean losing the respective breeding pairs, which would be detrimental to all our other experiments. While we agree with the reviewer that assessing *Emx2* expression in sugar glider prenatal stages would be very informative, we don't think this would fundamentally change our conclusions (i.e., *Emx2* is expressed at early postnatal stages, prior to patagium outgrowth, and remains high for at least two weeks. In contrast, it is transiently expressed in prenatal lab mice and remains very lowly expressed in skin during late embryonic and postnatal stages). Another important consideration related to this comment is that, for reasons outlined above (point 3), it is quite difficult to establish what constitutes a similar/comparable prenatal stage between mouse and sugar gliders. We completely acknowledge it would be extremely interesting and valuable to perform an analysis of *Emx2* expression across the entire prenatal period of sugar gliders. However, for the reasons stated above, this analysis is currently difficult to do.

We hope the Reviewer can understand the challenges and limitations of working with entirely new mammalian model systems.

Minor comments

7. Besides those GARs within the *Emx2* TAD, have the authors evaluated if there are GARs also within the *Emx2* ChIP-seq peaks? Could those be associated with genes that are differentially expressed in patagium? This might reveal functional rewirings of the *Emx2* networks that might be relevant for patagium formation.

Thanks for this excellent suggestion. In response to this comment, we performed an analysis to determine whether any of the genes upregulated in the native patagium and downregulated in response to shRNA-mediated *Emx2* downregulation had assigned GARs that overlapped with *EMX2*-bound sites. We found that 8 genes fulfilled this criteria (*Adamts15*, *Cacna2d3*, *Cadm1*, *Cdc42ep4*, *Dok6*, *Emx2*, *Hmcn1*, *Plekhhg1*). Though none of these genes have been previously shown to interact with *Emx2*, it is possible that some of them have undergone glider-specific functional rewirings to regulate processes in patagium formation, such as muscle development (*Adamts15*, metalloprotease), cell migration (*Cdc42ep4*, pseudopodia formation), and cell division (*Hmcn1*, cleavage furrow maturation). We have incorporated the results from this analysis in the new version of the manuscript (Lines 280-290).

8. Are there any *Emx2* binding sites in the *Wnt5a* locus that are conserved across non-gliding or in other species?

To address this comment, we focused on the 8 candidate regulatory elements assigned to *Wnt5a* and which also contained overlapping *EMX2*-bound sites. Analysis of the 108 *Emx2* binding motifs found across all these candidate regulatory elements, revealed that none of these sites were uniquely conserved in gliding marsupials. In fact, for 7 of the 8 candidate regulatory elements, more than 70% of the *Emx2* motifs were conserved in at least 50% of all the species in our alignment. While the remaining candidate regulatory element had 40% of the *Emx2* motifs conserved in at least 50% of all the species, there wasn't any obvious distinction between gliding and non-gliding species. Altogether, we find that most *Emx2* binding sites in the *Wnt5a* locus were present in most of the species in our alignment, which further suggests that the interaction between *Emx2* and *Wnt5a* may be conserved. We have incorporated the results from this analysis in the new version of the manuscript (Lines 307-309).

9. Could there be evidence of conservation for the *Emx2*-*Wnt5a* axis in other species, or is it specific for glider patagium?

As highlighted in the original version of our manuscript, there is evidence that *Emx2* forms a regulatory loop with *Wnt5a* and other Wnt ligands that controls cell proliferation and promotes expansion of the cerebral cortex⁴. In *Emx*^{-/-} mutants, expression of *Wnt5a* is

considerably reduced, causing progenitors to exit the cell cycle prematurely. As a result, mutants have a marked reduction in the size of the occipital cortex and hippocampus.

We thank the Reviewer for bringing up this very interesting question because it prompted us to examine the relationship between *Emx2* and *Wnt5a* further: we obtained publicly available highly resolved spatial transcriptomics datasets from multiple stages of laboratory mouse embryos with near-single-cell resolution (i.e., Stereo-seq⁵) and examined whether *Emx2* and *Wnt5a* were co-expressed in other developing tissues. In agreement with the study mentioned above, analysis from the forebrain region of E14.5 embryos showed that *Emx2* and *Wnt5a* were co-expressed in the hippocampus. Moreover, we observed robust co-expression of *Emx2* and *Wnt5a* in the craniofacial mesenchyme of E14.5 embryo samples, particularly within the spatial domain originally annotated as “jaw and tooth”. We performed additional analyses that revealed the *Emx2-Wnt5a* axis likely coordinates osteoblast fate commitment in the embryonic craniofacial mesenchyme. Thus, the new data strongly suggest that the *Emx2-Wnt5a* axis is not exclusive to the sugar glider patagium. The results from our new experiments and analyses are presented in Extended Data Fig 7. and Lines 297-332.

9. Line105: it would be worth to inform the non-expert reader of the link between the ChIP-seq and ATAC-seq data with regards to identifying CREs.

Thanks for this suggestion. We have amended the main text to explain how ATAC-seq and ChIP-seq can be used as complementary approaches to identify CREs and have cited relevant studies. The new changes can be found in Lines 112-115.

10. Line 123 What do the authors refer to with “rapid” regulatory evolution? What is the time scale here?

We used the term ‘rapid’ to refer to candidate cis-regulatory regions displaying accelerated evolution, compared to those regions that did not. This sentence was meant to highlight the fact that the three glider species have a reduced set of overlapping GARs. In other words, each lineage has primarily evolved a (mostly) unique set of GARs. We have reworded this part for clarity (Lines 131-133).

In other instances of the text, we use the term ‘rapid’ to refer to the fact that enhancers have a higher turnover rate and evolve more quickly than other regions of the genome, such as promoters. We have made this explicit and have cited the study that reported this finding (Lines 136-137).

11. The Micro-C data from figure 4c looks very different from the data in Fig 2c. Maybe the authors could consider displaying the data with a different resolution or color scale, so that the 3D structures are more visible.

Thanks for this suggestion. Since the panels in Figure 4c and 2c were redundant, we have decided to remove the Micro-C data from panel 4c. Instead, in response to a suggestion from Reviewer #2, we now include Micro-C data for the *Wnt5a* region (Please see our comment to point 4 brought up by Reviewer #2 for a detailed explanation).

12. Line 299: It is inappropriate to refer to this putative CREs as enhancers, because just one of these elements was functionally tested.

We agree with this suggestion and have made the relevant changes throughout the manuscript.

Referee #2

In this manuscript, Moreno et al. sequenced and assembled several marsupial genomes and identified accelerated regions in glider versus non-glider species (GARs) as proxy for regulatory variation in the latter. To gain insight into the relevance of these alterations in patagium development in the sugar glider, the authors generated ATAC-seq and H3K27ac ChIP-seq datasets and integrated them with RNA-seq analyses (generated by the same group in a previous study; Feigin et al., 2023), leading to the identification of *Emx2* as a good candidate to have accumulated regulatory alterations causally related to the formation of the patagium. Through a series of elegant experiments, the authors show that *Emx2* is required in vivo for patagium development being *Wnt5a* (another gene relevant for patagium formation; Feigin et al. 2023) likely one of its direct transcriptional targets. Supported by gain of function experiments in the mouse, the authors hypothesize that the regulatory alterations in GARs identified in the *Emx2* locus may participate in maintaining postnatal *Emx2* expression in the flank dermis of the joey, leading to changes in cell proliferation and density that are required for patagium formation. This is a beautiful and original manuscript that provides compelling evidence that *Emx2* plays a fundamental role in the development of the patagium in the sugar glider. While I am positive about its relevance, there are a few aspects that should be taken into consideration in revising the manuscript:

Thank you for your overall positive assessment of our study and for taking the time to provide very helpful comments and suggestions. We hope you find the new changes to our manuscript satisfactory. All changes made to the main manuscript and supplemental information appear in blue font.

Below please find a point-by-point response to your comments with details on which changes were made and where they were made.

1) the authors place significant weight on regulatory convergence at the *Emx2* locus in different glider species. While the authors identify GARs in the *Emx2* genomic landscape of three species with patagium, all the functional genomics and in vivo data was generated on sugar gliders (*P. breviceps*), so the claim on regulatory convergence is essentially based on one luciferase assay using GAR16518 orthologous regions from two glider species (*P. breviceps* and *A. pigmaeus*). There is no evidence provided indicating that *P. volans* GARs are indeed cis-regulatory regions controlling *Emx2* levels in the patagium nor that *Emx2* plays a role in patagium development in this species. I do not see that the data supporting regulatory convergence is strong enough to make such a statement. This claim should be removed from the title and tuned-down throughout the text. Alternatively, additional evidence could be gathered, but it would minimally require RNA-seq from patagium primordia vs dorsal skin in other glider species and extensive characterization of their putative *Emx2* GARs in vitro and in vivo (eg. via reporter constructs).

We agree with this comment. As suggested by all Reviewers, we have performed new functional assays for all our identified GARs. Please see our response to a similar point brought up by Reviewer #1 (point 1). To summarize, we find that each of the three gliding species has one *Emx2*-associated GAR that drives strong luciferase reporter activity. These new experiments are now presented in Fig 2d, Extended Data Fig. 4, and Lines 167-205.

In addition, following the Reviewer's suggestion, we have modified the title of our manuscript and have amended the title and various portions of the text to tune-down the language and acknowledge the fact that *in vitro* tests provide important but not definitive evidence of enhancer activity. Similarly, we have added text to our discussion acknowledging the limitations of not having access to tissues from the other glider species (Lines 467-469).

Altogether, we believe that the changes introduced to the Title, Main text and Discussion, coupled to the new experiments, adequately reflect the conclusions obtained from our data.

2) In relation to the previous point, have the authors checked all *Emx2* GARs/orthologous sequences from all three gliding species in luciferase assays to evaluate if they also potentiate reporter transcription? While I do not think these experiments would suffice to provide compelling evidence for regulatory convergence, they would at least provide grounds for it to be discussed.

Yes, we have now tested all of our GARs using our *in vitro* system and that each of the three gliding species has one *Emx2*-associated GAR that drives strong luciferase reporter activity. These new experiments are now presented in Fig 2d, Extended Data Fig. 4, and Lines 167-205. We completely agree that while these *in vitro* experiments do not constitute irrefutable evidence for regulatory convergence, they do provide important insights into the function of these regulatory elements and grounds for discussion. In addition, as described above, we have made several changes throughout the text, as well as in our title, to tone down the language.

3) The authors show that the sugar glider distant GAR16518 enhancer contacts the *Emx2* promoter and is able to increase reporter expression *in vitro*, while the orthologous sequence from a non-glider species cannot. This region, which is not a GAR in *P. breviceps*, does have a minor but statistically significant effect on luciferase expression, lower than that of *A. pigmaeus*. Have the authors tested the activity of GAR16518 from any of these two species in reporter assays in transgenic mice? The luciferase experiments they report may indicate that this element potentiates the transcriptional activity of a basal promoter through yet-unidentified transcription factors present in sugar glider fibroblasts, but the working hypothesis is that the roles of *Emx2* in patagium development are, at least partially, driven by its heterochronic expression in the patagium primordia when compared to *Emx2* expression pattern in mouse embryos. Assaying the activity of this element in mice (eg. if they observe that glider enhancers can drive expression to the inter-limb region in mouse embryos) may help elucidating if this particular element contributes to shaping the spatio-temporal pattern of expression

of *Emx2* in the patagium primordia. This experiment, while not conclusive if negative (mouse trans-acting factors may simply not bind the glider enhancers), is far simpler than transgenesis in joeys using constructs to drive reporter expression to the patagium primordia, which is the other possibility.

This is a very good point and one that was also brought up by Reviewer #1. Following the suggestions, we have now tested the relevant GARs using mouse transgenic experiments. We do not find evidence they drive expression in laboratory mice and the lack of evolutionary conservation suggests this result is explained by inherent differences in the trans-acting environment between laboratory mice and marsupials and/or by the inability of mouse trans-acting factors to bind to glider cis-regulatory elements. Thus, we completely agree with this Reviewer that a negative result in a mouse transgenic experiment does not allow drawing conclusive results. Please see our response to Reviewer #1 (point 3) for a detailed response and a description of our new experiments. The new version of our manuscript includes the new mouse transgenic experiments and evolutionary conservation analyses (Extended Data Fig. 8 and Lines 396-417).

4) The regulatory relation between *Emx2* and *Wnt5a* should be further supported. It is surprising that the authors limit their analysis to 5Kb upstream and downstream of the TSS of their 59 candidate targets, as tissue-specific transcription factors frequently exert their activities through distant cis-regulatory regions (GAR16518 is itself a good example of this). Are there additional *Emx2*-binding regions in the *Wnt5a* genomic landscape?

This is an excellent point. In the new version of the manuscript, the analysis has been expanded to include the entire region for each of the 59 genes, defined by the corresponding TAD, rather than restricting it to a 5 kb region. The new changes are reflected in Lines 280-290.

Moreover, prompted by this and the following set of comments, we have now added a new Results section (and corresponding main figure) titled '*Emx2* directly regulates *Wnt5a*'. In this new section, we present the TAD containing *Wnt5a* as well as the ATAC/ChIP (H3k27Ac) peaks that were assigned to *Wnt5a* in our analysis. We now include the EMX2-ChIP peaks and highlight those that overlap with ATAC/ChIP (H3k27Ac) peaks. In addition, as detailed below, we have performed new experiments/assays to strengthen our claims that *Emx2* and *Wnt5a* directly interact. These new results are now included in a new main Figure (Fig. 4) and Lines 297-332.

5) Are there any distant interactions that can be predicted from the Micro-C dataset, as done for GAR16518? This data should be also included, as well as the ChIP-seq binding profile of *Emx2* in the *Wnt5a* locus, with particular focus on the peak identified in Fig. 3j.

We reanalyzed our data and did not find evidence of interactions between *Emx2* and any of the distant putative cis-regulatory elements we identified. This is now explicitly stated in the

text (Lines 1360-1361). In addition, as suggested by the Reviewer, we now include the ChIP-seq binding profile of the *Wnt5a* locus in a new main figure (Fig. 4).

6) Concerning this particular region, it would be important to assay if it responds to *Emx2* levels. For example, the ATAC-seq peak containing the *Emx2* binding region should be assayed in luciferase assays to see if it responds to co-transfection with an *Emx2*-expressing plasmid. Ideally, the *Emx2* binding motifs could be mutated/deleted to further sustain that the enhancer capacity of this particular cis-regulatory element relies, at least partially, on *Emx2* binding.

This is an excellent suggestion, thank you. We have now performed a set of luciferase assays to test the ability of the *Wnt5a* region to respond to *Emx2*. Co-transfection of a plasmid carrying the *Wnt5a* promoter and a plasmid expressing *Emx2* led to a marked increase in reporter activity, compared to co-transfection with a control plasmid expressing GFP. Moreover, co-transfection of *Emx2* and a reporter plasmid in which we mutated the *Emx2* binding motifs led to a decrease in reporter activity. Together, these experiments demonstrate that the region upstream of *Wnt5a* responds to *Emx2* and that its activity depends on *Emx2* binding. This considerably strengthens our claim that *Emx2* binds directly to *Wnt5a*. These new experiments have now been incorporated in the revised version of our manuscript (Lines 297-332; Figure 4).

7) Finally, this element (or its mouse ortholog) could be assayed for its enhancer activity in mouse transgenic assays, as the *Emx2*-*Wnt5a* regulatory relation appears to be evolutionary conserved in other tissue contexts, as the authors point out in the case of the brain.

Following this advice, we have performed LacZ-based transgenic reporter assays in laboratory mice and find that this sequence does not drive reporter activity, when assayed at E9.5 and E11.5. Similar to most of our GARs, we were unable to find a high-confidence orthologous match between this element and laboratory mouse/human sequences, while we did find a match in the opossum genome. This demonstrates that this element is conserved across marsupials but not conserved in eutherians.

8) Finally, the regulatory relation between *Wnt5a* and *Emx2* should be further supported to show their co-expression in the same cells of the developing patagium (for example via HCR ISH or, alternatively, through scRNA-seq).

Excellent suggestion. The new section of our manuscript now includes an image showing the co-expression of both genes in the same cells of the developing patagium. The new data is presented in Fig. 4a.

Minor points

9) The total number of called ATAC-seq and H3K27ac ChIP-seq regions should be indicated in figure 1.

This has been added to Figure 1.

10) Quantitation of Emx2 levels by qPCR was done from 4 biological replicates, but the figure shows five data points. Please clarify.

Thanks for catching this mistake. We used 5 biological replicates for our qPCR quantification and 5 biological replicates for RNA-seq libraries. The numbers were incorrectly stated in the Methods section. We have now included the correct numbers (Line 931).

11) As reported in the methods section, ChIP-seq experiments were performed in triplicates, but in the reporting summary only two replicates are mentioned for H3K27ac and only one sample for Emx2 ChIP-seq. Please clarify. A single replicate is still acceptable when processing difficult to obtain tissue from non-conventional species, as is the case here.

We acknowledge that our description was confusing and contained insufficient details. The word 'triplicate' refers to the number of ChIP assays (i.e., pull-down reactions) that were done for each experimental sample, whereas the word 'replicates' refers to the number of libraries that were generated for each experiment.

In the new version of our manuscript, we have modified the relevant section in the Methods to clarify this aspect and provide more details on the way our ChIP experiments were carried out. The changes are reflected in Lines 593-602.

Referee #3 (Remarks to the Author):

This is an exciting manuscript investigating the genomic and developmental basis of the gliding membrane in gliding marsupials. Using the sugar glider as an emerging model organism, the authors show that *Emx2* has a high number of patagium cis regulatory elements that are accelerated in different glider species. *Emx2* is an attractive candidate as it is upregulated during patagium development and likely controls the expression of Wnt genes that were previously implicated in patagium development. Experiments demonstrate that knock-down of *Emx2* in the developing patagium of the sugar glider reduces patagium size and results in downregulation of 420 development and other genes. Linking sequence acceleration to *Emx2* expression, reporter assays with one GAR that contact the *Emx2* promoter show that the glider sequence has higher activity than the sequence from a non-glider species. Finally, the authors show that manipulating *Emx2* expression in the dermis of laboratory mice induces cellular and tissue phenotypes that are observed in patagium development, suggesting that gliders modified an existing developmental program.

Overall, the manuscript is exciting and makes an important contribution to understanding the origin of evolutionary novelty. However, a few key experiments are missing to support the claims made and I would like to see some data reanalyzed with more appropriate methods.

Thank you for your overall positive assessment and your relevant and helpful suggestions. Following your comments, we have carried out additional experiments and reanalyzed portions of our data. All changes made to the main manuscript and supplemental information appear in blue font.

Below please find a point-by-point response to your comments with details on which changes were made and where they were made.

Comments:

Assigning cis regulatory elements to genes:

1) It is unclear how exactly this was done. Line 527 describes that each cis element was assigned to the closest gene TSS, likely within the same TAD. However, figure 2 shows that many GARs are much closer to other genes than to the TSS of *Emx2*.

Thanks for pointing out this was not clear. The reason why some GARs look closer to other genes than to *Emx2* is explained by how the TSS were predicted for each of the genes. The TSS for each gene was determined based on the location of the first annotated exon. If this exon did not start with an ATG, it was considered the 5' untranslated region (5'UTR), and the TSS was annotated as the site 1 base pair directly upstream of the first exon. For exons that did begin with an ATG codon, the TSS was estimated to be approximately 1 kb upstream from the translation start site. To assign candidate cis-regulatory elements (CREs) to the

TSSs of genes, we use the tool "bedtools closest". This tool selects which ATACseq peaks are closest to a TSS based on distance and regardless of directionality. While this method may not be perfect, it provides the best approximation when constrained by the topologically associated domain data used in the study. We realize this was not clearly stated in our original manuscript. We have amended the Methods section to clarify this (Lines 692-716).

2) A TSS was determined for every annotated transcript (lines 522-525). What happens if a gene has 2 or more transcripts with different TSSs? Did the authors later chose one of the transcripts per gene or did they test all transcripts? Does the number of 11044 (line 532) refer to genes or transcripts?

For all genes we grouped all transcripts, sorted them by length, and selected the largest transcript. In other words, we used the longest transcript as a representative for each gene. In this context, "genes" and "transcripts" are considered the same (i.e., the number 11044 refers to both genes and transcripts). Thanks for bringing this up because it made us realize our original statement was imprecise. We have amended the Methods section to clarify this (Lines 692-716).

3) The binomial distribution models sampling with replacement. I believe that a hypergeometric distribution would be more appropriate to compute the probability that n of N cis elements assigned to a gene are accelerated, as the set of elements is finite. The binomial distribution will approximate the hypergeometric one, but only for large numbers. The method states that the authors compute a probability, but what is reported in Table S3 is the P-value, likely obtained summing the right tail of the distribution. This should be clarified.

We agree with this comment. Following the suggestion from the Reviewer, we have reanalyzed our enrichment data using a hypergeometric distribution. Crucially, *Emx2* still comes up as our top candidate and is the only gene that has associated GARs in all three gliding species. The new version of our manuscript presents the new analysis using the hypergeometric distribution (Lines 147-149; 709-716).

4) Figure 3d: This plot shows the size deviation from a 1:1 ratio. I find this counterintuitive, also since the results report a 'significant size difference' (line 185) without mentioning the direction of change. Since the bar for shEmx2 is higher than the control bar, this indicates that Emx2 knockdown results in a larger patagium. I would suggest to plot the size of the injected site relative to the size of the uninjected site. One should then clearly see the size reduction.

Following the suggestion from the Reviewer, we now display the size of the injected side relative to the uninjected site. We agree this is a clearer way of showing the direction of the change. We have also amended the text to specify the direction of the change. The new changes can be found in Fig. 3f and Lines 245-246.

5) line 197: 59 of 420 genes downregulated with shEmx2 overlap genes upregulated in the patagium primordium. What is the P-value for this observation. Is this more or less than expected?

This result is higher than expected by chance. In the new version of the manuscript, we now include the corresponding P-value for this and other observations (Lines 260-261; 282-283).

6) line 217: I don't understand why Emx2 CHIP-seq data was overlapped with a 5 KB window around the TSS for the 59 genes. This is an oversimplification since cis regulatory elements can be located distal, as nicely demonstrated in this manuscript. Instead of focusing on the 5 KB window, the authors should overlap Emx2 bound sites with the ATAC+H3K27ac cis elements that are assigned to these 59 genes.

This is a good point, and one that was also brought up by Reviewer #2. In the new version of the manuscript, rather than restricting it to a 5 kb region, we have expanded the analysis to include the entire region defined by the corresponding TAD for each of the 59 genes. The new data is presented in Lines 280-290)

GARs:

7) Alignments of the GARs assigned to Emx2 should be shown in the supplement.

We agree with this suggestion and have now included a new Supplemental Data file with this information. Please see Supplemental Data File 1.

8) The procedure to infer orthologous sequences for PhyloP analysis is unclear and not well justified. For example, it is not clear where the orthologous genes used as a starting point come from (line 467). Why were 1 KB flanks trimmed? Why requiring that 75% of the species have a 20 bp window with less than or equal to 5 gaps/Ns? This feels arbitrary and the reason is not clear.

We thank the reviewer for bringing this up; we acknowledge that the methods were not clearly stated and some of our rationale was not explicitly laid out. We address the specific points you raise here and have amended the text in the revised manuscript. The changes can be found in Lines 612-675).

Source of orthologous genes: To briefly summarize how we identified orthologous genes across species, we started from the high-quality, transcriptome-based RefSeq annotation of the Koala (our primary outgroup species), and we lifted over gene models to each marsupial genomes via LiftOff. This method is exceptionally efficient in terms of computational time/resources and performed very well across the dataset. This mirrors the approach used in one of our recent comparative genomics papers⁶.

Explanation of flank sequences: The nature and purpose of the flank sequences used during the annotation of putative enhancer orthologs (**line 463 in the original text**) was not

described clearly enough in our original text. First, putative cis-regulatory elements (CREs) in the sugar glider were identified using chromatin data (transposase accessibility and the H3k27Ac histone marks) and their orthologs across species were annotated by lift-over with LiftOff. The flank sequence represents the genomic sequence in the sugar glider genome immediately outside the boundaries of the putative cis-regulatory elements (i.e. flank sequences are not supported as being active regulatory sequence by our chromatin state data). The flank sequences were temporarily added to each sugar glider element to increase their mappability and mapping specificity between the sugar glider and the other marsupial genomes in our study. In other words, increasing the length of the mapped sequence improves ortholog recovery and reduces the frequency of multi-mapping which could lead to ambiguous orthology inferences. This is a common strategy in many comparative annotation pipelines similar to our own (e.g., the Phyluce pipeline for cross-species annotation and analysis of ultraconserved elements)⁷.

This flank sequence, however, is not actually considered to be part of the CRE based on our ATAC/ChIP data. Therefore, when we perform our phyloP analyses of evolutionary rates on putative CRE orthologs, we do not want these presumptively non-functional flank sequences to be included in our alignments because they should experience a different regime of selection than the candidate functional elements that they flank. Given that these sequences are not relevant to our analysis and only served to improve mappability during element annotation, we trimmed the elements away before performing phyloP analysis.

Trimming/filtering parameters: The motivation for our sliding window approach for end-trimming our alignments is that gaps are often introduced due to evolutionary sequence divergence between species (e.g. indels) or from local assembly gaps, and that we observed that gappiness tended to be focused on alignment ends. Differences in sequence lengths mean that flank removal from species other than the sugar glider isn't as simple as clipping 500bp from each side. Additionally, it was prudent to remove gappy alignment ends to avoid them impacting our evolutionary analysis (phyloP) which is based on nucleotide substitution rates.

In our original methods section, we missed the detail that, in each ortholog alignment, we also included a copy of the sugar glider CRE without flanks in order to assist in flank trimming. Briefly, we scanned across each alignment to locate where each flankless sugar glider CRE sequence began and then used this position to remove flank from the other species. We then used the 20bp sliding window to polish the ends of the remaining alignments, removing extremely gappy regions (i.e., those in which a strong majority, 75%, had a gap percentage >25% within the window).

With regards to the specific values used in this sliding window, we admit that there is a degree of arbitrariness. The 20bp size of the window is comparable to that used to quality trim other types of sequence data (e.g., sliding window quality trimming with trimmomatic which has a default of 15bp for reads with lengths in the same order of scale as our elements). In terms of the number of species driving gap percentage, we used a similar value (75%) to that used by Sackton et al 2018 (70%) to trim alignment columns of orthologous CREs among ratite bird CRE orthologs, which have a fairly comparable evolutionary divergence. We sought to trim to only windows where a majority of species (75% as above) had a majority of columns being non-gap (75%). In a 20bp window, this equates to no more than 5bp of gap sequence. These specific points of reference from

similar literature and common tools pointed us to a specific range of values. We ultimately went with a slightly larger window length due to our CREs being longer than a typical sequencing read and decided to be slightly stricter about gap percentages in our dataset than Sackton et al, as our most distantly related species are slightly closer than the most distantly related ratites are to each other.

9. Instead, the authors should perform a whole genome alignment, obtain the sequences that align to the 52169 cis elements and use them for PhyloP.

While we are enthusiastic about gradual improvements in the performance and resource requirements of whole genome alignment tools/workflows, we believe that the approach we have taken here is appropriate for the goals of our study, and that whole genome alignments are unlikely to produce significant benefits to the core questions of our manuscript.

Our approach can be essentially summarized as 1) identify/annotate elements of interest in a focal species 2) identify/annotate orthologs in other relevant species using sequence similarity and synteny anchoring 3) extract, align, filter orthologous sequences 4) perform evolutionary analyses.

Workflows based on annotation, orthology inference and local alignment are common in the field, particularly for analyses that are focused on a small subset of elements, across a large number of genomes^{8,9} and our approach is similar to that of pipelines built for identification and analysis of non-coding elements such as Phyluce⁷. Particularly for mammals, it is more computationally efficient to extract and process elements of interest than it is to align whole genomes. Additionally, there is not currently strong evidence that whole genome alignments will outperform what we have done here. For instance, while the Zoonomia placental mammal alignment is a great new resource, overall alignability between human (the best-quality assembly and the reference for the alignment) and more distantly-related species is not exceptionally high. Overall, the most likely outcome for generating marsupial whole genome alignments would be to approximately match the performance of the more common approach we have used here.

In the future, we and other groups that use marsupial model species are interested in the idea of producing Zoonomia-like marsupial alignments, particularly as whole genome alignment tools improve in performance and efficiency, but this is a significant task beyond the scope and conceptual focus of the present study.

10) The authors did not recover a single GAR that is accelerated in all 3 gliders, indicating that convergent evolution is rare. However, in the PhyloP analysis all 3 gliders were tested simultaneously for acceleration. This likely limits the power, as absence of acceleration in 2 gliders makes it harder to detect significant acceleration in the 3rd glider if all species are tested at once. How many GARs are detected if only one individual glider lineage is tested for acceleration while the other other 2 are removed

from the alignment?

Reviewer # 1 brought up a very similar point, which made us realize our original description of this analysis was not very clear. To clarify, phyloP analysis was not performed on all 3 gliders simultaneously. It was performed on each glider species and the resulting lists of accelerated elements were compared to each other.

We have now amended the Main text as well as the Methods to clarify and explicitly state that tests were performed on one glider at a time and resulting GARs were overlapped. The changes are reflected in Lines 125-130; 670-675.

11) Figure 4a convincingly shows that GAR16519 interacts with the *Emx2* TSS. However, the yellow GAR that is even further downstream does not seem to have such a signal. Does this GAR and also the other GARs interact with *Emx2*? Together with additional reporter assays, this would be important to link accelerated evolution to *Emx2*.

GAR16519 and the other GAR that the Reviewer is referring to are located distantly from the *Emx2* TSS and were assigned to a different gene. The reason we tested GAR16519 is because there was a clear loop linking it to the *Emx2* promoter. However, we did not have such evidence for the GAR that is even further downstream of *Emx2*. While this GAR falls under the same TAD as *Emx2*, it is much closer to another gene than to *Emx2*, so it was not assigned to *Emx2* in our enrichment analysis. To avoid confusion, Figure 2 now contains only *Emx2*-associated TADs (i.e., those assigned to *Emx2* by our analysis + the one for which we identified the long-range loop).

12) If these GARs are novel enhancers, they may have arisen either from nonfunctional sequence or new enhancer activity was incorporated into an existing element. If the former, the GAR sequence in other marsupials should evolve neutrally. If the latter, it is likely that the GAR sequence evolved under purifying selection in other marsupials. Do any of the *Emx2* GARs overlap conserved marsupial sequences?

We agree that analyzing the origin of these sequences can yield relevant insights. To explore whether *Emx2*-associated GARs represent novel enhancers or mammalian/marsupial conserved regulatory elements, we assessed conservation using all the genomes used in our study. Briefly, for a ~20 kb region surrounding each GAR, we generated conservation scores using phyloP. We found that many of these elements show high conservation while others show less. Interestingly, however, we found that all *Emx2*-associated GARs could be identified in *Monodelphis domestica* (a non-placental marsupial), in the same syntenic position, but only the GAR overlapping the *Emx2* promoter was found in the mouse and human genome. These results indicate that most of the elements we uncovered in our study are conserved across marsupials but are not conserved in eutherians. These new analyses are now presented in Extended Data Fig. 8 and Lines 396-417.

13) A key conclusion in the manuscript is that patagia evolved many times in gliders by extending *Emx2* expression in the patagium primordium. However, using a reporter assay, enhancer activity was only demonstrated for one GAR that is accelerated in only one glider and extended *Emx2* expression was shown only for the sugar glider. I understand that the difficulties of working with non-model organism, therefore testing for extended *Emx2* expression in the other gliders is likely not feasible. However, using the sugar glider fibroblast cell line to test whether the GARs in other gliders also have higher enhancer activity should be feasible and this would be crucial to link "Regulatory convergence in the *Emx2* locus to the independent evolution of mammalian gliding membranes", as the title states.

We completely agree with this comment and it was a point brought by the other two Reviewers. In response, we have performed additional luciferase assays to test the function of all the GARs identified by comparative genomics. To summarize, we find that each of the three gliding species has one *Emx2*-associated GAR that drives strong luciferase reporter activity. In addition, we have changed the title and tuned-down the language of the manuscript to acknowledge the fact that *in vitro* tests provide important but not definitive evidence of enhancer activity. These new experiments are now presented in Fig 2d, Extended Data Fig. 4, and Lines 167-205.

Minor:

14) line 99: 'high recovery of complete single-copy orthologs' is probably an overstatement. Extended Figure 1 shows that most assemblies have less than 75% complete BUSCO genes. The version of BUSCO used for this assessment is also not mentioned in the manuscript.

We agree with this comment and have now modified the text in response. Moreover, we have added the BUSCO version used in our analysis. Changes are reflected in Lines 100-102; 547-549).

15) What is the unit for scaffold N50 in Table S1? Likely not bp.

We would like to clarify that we used N50 to refer to the smallest number of scaffolds that make up half of the genome assembly. We have updated our table and now report N50 as the largest length such that half of the genome assembly (nucleotides) are contained in scaffolds/contigs of at least that size. We now report L50 as the statistic representing the smallest number of scaffolds/contigs which make up half of the assembly. Our original error likely stemmed from the fact that these terms are used differently in the literature. We have amended the Table legend to clarify what we refer to (Supplementary Table 1).

16) The phylogenetic tree in Figure 1c was inferred with four-fold degenerate sites, which evolve faster. Even though the species are likely closely related, the topology should be confirmed with codon site 1 and 2.

We have performed the new analysis suggested by the Reviewer. We find that the topology inferred from codon sites 1 and 2 exactly matches the original topology inferred using four-fold degenerate sites. This new result is presented in Extended Data figure 9a.

17) PdfgraCreERT2: If the authors refer to <https://www.jax.org/strain/032770>, it should be PdfgraCreERT2

Thanks for pointing this out. We did use the mouse strain you allude to and have now corrected this mistake. Please see Line 1036.

18) How many read pairs were sequenced for each ATAC-seq library?

We generated a total of 62,387,513 and 59,728,230 million reads per ATAC-seq library. We have now added this information to the methods (Lines 575-576).

19) What is the size distribution of the 52169 cis elements? Please add a size histogram to the supplement.

The new version of our manuscript now includes a new figure showing the size distribution of all our elements. Please see new Extended Data Figure 9b.

20) Figure 2c: Please label GAR 16519 in the figure.

We have now labeled all GARs in the figure.

21) This panel is almost identical with figure 4a, but why does the Micro-C contact map has a different intensity than Fig 2c, despite the same scale bar?

Thanks for helping us realize that this was confusing. The reason why the two contact maps have different intensities is because the scales of the displayed regions were different. Since the panels in Figure 4c and 2c were redundant, as pointed out by this Reviewer and by Reviewer #1, we have decided to remove the Micro-C data from panel 4c. We now include a second Micro-C plot displaying the contact maps for the TAD containing the *Wnt5a* locus.

21) I find it intriguing that *Emx2* is expressed in the mouse Wolffian at E11.5 but not at E13.5. Is there any evidence that in mouse and other non-gliding mammals and marsupials, *Emx2* expression at E13.5 equivalent time points is actively repressed to suppress patagium development? If that is the case, gliders could have overcome this repression not only by evolving new enhancers but also by abolishing the activity of the repressor element.

We agree with the Reviewer that this is an interesting possibility. We are not aware of any evidence suggesting that *Emx2* is being actively repressed in mice or other non-gliding

species. To gain insights into this question, however, we made use of our existing skin single-cell RNA-seq data from different embryonic timepoints in lab mouse (E12-E15)¹⁰ and asked whether any of the known *Emx2* repressors (i.e., those confirmed through experimental data) were expressed in an anti-correlated way with *Emx2*.

Figure 1. Temporal relationship between the expression of *Emx2* and its known repressors. a, Cell types recovered from a scRNA-seq dataset of E12-E15 laboratory mouse skin (left). Expression of *Emx2* is restricted to dermal fibroblasts (right). **b,** Dermal fibroblast-specific expression through time of *Emx2* and genes experimentally known to repress *Emx2*. Shown are expression levels for the 11 dermal fibroblast clusters identified in our data.

First, we integrated data from all four developmental stages and confirmed that *Emx2* was expressed exclusively in dermal fibroblasts (Fig. 1a). Then, by focusing only on the dermal fibroblasts, we reclustered the data and examined the expression of *Emx2* through time. Among dermal fibroblasts, *Emx2* was mainly expressed in clusters 1, 2, 6, 8, and 11 (Fig. 1b). Additionally, our longitudinal data shows that *Emx2* expression is high at E12.5 and decreases by E15.5 (Fig. 1b), which is a pattern that confirms what we found using whole mount in situ hybridizations. We next plotted the expression of genes known to repress *Emx2* in the context of other mammalian tissues, including *Sp8* (forebrain)¹¹, *Pax6* (forebrain)¹², *Foxg1* (forebrain)¹², *Noggin* (forebrain)¹², *HoxA10* (Urogenital tract)¹³, and *Fgf8*

(telencephalic and optic vesicles)¹⁴. Our analysis indicated that none of the known *Emx2* repressors are co-expressed in the same dermal fibroblast clusters as *Emx2* (Fig. 1c). Moreover, the expression of all known repressors is very low and none of them show increased expression through time (Fig. 1b). Thus, while our analysis does not rule out that other repressors may be actively down regulating *Emx2* in mouse skin, it suggests that none of the established ones have this effect.

Given that definitively addressing this point falls outside the scope of our study, we prefer not to include this analysis in the manuscript. However, in response to this comment, we have modified the discussion of our manuscript to acknowledge this interesting point.

Specifically, instead of:

'the evolution of marsupial patagia was facilitated by lineage-specific increases in the levels and duration of *Emx2* expression' we now say:

'the evolution of marsupial patagia was facilitated by lineage-specific changes in the levels and duration of *Emx2* expression'.

We believe this change encompasses the possibility that *Emx2* is de-repressed in gliders, relative to non-gliders (Lines XX).

References

1. Kvon, E. Z. *et al.* Progressive Loss of Function in a Limb Enhancer during Snake Evolution. *Cell* **167**, 633-642.e11 (2016).
2. Rezza, A. *et al.* Signaling Networks among Stem Cell Precursors, Transit-Amplifying Progenitors, and their Niche in Developing Hair Follicles. *Cell Rep.* **14**, 3001–3018 (2016).
3. Sennett, R. *et al.* An Integrated Transcriptome Atlas of Embryonic Hair Follicle Progenitors, Their Niche, and the Developing Skin. *Dev. Cell* **34**, 577–591 (2015).
4. Muzio, L., Soria, J. M., Pannese, M., Piccolo, S. & Mallamaci, A. A mutually stimulating loop involving *emx2* and canonical *wnt* signalling specifically promotes expansion of occipital cortex and hippocampus. *Cereb. Cortex* **15**, 2021–2028 (2005).
5. Chen, A. *et al.* Spatiotemporal transcriptomic atlas of mouse organogenesis using DNA nanoball-patterned arrays. *Cell* **185**, 1777-1792.e21 (2022).
6. Richardson, R. *et al.* The genomic basis of temporal niche evolution in a diurnal rodent. *Curr. Biol.* (2023) doi:10.1016/j.cub.2023.06.068.
7. Faircloth, B. C. PHYLUCE is a software package for the analysis of conserved genomic loci. *Bioinformatics* **32**, 786–788 (2016).
8. Kolora, S. R. R. *et al.* Origins and evolution of extreme life span in Pacific Ocean rockfishes. *Science* **374**, 842–847 (2021).
9. Tong, C., Avilés, L., Rayor, L. S., Mikheyev, A. S. & Linksvayer, T. A. Genomic signatures of recent convergent transitions to social life in spiders. *Nat. Commun.* **13**, 1–12 (2022).
10. Johnson, M. R. *et al.* *Sfrp2* is a multifunctional regulator of rodent coat patterns. *Nature Ecology and Evolution* doi:10.1101/2022.12.12.520043.

11. Zembrzycki, A., Griesel, G., Stoykova, A. & Mansouri, A. Genetic interplay between the transcription factors Sp8 and Emx2 in the patterning of the forebrain. *Neural Dev.* **2**, 8 (2007).
12. Mallamaci, A. & Stoykova, A. Gene networks controlling early cerebral cortex arealization. *Eur. J. Neurosci.* **23**, 847–856 (2006).
13. Troy, P. J., Daftary, G. S., Bagot, C. N. & Taylor, H. S. Transcriptional repression of peri-implantation EMX2 expression in mammalian reproduction by HOXA10. *Mol. Cell. Biol.* **23**, 1–13 (2003).
14. Crossley, P. H., Martinez, S., Ohkubo, Y. & Rubenstein, J. L. Coordinate expression of Fgf8, Otx2, Bmp4, and Shh in the rostral prosencephalon during development of the telencephalic and optic vesicles. *Neuroscience* **108**, 183–206 (2001).

Reviewer Reports on the First Revision:

Referees' comments:

Referee #1 (Remarks to the Author):

The authors have successfully addressed all my concerns.

I would like to highlight their commendable efforts in making their study more robust, in particular by performing additional assays for regulatory activity in two different species, among others experiments.

Despite some non-conclusive results (i.e. LacZ in mouse transgenics), the overall experimental evidence makes a compelling case for the involvement of Emx2 in the evolution of the patagium in gliders.

Referee #3 (Remarks to the Author):

The authors have done a great job in revising the manuscript and addressing the open issues. Regarding the new analyses that were added in the revision, I have two points. None of them will affect the main conclusions of this study, but it would be important to address them before publication.

1)

Blat is a method that suitable for detecting highly similar sequences, which is not the case for comparisons between marsupials, mice and humans. The authors mentioned that they also used NCBI's blastn, which should have a higher sensitivity if the right parameters are used. However, the respective methods section only mentions Blat and not blastn. This methods section needs to also describe the blastn analysis.

2)

Lines 274-290:

The authors tested whether the TADs of the 59 genes contained at least one enhancer bound by Emx2, which would indicate direct regulation by Emx2.

For 38 of the 59 genes this was the case and a hypergeometric test showed that this is significantly more than expected.

I suppose the hypergeometric test was computed considering all 11044 TADs?

In other words, 39 of 59 TADs for the differentially expressed genes vs. n of 11044-59 (N) TADs.

If the 59 TADs are much longer than the 11044-59 TADs, this would create a bias since longer TADs will have more enhancers.

It would be necessary to correct for the length of the TAD or showing that the 59 TADs have a similar or shorter size.

An alternative test would be similar to what GREAT <https://www.nature.com/articles/nbt.1630> does.

One can consider the 59 genes as genes having a certain GO term and use a binomial test over the enhancers to test whether the enhancers in the 59 TADs are significantly more often bound by Emx2.

The success probability here is the proportion of Emx2 bound enhancers of all enhancers.

Minor point

I think the main point of Figure 2d is to assess whether the glider GAR has a different (higher or lower) activity compared to the non glider GAR. It would then be helpful to also test significance by comparing the glider vs. non glider activity. For the first two GARs, this is most likely also significant, as the data for the non glider GARs looks like the control.

Referee #4 (Remarks to the Author):

In this work, the authors present an interesting and thorough study on the independent emergence of an evolutionary novelty, the gliding patagium, in three species of gliding marsupials, which constitutes a compelling example of evolutionary convergence. They employ a set of techniques including comparison of different genomes to locate species-specific accelerated regions (for which they have sequenced and assembled 15 marsupial genomes), epigenomic and transcriptomic assays, and functional studies such as transgenics in one of the gliding marsupial species, as well as gain and loss-of-function assays in this species and in mice. Based on the data obtained from these experiments, the authors highlight the crucial role of the Emx2 gene in the developing of the patagium in these species, supporting the hypothesis that this gene has been independently upregulated through species-specific modification of its transcriptional regulation.

Overall, I think this is a very original manuscript, well written and structured and very comprehensive. The experiments are innovative and appropriate, and strongly support the proposed hypotheses. It is worth noting the use of a non-model species for some of these experiments, which poses a significant added challenge. Besides, these 15 new marsupial genome assemblies constitute a very important resource for future comparative genomics studies.

The work of the three reviewers in general, and in particular that of reviewer 2, has been meticulous and has significantly contributed to the improvement of the manuscript. In my opinion, the authors have satisfactorily addressed the questions and suggestions raised by the reviewer. However, the results of the transgenesis experiments in mice suggested by reviewers 1 and 2 are quite surprising. Although marsupial species are phylogenetically distant from mice, my experience with non-conserved regulatory elements from much more distant species suggests that there is usually some regulatory activity observed, even if it is not reproducible or the observed expression domains are different from expected (which can be well-explained by differences in trans). It is unfortunate that the authors did not test more regulatory elements from these species and only conducted transgenesis experiments with three of these elements, as the specificity of transcription factors preventing marsupial elements from exhibiting any regulatory activity in mice would deserve further

in-depth study.

I suggest that, using transcription factor binding site analysis on these elements, the authors discuss possible conserved binding sites among different marsupial species that could explain the supposed differences in trans responsible for the lack of regulatory activity in mouse transgenesis experiments, due to differences in expression and/or function of these transcriptions factors among marsupial and non-marsupial mammals.

Apart from this, I believe that the article has significantly improved after incorporating the reviewers' suggestions and deserves publication in Nature.

Author Rebuttals to First Revision:

Referees' comments:

Referee #1 (Remarks to the Author):

The authors have successfully addressed all my concerns. I would like to highlight their commendable efforts in making their study more robust, in particular by performing additional assays for regulatory activity in two different species, among others experiments. Despite some non-conclusive results (i.e. LacZ in mouse transgenics), the overall experimental evidence makes a compelling case for the involvement of Emx2 in the evolution of the patagium in gliders.

Thank you for all your key feedback!

Referee #3 (Remarks to the Author):

The authors have done a great job in revising the manuscript and addressing the open issues.

We appreciate your help making this manuscript much stronger.

Regarding the new analyses that were added in the revision, I have two points. None of them will affect the main conclusions of this study, but it would be important to address them before publication.

1) Blat is a method that is suitable for detecting highly similar sequences, which is not the case for comparisons between marsupials, mice and humans. The authors mentioned that they also used NCBI's blastn, which should have a higher sensitivity if the right parameters are used. However, the respective methods section only mentions Blat and not blastn. This methods section needs to also describe the blastn analysis.

Thanks for pointing out that the description of the blastn analysis was missing. We have now incorporated this in our methods section. Please see Lines: 1217-1222

2) Lines 274-290: The authors tested whether the TADs of the 59 genes contained at least one enhancer bound by Emx2, which would indicate direct regulation by Emx2. For 38 of the 59 genes this was the case and a hypergeometric test showed that this is significantly more than expected. I suppose the hypergeometric test was computed considering all 11044 TADs? In other words, 39 of 59 TADs for the differentially expressed genes vs. n of 11044-59 (N) TADs. If the 59 TADs are much longer than the 11044-59 TADs, this would create a bias since longer TADs will have more enhancers. It would be necessary to correct for the length of the TAD or showing that the 59 TADs have a similar or shorter size. An alternative test would be similar to what GREAT

<https://www.nature.com/articles/nbt.1630> does. One can consider the 59 genes as genes having a certain GO term and use a binomial test over the enhancers to test whether the enhancers in the 59 TADs are significantly more often bound by Emx2. The success probability here is the proportion of Emx2 bound enhancers of all enhancers.

This is an important point to consider and we thank the reviewer for bringing it up. We agree that, in principle, longer TADs will tend to have more enhancers. However, it is worth noting that our analysis was done considering only active enhancers (i.e., regions marked by ATAC/H3k27Ac) and those are not necessarily expected to be enriched in longer TADs.

To address this comment and gain more insights into the relationship between TAD size and enhancers, we examined the size distribution of our TADs alongside the corresponding number of enhancers. Our analysis revealed a low correlation between TAD size and the number of enhancers ($R^2 = 0.05$). The majority of TADs had 25 or fewer enhancers and were between 100,000 and 1,000,000 bps in size. Notably, the TADs containing our genes of interest are no larger than other TADs and do not contain the greatest number of enhancers. Moreover, the largest TAD (8,000,000 bp) did not harbor the highest number of enhancers and the TAD containing the highest number of enhancers (~4,000,000 bps and 449 enhancers) does not contain any of our 59 genes of interest. Altogether, this analysis supports the notion that longer TADs do not contain more active enhancers.

We have added this new analysis in the new version of our manuscript. Please see Methods (Lines 1183-1193) and New Extended Data Fig. 10c-e.

Minor point

I think the main point of Figure 2d is to assess whether the glider GAR has a different (higher or lower) activity compared to the non glider GAR. It would then be helpful to also test significance by comparing the glider vs. non glider activity. For the first two GARs, this is most likely also significant, as the data for the non glider GARs looks like the control.

We agree with this suggestion and have now included this comparison. Please see the new Fig. 2d

Referee #4 (Remarks to the Author):

In this work, the authors present an interesting and thorough study on the independent emergence of an evolutionary novelty, the gliding patagium, in three species of gliding marsupials, which constitutes a compelling example of evolutionary convergence. They employ a set of techniques including comparison of different genomes to locate species-

specific accelerated regions (for which they have sequenced and assembled 15 marsupial genomes), epigenomic and transcriptomic assays, and functional studies such as transgenics in one of the gliding marsupial species, as well as gain and loss-of-function assays in this species and in mice. Based on the data obtained from these experiments, the authors highlight the crucial role of the *Emx2* gene in the developing of the patagium in these species, supporting the hypothesis that this gene has been independently upregulated through species-specific modification of its transcriptional regulation.

Overall, I think this is a very original manuscript, well written and structured and very comprehensive. The experiments are innovative and appropriate, and strongly support the proposed hypotheses. It is worth noting the use of a non-model species for some of these experiments, which poses a significant added challenge. Besides, these 15 new marsupial genome assemblies constitute a very important resource for future comparative genomics studies. The work of the three reviewers in general, and in particular that of reviewer 2, has been meticulous and has significantly contributed to the improvement of the manuscript. In my opinion, the authors have satisfactorily addressed the questions and suggestions raised by the reviewers. However, the results of the transgenesis experiments in mice suggested by reviewers 1 and 2 are quite surprising. Although marsupial species are phylogenetically distant from mice, my experience with non-conserved regulatory elements from much more distant species suggests that there is usually some regulatory activity observed, even if it is not reproducible or the observed expression domains are different from expected (which can be well-explained by differences in trans). It is unfortunate that the authors did not test more regulatory elements from these species and only conducted transgenesis experiments with three of these elements, as the specificity of transcription factors preventing marsupial elements from exhibiting any regulatory activity in mice would deserve further in-depth study.

I suggest that, using transcription factor binding site analysis on these elements, the authors discuss possible conserved binding sites among different marsupial species that could explain the supposed differences in trans responsible for the lack of regulatory activity in mouse transgenesis experiments, due to differences in expression and/or function of these transcriptions factors among marsupial and non-marsupial mammals.

We think this is a great suggestion - thank you!

In the revised version of our manuscript, we have included an analysis in which we set out to determine whether *Emx2*-associated GARs exhibit shared transcription factor binding motifs. We identified four enriched motifs within the GARs, compared to orthologous sequences from non-glider species. Among the transcription factors predicted to bind to these motifs, 13 were found to be upregulated in the sugar glider patagium. Next, we reanalyzed a scRNA-seq data set from E12.5-E14.5 laboratory mouse skin generated by our lab¹ to determine whether mouse orthologs of any of these

13 transcription factors were co-expressed in the same cells as *Emx2*. Our underlying rationale was that a lack of co-expression could potentially indicate differences in the trans environment between mouse and gliders.

Among the different cell types identified in embryonic laboratory mouse skin, *Emx2* was specifically expressed in dermal fibroblasts. Analysis of reclustered fibroblasts revealed that only 5 of the 13 transcription factors (*Maz*, *Rara*, *Sp3*, *VeZF1*, and *Znf740*) were expressed in the same subcluster of dermal fibroblasts as *Emx2*. Within this subcluster, we observed very little overlap between cells expressing these transcription factors and *Emx2*, as indicated by co-expression analysis. Thus, while the upstream factors controlling *Emx2* expression in marsupial gliders remain unknown, the results from this analysis point to potential differences in the trans regulatory environment between marsupial gliders and laboratory mice.

We completely agree with the reviewer that studying why mouse factors do not activate marsupial regulatory elements represents a fascinating avenue for future research. While providing a definitive answer to this question would require additional experiments that fall outside the scope of our study, we believe our new analysis constitutes an interesting framework for discussing possible reasons for inter-species differences in trans regulatory environments.

We have incorporated this new analysis in the revised version of our manuscript. Please see the Main text (Lines 419-433), Methods (Lines 1251-1273), and New Extended Data Fig. 9.

Apart from this, I believe that the article has significantly improved after incorporating the reviewers' suggestions and deserves publication in Nature.

Thank you for assessment and your feedback!

Reference

Johnson, M. R. *et al.* *Sfrp2* is a multifunctional regulator of rodent coat patterns. *Nature Ecology and Evolution* doi:10.1101/2022.12.12.520043.

Reviewer Reports on the Second Revision:

Referees' comments:

Referee #3 (Remarks to the Author):

The authors have fully addressed all my remaining concerns. Hence I recommend publishing this exciting manuscript.

Referee #4 (Remarks to the Author):

The authors made a significant effort to implement my suggestion. I congratulate them on this compelling manuscript, which I endorse for publication.